# Target-Oriented Soft-Robust Inverse Reinforcement Learning

## Abstract

In imitation learning, when the learning agent is at a state that is outside the demonstration of the expert, it could be difficult for her to choose an action. To overcome this challenge, inverse reinforcement learning (IRL) learns a parameterized reward function based on which we can generalize the expert's behavior to those states that are unseen in the demonstration. However, on the one hand, there could be multiple reward functions that can explain the expert's behavior, leading to reward ambiguity in IRL. On the other hand, though we often consider the transition kernel of the expert to be known to the agent, sometimes the transition kernel of the agent is different from the expert's and is unknown, leading to transition kernel ambiguity in IRL. Drawing on the notion of soft-robust optimization, we build a target-oriented soft-robust IRL (SRIRL) model where the performance of the output policy strikes a flexible balance between risk aversion and expected return maximization towards reward uncertainty in IRL. Moreover, by employing the robust satisficing framework, our SRIRL is also robust to transition kernel ambiguity in IRL. In our target-oriented SRIRL, we keep a target for the performance of the output policy that balances expected return and risk, and we minimize the constraint violation incurred by the difference between the ambiguous transition kernel and the empirical one. We derive tractable reformulation for SRIRL, and we design tailored first-order methods for SRIRL. Numerical results showcase the soft robustness towards reward uncertainty and the robustness against transition kernel ambiguity of SRIRL, as well as the stronger scalability of our first-order methods compared to a state-of-the-art commercial solver.

## 1 Introduction

In imitation learning (IL) (Hussein et al., 2017; Osa et al., 2018), we train an agent to imitate the behavior of an expert based on her demonstration, via either directly mimicking the behavior of the expert as in behavior cloning (Pomerleau, 1991) or inferring the reward function of the expert as in inverse reinforcement learning (IRL) (Hadfield-Menell et al., 2016; Kalman, 1964; Ng et al., 2000). For behavior cloning, it could be challenging to imitate the behavior of the expert in situations that are not considered in the expert demonstration. IRL addresses this challenge by learning the (parameterized) reward function from the expert demonstration that does not necessarily consider all the situations/states in the environment (Golmisheh & Shamaghdari, 2024; Zeng et al., 2024). After learning the parameters of the reward function, one can generalize the behavior of the expert to the unobserved states (by learning a policy that is optimal under the learned reward function). Other than its application in IL, IRL is also an important approach where estimating the rewards of the expert is our main interest, for example, in reward discovery for animal behavior study (Hirakawa et al., 2018; Pinsler et al., 2018; Yamaguchi et al., 2018).

IRL comes with inevitable uncertainty in the rewards, and we observe that the risk attitude towards reward uncertainty should vary and depend largely on application. For example, in autonomous driving (Alozi & Hussein, 2024; Huang et al., 2023; You et al., 2019), when an autonomous vehicle meets an object that has never been detected before, it should take a highly risk-averse attitude and thus should keep away from this object. On the contrary, for a robot vacuum cleaner, the risk of handling an unseen object would be much smaller than that in autonomous driving, and always staying away from unseen objects ensues low efficiency. Motivated by this observation, in this

paper, we adopt the notion of soft robustness (Ben-Tal et al., 2010; Lobo et al., 2020) under which the decision maker optimizes a convex combination of the mean and percentile performances. Soft-robust IRL has been popular in recent years (Brown et al., 2020b; Javed et al., 2021; Javed, 2022), via which the decision maker strikes a flexible balance between the maximization of expected return and minimization of risk towards reward uncertainty in IRL. By setting a large (respectively, small) weight parameter for the expected return in the objective function of our model, the output policy (exploited by the agent) tends to be less (respectively, more) risk-averse.

In addition to the flexibility in taking a different risk attitude towards reward uncertainty, our model is also robust to potential transition kernel ambiguity in IRL. Traditional IRL approaches often assume that the learning agent and the expert share the same transition kernel (Levine et al., 2011; Lindner et al., 2022; Ramachandran & Amir, 2007; Ratliff et al., 2006; Ziebart et al., 2008). Unfortunately, we observe that this assumption may fail to hold in practice. For example, in robot learning, the real dynamics model of the robot may differ from the empirical one, leading to possible deterioration or even failure of the trained policies (Brunke et al., 2022). To hedge against transition kernel ambiguity (of the learning agent), we apply the robust satisficing approach (Long et al., 2023) in building our model—*target-oriented* soft-robust IRL (SRIRL), where we prescribe in the constraint that the performance of the output policy should reach a user-specified target. We minimize the violation of the Bellman flow constraints (incurred by the deviation of the ambiguous transition kernel from the empirical one) in our SRIRL, where a smaller target comes with a smaller violation, thus corresponding to stronger robustness against transition kernel ambiguity. Note that the performance of the output policy of SRIRL here is taken as a weighted sum of the expected return and some risk measure of return, reflecting its soft robustness towards reward uncertainty.

We summarize our contributions as follows. (*i*) We propose the target-oriented soft-robust inverse reinforcement learning framework by which we not only achieve a balance between risk aversion and expected return maximization in the face of reward uncertainty but also perform robustly against transition kernel ambiguity in IRL. (*ii*) We provide a tractable reformulation of SRIRL (as a conic program) for computing its output policy. (*iii*) We propose tailored first-order methods for solving SRIRL that is more scalable than the Gurobi solver (Gurobi Optimization, LLC, 2022), and thus could be preferable in large-scale problems. (*iv*) In experiments, we compare our SRIRL with other benchmarks in a simulation study and a quadruped robot navigation application. Results showcase that our SRIRL achieves not only soft robustness towards reward uncertainty but also robust performance against transition kernel ambiguity.

We organize the remainder of this paper as follows. We provide necessary preliminaries in Section 2. In Section 3, we study SRIRL and provide its tractable reformulation. Tailored first-order methods for SRIRL are introduced in Section 4. Numerical experiments are conducted in Section 5. A conclusion is drawn in Section 6. Due to page limit, we conduct a survey on related work on soft-robust Markov decision processes (MDPs), robust satisficing, dual formulation of MDPs, and imitation learning in Appendix A.

**Notation.** We use boldface lowercase (*e.g.*, $\boldsymbol{r}$) and uppercase letters (*e.g.*, $\boldsymbol{A}$) to denote vectors and matrices, respectively, and their corresponding regular-font letters to represent their entries (*e.g.*, $r_i$ is the $i$-th entry of $\boldsymbol{r}$, and $A_{ij}$ is the $(i,j)$-th entry of $\boldsymbol{A}$). Special vectors and matrices are $\mathbf{e}$, $\mathbf{e}_i$, and $\boldsymbol{I}$, which denote the all-ones vector, the standard basis, and the identity matrix, respectively, whose dimensions are implied by context or will be explicitly specified otherwise. The running indices up to $S \in \mathbb{Z}_{++}$ is denoted as $[S] = \{1, \ldots, S\}$. The positive part of $x \in \mathbb{R}$ is $[x]_+ = \max\{0, x\}$. A Dirac distribution $\delta_{\boldsymbol{w}}$ concentrates unit mass at the real vector $\boldsymbol{w}$.

## 2 PRELIMINARIES

### 2.1 MARKOV DECISION PROCESSES

We model our environment by a Markov decision process (Puterman, 2014) that is represented by the tuple $< \mathcal{S}, \mathcal{A}, \boldsymbol{r}, \boldsymbol{p}, \gamma, \boldsymbol{d} >$. The state and action spaces are $\mathcal{S} = [S]$ and $\mathcal{A} = [A]$, respectively. The rewards are condensed to the vector $\boldsymbol{r} \in \mathbb{R}^{S \cdot A}$, in which the agent receives a reward $r_{s,a}$ if she takes action $a \in \mathcal{A}$ at state $s \in \mathcal{S}$. The transition kernel $\boldsymbol{p} \in (\Delta^S)^{S \cdot A}$ records the transition probabilities of the agent, where $p_{s,a,s'}$ is the probability of the agent being transitioned to the next

state $s' \in \mathcal{S}$, given that the current state is $s \in \mathcal{S}$ and the action taken is $a \in \mathcal{A}$. We use $\gamma \in (0, 1)$ to denote the discount factor and $\boldsymbol{d} \in \mathbb{R}^S_{++}$ to denote the distribution of the initial state.

## 2.2 Linear Reward Functions

Reward functions can often be expressed as a linear combination of (known) feature vectors (Abbeel & Ng, 2004; Sadigh et al., 2017; Ziebart et al., 2008) or deep neural networks (Fu et al., 2017; Ho & Ermon, 2016). We follow the former and assume that the rewards $\boldsymbol{r} \in \mathbb{R}^{S \cdot A}$ are expressable as $\boldsymbol{r} = \boldsymbol{F}\boldsymbol{w}$, where $\boldsymbol{F} \in \mathbb{R}^{S \cdot A \times K}$ is a feature matrix storing the $K$ feature vectors as its columns, and $\boldsymbol{w} \in \mathbb{R}^K$ are the (reward) weights. Therefore, the reward uncertainty can be equivalently treated as uncertainty in weights. The distribution of the reward uncertainty can be a prior distribution learned via previous tasks (Xu et al., 2019) or a posterior one learned given expert demonstration (Brown et al., 2020a; Ramachandran & Amir, 2007; Sadigh et al., 2017) or human-specified proxy reward (Hadfield-Menell et al., 2017; Ratner et al., 2018). Note that this distribution is often not directly accessible and only approximated via samples drawn by approaches such as Markov chain Monte Carlo sampling (Brown et al., 2020a; Hadfield-Menell et al., 2017; Ramachandran & Amir, 2007).

## 2.3 Risk Measures

Given a risk threshold $\varepsilon \in (0, 1)$, the value-at-risk (VaR) is defined as $\mathbb{P}\text{-VaR}_\varepsilon[\tilde{\xi}] = \sup\{x \mid \mathbb{P}[\tilde{\xi} \geq x] \geq 1 - \varepsilon\}$, and the conditional value-at-risk (CVaR, which is also referred to as average value-at-risk, expected tail risk, or expected shortfall) is defined as $\mathbb{P}\text{-CVaR}_\varepsilon[\tilde{\xi}] = \max_{x \in \mathbb{R}}\{x - (1/\varepsilon) \cdot \mathbb{E}_\mathbb{P}[[x - \tilde{\xi}]_+]\}$, where we also have $\mathbb{P}\text{-CVaR}_\varepsilon[\tilde{\xi}] = \mathbb{E}_\mathbb{P}[\tilde{\xi} \mid \tilde{\xi} \leq \mathbb{P}\text{-VaR}_\varepsilon[\tilde{\zeta}]]$ when $\mathbb{P}$ is a continuous distribution (Pflug, 2000; Rockafellar et al., 2000). Value-at-risk with a risk threshold $\varepsilon$, as implied by its definition, coincides with the lower $\varepsilon$-percentile of the random input, and the conditional value-at-risk is essentially the conditional expectation of the random input given that the random input is no larger than its VaR (with a same risk threshold $\varepsilon$). Although being a popular choice for risk-averse decision-making, computing a policy that maximizes the VaR of performance (in the face of reward uncertainty) could be confronted with a number of issues. *E.g.*, optimizing VaR could lead to an NP-hard optimization problem (Delage & Mannor, 2010). Besides, VaR does not account for the severity of losses beyond the VaR threshold. Such an ignorance of the potential impact could be problematic for those applications where rare but catastrophic events are possible to happen. In contrast to VaR, maximizing CVaR results in a convex optimization problem. CVaR takes the magnitude of losses below the VaR threshold into consideration, and it is a lower bound of VaR. As will be introduced soon, we can obtain an equivalent reformulation of our SRIRL that is equipped with CVaR as a convex optimization problem.

## 2.4 Bayesian Robust Optimization for Imitation Learning

Facing reward uncertainty in IRL, instead of optimizing an objective that is purely risk-neutral or risk-averse, BROIL (Brown et al., 2020b) seeks to maximize a soft-robust objective function that tailors a balance between average and tail performances (of the output policy):

$$
\begin{aligned}
\mathrm{T_B}(\boldsymbol{p}) = \quad &\max \quad \omega \cdot \mathbb{E}_\mathbb{P}[f(\boldsymbol{u}, \tilde{\boldsymbol{w}})] + (1 - \omega) \cdot \mathbb{P}\text{-CVaR}_\varepsilon[f(\boldsymbol{u}, \tilde{\boldsymbol{w}})] \\
&\text{s.t.} \quad \mathbf{e}^\top \boldsymbol{u}_s - \boldsymbol{p}^\top \boldsymbol{Q}_s \boldsymbol{u} - d_s = 0 \quad \forall s \in \mathcal{S} \\
&\qquad \boldsymbol{u} \in \mathbb{R}^{S \cdot A}_+.
\end{aligned}
\tag{1}
$$

Here $\boldsymbol{Q}_s = \gamma \cdot \mathrm{diag}(\mathbf{e}_s, \cdots, \mathbf{e}_s) \in \mathbb{R}^{S \cdot A \cdot S \times S \cdot A}$ and $\mathbf{e}_s$ is the $s$-th standard basis in $\mathbb{R}^S$, so it holds that $\gamma \sum_{s' \in \mathcal{S}} \sum_{a \in \mathcal{A}} p_{s',a,s} u_{s',a} = \boldsymbol{p}^\top \boldsymbol{Q}_s \boldsymbol{u}$. The weight parameter $\omega \in [0, 1]$ balances the preference between average ($\mathbb{E}_\mathbb{P}[f(\boldsymbol{u}, \tilde{\boldsymbol{w}})]$) and tail performance ($\mathbb{P}\text{-CVaR}_\varepsilon[f(\boldsymbol{u}, \tilde{\boldsymbol{w}})]$) maximization, and $\mathbb{P}$ is some given distribution of the uncertain reward weights $\tilde{\boldsymbol{w}}$, *e.g.*, the posterior distribution (given expert demonstration) of reward obtained by Bayesian IRL (Brown et al., 2019; Hadfield-Menell et al., 2017; Ramachandran & Amir, 2007; Sadigh et al., 2017). By the Bellman constraints (*i.e.*, the first $S$ constraints in (1)) and the nonnegativity condition (*i.e.*, the constraints $\boldsymbol{u} \in \mathbb{R}^{S \cdot A}_+$), we can interpret the feasible solution $u_{s,a}$ as the total discounted occupancy probability of the state-action pair $(s, a)$ when the agent extract the policy $\boldsymbol{\pi} \in \mathbb{R}^{S \cdot A}$ of BROIL as $\pi_{s,a} = u_{s,a}/(\sum_{a' \in \mathcal{A}} u_{s,a'}) \quad \forall s \in \mathcal{S}, a \in \mathcal{A}$. This interpretation implies that by solving BROIL (1),

we compute the optimal policy that maximizes a weighted sum of the mean and CVaR of the random performance $f(\boldsymbol{u}, \tilde{\boldsymbol{w}})$. Brown et al. (2020b) provide two choices of $f(\boldsymbol{u}, \tilde{\boldsymbol{w}})$, for ease of exposition, we focus only on the robust baseline regret objective $f(\boldsymbol{u}, \boldsymbol{w}) = \boldsymbol{w}^\top (\boldsymbol{F}^\top \boldsymbol{u} - \boldsymbol{f}_{\mathrm{E}})$ [1]. Here, given a set of length-$L$ demonstrated trajectories $\mathcal{T} = \{T_1, \ldots, T_{|\mathcal{T}|}\}$ with $T_t = \{s_{t,l}, a_{t,l}\}_{l \in [L]} \ \forall t \in [|\mathcal{T}|]$, $\boldsymbol{f}_{\mathrm{E}} = (1/|\mathcal{T}|) \sum_{t \in [|\mathcal{T}|]} \sum_{l \in [L]} \gamma^{l-1} \boldsymbol{f}_{s_{t,l}, a_{t,l}}$ is the empirical expected feature count of the expert demo, where the column vector $\boldsymbol{f}_{s,a}$ is the $(s, a)$-th row of the feature matrix $\boldsymbol{F}$ corresponding the feature counts of the state-action pair $(s, a) \in \mathcal{S} \times \mathcal{A}$. Therefore, for any weight vector $\boldsymbol{w} \in \mathbb{R}^K$, $f(\boldsymbol{w}, \boldsymbol{u}) = \boldsymbol{w}^\top (\boldsymbol{F}^\top \boldsymbol{u} - \boldsymbol{f}_{\mathrm{E}})$ is essentially the performance margin between BROIL and the expert, *i.e.*, the difference of the expected return $\boldsymbol{w}^\top \boldsymbol{F}^\top \boldsymbol{u}$ of the agent and the one $\boldsymbol{w}^\top \boldsymbol{f}_{\mathrm{E}}$ of the expert demonstration. Equipped with such an $f(\cdot, \cdot)$, BROIL aims to maximize a weighted sum of the average and CVaR of the uncertain performance margin with the uncertainty stemming from the uncertain reward (weights). We provide the tractable reformulation of BROIL (1) proposed by Brown et al. (2020b) in Appendix B.

## 3 Soft-Robust Inverse Reinforcement Learning

BROIL (1) only takes the nominal transition kernel into consideration. This nominal kernel is often taken as the one that is used in the expert demonstration in IRL. However, when the agent is equipped with a different (unknown) transition kernel, the performance of BROIL could be disappointing due to model mismatch. To account for the transition kernel ambiguity, we apply the robust satisficing framework (Long et al., 2023) and propose the soft-robust IRL (SRIRL) model as follows:

$$
\begin{aligned}
\mathrm{T}_{\mathrm{RS}}(\hat{\boldsymbol{p}}) = \quad &\min \quad \boldsymbol{\phi}^\top \boldsymbol{k} \\
&\text{s.t.} \quad \mathbf{e}^\top \boldsymbol{u}_s - \boldsymbol{p}^\top \boldsymbol{Q}_s \boldsymbol{u} - d_s \leq k_s \cdot \ell(\boldsymbol{p}, \hat{\boldsymbol{p}}) && \forall \boldsymbol{p} \in \mathcal{P}, s \in \mathcal{S} \\
&\quad\quad\ \mathbf{e}^\top \boldsymbol{u}_s - \boldsymbol{p}^\top \boldsymbol{Q}_s \boldsymbol{u} - d_s \geq -k_s \cdot \ell(\boldsymbol{p}, \hat{\boldsymbol{p}}) && \forall \boldsymbol{p} \in \mathcal{P}, s \in \mathcal{S} \quad (2) \\
&\quad\quad\ \omega \cdot \mathbb{E}_{\mathbb{P}}[\tilde{\boldsymbol{w}}^\top (\boldsymbol{F}^\top \boldsymbol{u} - \boldsymbol{f}_{\mathrm{E}})] + (1 - \omega) \cdot \mathbb{P}\text{-CVaR}_\varepsilon \left[ \tilde{\boldsymbol{w}}^\top (\boldsymbol{F}^\top \boldsymbol{u} - \boldsymbol{f}_{\mathrm{E}}) \right] \geq \tau \\
&\quad\quad\ \boldsymbol{u} \in \mathbb{R}_+^{S \cdot A}, \boldsymbol{k} \in \mathbb{R}_+^S,
\end{aligned}
$$

where $\hat{\boldsymbol{p}}$ is the empirical transition kernel. In the context of IRL, a natural choice for $\hat{\boldsymbol{p}}$ would be the transition kernel of the expert. The distance between the ambiguous transition kernel $\boldsymbol{p}$ and the empirical one is measured by $\ell(\boldsymbol{p}, \hat{\boldsymbol{p}})$, where common choices include a general $L_q$-norm (*i.e.*, $\ell(\boldsymbol{p}, \hat{\boldsymbol{p}}) = \|\boldsymbol{p} - \hat{\boldsymbol{p}}\|_q$) or the KL divergence. The support set $\mathcal{P} = \{\boldsymbol{p} \in \mathbb{R}_+^{S \cdot A \cdot S} \mid \mathbf{e}^\top \boldsymbol{p}_{s,a} = 1 \ \forall s \in \mathcal{S}, a \in \mathcal{A}\}$ contains all the possible values of the transition kernel, implying that our SRIRL accounts for all the possible values of the transition kernel. Comparing the $S$ Bellman flow constraints in (1) and the first two sets of constraints in (2), we observe that the decision variables $\boldsymbol{k}$ in SRIRL represent the magnitude of violation to the Bellman flow constraints. Following this interpretation, we further observe that our SRIRL, instead of maximizing the performance (of the output policy) as in BROIL, minimizes a weighted sum of the constraint violations stemming from the deviation of the ambiguous transition kernel from the empirical one. When no additional information is available, we set the weight parameters $\boldsymbol{\phi} \in \mathbb{R}_{++}^S$ as an all-ones vector because the Bellman flow constraints appear to be symmetric. In our SRIRL, the performance of the output policy is constrained to reach a user-specified target $\tau$. By varying the value for $\tau$, we can flexibly adjust the robustness (against transition kernel ambiguity) of SRIRL, where a smaller $\tau$ corresponds to stronger robustness. This is because the values of $k_s, s \in \mathcal{S}$ tend to be smaller with a smaller $\tau$.

We retrieve the optimal policy of SRIRL (2) as in BROIL. As implied by its formulation, the constraint violation of SRIRL (2) depends on the deviation of the ambiguous transition kernel $\boldsymbol{p}$ from $\hat{\boldsymbol{p}}$. When the true transition kernel is the same as the empirical one, no constraint violation will occur, and our SRIRL thus can reach its target $\tau$. Indeed, by setting the target $\tau = \mathrm{T}_{\mathrm{B}}(\hat{\boldsymbol{p}})$ in our SRIRL, the optimal policy of BROIL can be recovered.

**Proposition 1** *For any $\hat{\boldsymbol{p}} \in \mathcal{P}$, it holds that: (i) when $\tau = \mathrm{T}_{\mathrm{B}}(\hat{\boldsymbol{p}})$, any optimal solution of $\mathrm{T}_{\mathrm{RS}}(\hat{\boldsymbol{p}})$ is also optimal in $\mathrm{T}_{\mathrm{B}}(\hat{\boldsymbol{p}})$. (ii) $\mathrm{T}_{\mathrm{RS}}(\hat{\boldsymbol{p}})$ is infeasible whenever $\tau > \mathrm{T}_{\mathrm{B}}(\hat{\boldsymbol{p}})$.*

---

[1]Note that the other choice provided by Brown et al. (2020b) is $f(\boldsymbol{u}, \tilde{\boldsymbol{w}}) = \boldsymbol{w}^\top \boldsymbol{F}^\top \boldsymbol{u}$, with which our framework is also compatible but we omit it for a concise presentation.

Via the Bayesian IRL (Brown et al., 2019; Hadfield-Menell et al., 2017; Ramachandran & Amir, 2007; Sadigh et al., 2017), we have $N$ samples $\{\boldsymbol{w}_i\}_{i \in [N]}$ generated from the posterior reward weights distribution. When equipped with an empirical distribution $\hat{\mathbb{P}} = (1/N) \sum_{i \in [N]} \delta_{\boldsymbol{w}_i}$, our SRIRL (2) can be reformulated as an equivalent convex optimization problem by making use of the expression of CVaR as a maximization problem in Section 2.3.

**Proposition 2** *When equipped with a general norm $\ell(\boldsymbol{p}, \hat{\boldsymbol{p}}) = \|\boldsymbol{p} - \hat{\boldsymbol{p}}\|$ and the empirical distribution $\mathbb{P} = \hat{\mathbb{P}}$, problem (2) is equivalent to a convex optimization problem as follows:*

$$
\begin{aligned}
\mathrm{T}_{\mathrm{RS}}(\hat{\boldsymbol{p}}) = \quad &\min \quad \boldsymbol{\phi}^\top \boldsymbol{k} \\
&\text{s.t.} \quad \mathbf{e}^\top \boldsymbol{u}_s - d_s \leq -\hat{\boldsymbol{p}}^\top \overline{\boldsymbol{\beta}}_s + \hat{\boldsymbol{p}}^\top \boldsymbol{Q}_s \boldsymbol{u} && \forall s \in \mathcal{S} \\
&\quad\quad -\mathbf{e}^\top \boldsymbol{u}_s + d_s \leq -\hat{\boldsymbol{p}}^\top \underline{\boldsymbol{\beta}}_s - \hat{\boldsymbol{p}}^\top \boldsymbol{Q}_s \boldsymbol{u} && \forall s \in \mathcal{S} \\
&\quad\quad \|\overline{\boldsymbol{\beta}}_s - \boldsymbol{Q}_s \boldsymbol{u} - \boldsymbol{B}^\top \overline{\boldsymbol{\alpha}}_s\|_* \leq k_s && \forall s \in \mathcal{S} \\
&\quad\quad \|\underline{\boldsymbol{\beta}}_s + \boldsymbol{Q}_s \boldsymbol{u} - \boldsymbol{B}^\top \underline{\boldsymbol{\alpha}}_s\|_* \leq k_s && \forall s \in \mathcal{S} \\
&\quad\quad \frac{\omega}{N} \cdot \sum_{i \in [N]} \boldsymbol{w}_i^\top (\boldsymbol{F}^\top \boldsymbol{u} - \boldsymbol{f}_{\mathrm{E}}) + (1-\omega)x - \tau \geq \frac{1-\omega}{N\varepsilon} \cdot \sum_{i \in [N]} y_i \\
&\quad\quad y_i \geq x - \boldsymbol{w}_i^\top (\boldsymbol{F}^\top \boldsymbol{u} - \boldsymbol{f}_{\mathrm{E}}) && \forall i \in [N] \\
&\quad\quad \boldsymbol{u} \in \mathbb{R}_+^{S \cdot A}, \ \boldsymbol{k} \in \mathbb{R}_+^S, \ x \in \mathbb{R}, \ \boldsymbol{y} \in \mathbb{R}_+^N \\
&\quad\quad \overline{\boldsymbol{\alpha}}_s \in \mathbb{R}^{S \cdot A}, \ \underline{\boldsymbol{\alpha}}_s \in \mathbb{R}^{S \cdot A}, \ \overline{\boldsymbol{\beta}}_s \in \mathbb{R}_+^{S \cdot A \cdot S}, \ \underline{\boldsymbol{\beta}}_s \in \mathbb{R}_+^{S \cdot A \cdot S} && \forall s \in \mathcal{S},
\end{aligned}
\tag{3}
$$

*where $\boldsymbol{B} = \mathrm{diag}(\mathbf{e}^\top, \cdots, \mathbf{e}^\top) \in \mathbb{R}^{S \cdot A \times S \cdot A \cdot S}$ and $\mathbf{e} \in \mathbb{R}^S$.*

## 4 FIRST ORDER METHODS

To retrieve the optimal solution of our SRIRL (2), we can directly input its equivalent problem (3) into state-of-the-art commercial solvers such as Gurobi (Gurobi Optimization, LLC, 2022) and MOSEK (Mosek & Copenhagen, 2021). However, when the problem scales up, their computation times can grow rapidly, making them unsuitable for large-scale problems. Motivated by the strong scalability of first-order methods (Beck, 2017), we consider solving SRIRL (2) via a first-order primal-dual algorithm (PDA) (Chambolle & Pock, 2016; Esser et al., 2010; Grand-Clément & Kroer, 2021; He & Yuan, 2012) with convergence rate $\mathcal{O}(1/M)$, where $M$ is the number of iterations. We first provide an equivalent reformulation of our SRIRL (2) as a convex-concave minimax problem for which PDA is suitable, and we provide its preceding lemma, Lemma 1, in Appendix C.

**Proposition 3** *SRIRL (2) is equivalent to the minimax problem*

$$
\begin{aligned}
\min_{\boldsymbol{u}, x, \boldsymbol{y}} \max_{\substack{\mu, \boldsymbol{\eta}, \boldsymbol{\xi}, \\ \overline{\boldsymbol{\lambda}}, \underline{\boldsymbol{\lambda}}, \overline{\boldsymbol{\theta}}, \underline{\boldsymbol{\theta}}}} \quad &\mu\left(\frac{1-\omega}{N\varepsilon} \cdot \mathbf{e}^\top \boldsymbol{y} - \frac{\omega}{N} \cdot \sum_{i \in [N]} \boldsymbol{w}_i^\top (\boldsymbol{F}^\top \boldsymbol{u} - \boldsymbol{f}_{\mathrm{E}}) - (1-\omega)x + \tau\right) + \sum_{i \in [N]} \eta_i(x \\
&- \boldsymbol{w}_i^\top (\boldsymbol{F}^\top \boldsymbol{u} - \boldsymbol{f}_{\mathrm{E}}) - y_i) + \sum_{s \in \mathcal{S}} \left\{ (\overline{\lambda}_s - \underline{\lambda}_s)(\mathbf{e}^\top \boldsymbol{u}_s - d_s) - (\overline{\boldsymbol{\theta}}_s - \underline{\boldsymbol{\theta}}_s)^\top \boldsymbol{Q}_s \boldsymbol{u} \right\} \\
\text{s.t.} \quad &(\overline{\lambda}_s, \overline{\boldsymbol{\theta}}_s) \in \mathcal{L}_q(\xi_s), \ (\underline{\lambda}_s, \underline{\boldsymbol{\theta}}_s) \in \mathcal{L}_q(\phi_s - \xi_s) && \forall s \in \mathcal{S} \\
&\boldsymbol{u} \in \mathbb{R}_+^{S \cdot A}, \ x \in \mathbb{R}, \ \boldsymbol{y} \in \mathbb{R}_+^N, \ \mu \in \mathbb{R}_+, \ \boldsymbol{\eta} \in \mathbb{R}_+^N, \ \boldsymbol{\xi}, \overline{\boldsymbol{\lambda}}, \underline{\boldsymbol{\lambda}} \in \mathbb{R}_+^S, \overline{\boldsymbol{\theta}}, \underline{\boldsymbol{\theta}} \in \mathbb{R}_+^{S \cdot S \cdot A \cdot S}
\end{aligned}
\tag{4}
$$

*when choosing a general $L_q$-norm $\ell(\boldsymbol{p}, \hat{\boldsymbol{p}}) = \|\boldsymbol{p} - \hat{\boldsymbol{p}}\|_q$ for arbitrary $q \in [1, \infty]$ in (2). Here $\mathcal{L}_q(\xi) = \{(\lambda, \boldsymbol{\theta}) \in \mathbb{R}_+ \times \mathbb{R}_+^{S \cdot A \cdot S} : \|\boldsymbol{\theta} - \lambda \cdot \hat{\boldsymbol{p}}\|_q \leq \xi, \lambda \cdot \mathbf{e} = \boldsymbol{B}\boldsymbol{\theta}\}$ for any $\xi \in \mathbb{R}_+$.*

In the remainder, we focus on our SRIRL (2) equipped with an $L_\infty$-norm, and we provide the pseudocode of the PDA in Algorithm 1. Here for any $(\mu, \boldsymbol{\eta}, \overline{\boldsymbol{\lambda}}, \underline{\boldsymbol{\lambda}}, \overline{\boldsymbol{\theta}}, \underline{\boldsymbol{\theta}}) \in \mathbb{R}_+ \times \mathbb{R}_+^N \times \mathbb{R}_+^S \times \mathbb{R}_+^S \times$

---

**Algorithm 1** Primal-Dual Algorithm (PDA) for Problem (2)

---

**Input:** Initial feasible solution $(\boldsymbol{u}^0, x^0, \boldsymbol{y}^0, \mu^0, \boldsymbol{\eta}^0, \boldsymbol{\xi}^0, \overline{\boldsymbol{\lambda}}^0, \underline{\boldsymbol{\lambda}}^0, \overline{\boldsymbol{\theta}}^0, \underline{\boldsymbol{\theta}}^0)$ for problem (4), stepsizes $\nu, \sigma > 0$, $k \leftarrow 0$
**while** $\|\boldsymbol{u}^k - \boldsymbol{u}^{k-1}\|_\infty \geq \varepsilon$ **do**
    // $Step\,1:\,Primal\,update$
    $(\boldsymbol{u}^{k+1}, x^{k+1}, \boldsymbol{y}^{k+1}) \leftarrow \mathscr{P}(\mu^k, \boldsymbol{\eta}^k, \overline{\boldsymbol{\lambda}}^k, \underline{\boldsymbol{\lambda}}^k, \overline{\boldsymbol{\theta}}^k, \underline{\boldsymbol{\theta}}^k; \boldsymbol{u}^k, x^k, \boldsymbol{y}^k)$
    // $Step\,2:\,Dual\,update$
    $(\mu^{k+1}, \boldsymbol{\eta}^{k+1}, \boldsymbol{\xi}^{k+1}, \overline{\boldsymbol{\lambda}}^{k+1}, \underline{\boldsymbol{\lambda}}^{k+1}, \overline{\boldsymbol{\theta}}^{k+1}, \underline{\boldsymbol{\theta}}^{k+1}) \leftarrow \mathscr{D}(2\boldsymbol{u}^{k+1} - \boldsymbol{u}^k, 2x^{k+1} - x^k, 2\boldsymbol{y}^{k+1} - \boldsymbol{y}^k; \mu^k, \boldsymbol{\eta}^k, \boldsymbol{\xi}^k, \overline{\boldsymbol{\lambda}}^k, \underline{\boldsymbol{\lambda}}^k, \overline{\boldsymbol{\theta}}^k, \underline{\boldsymbol{\theta}}^k)$
    $k \leftarrow k + 1$
**end while**
**Output:** Solution $(\boldsymbol{u}^{\mathrm{avg}}, x^{\mathrm{avg}}, \boldsymbol{y}^{\mathrm{avg}}) = (1/k) \sum_{i=1}^{k}(\boldsymbol{u}^i, x^i, \boldsymbol{y}^i)$, $(\mu^{\mathrm{avg}}, \boldsymbol{\eta}^{\mathrm{avg}}, \boldsymbol{\xi}^{\mathrm{avg}}, \overline{\boldsymbol{\lambda}}^{\mathrm{avg}}, \underline{\boldsymbol{\lambda}}^{\mathrm{avg}}, \overline{\boldsymbol{\theta}}^{\mathrm{avg}}, \underline{\boldsymbol{\theta}}^{\mathrm{avg}}) = (1/k) \sum_{i=1}^{k}(\mu^i, \boldsymbol{\eta}^i, \boldsymbol{\xi}^i, \overline{\boldsymbol{\lambda}}^i, \underline{\boldsymbol{\lambda}}^i, \overline{\boldsymbol{\theta}}^i, \underline{\boldsymbol{\theta}}^i)$

---

$\mathbb{R}_+^{S \cdot S \cdot A \cdot S} \times \mathbb{R}_+^{S \cdot S \cdot A \cdot S}$ and $(\boldsymbol{u}', x', \boldsymbol{y}') \in \mathbb{R}_+^{S \cdot A} \times \mathbb{R} \times \mathbb{R}_+^N$, the primal update operator is

$$
\begin{aligned}
&\mathscr{P}(\mu, \boldsymbol{\eta}, \overline{\boldsymbol{\lambda}}, \underline{\boldsymbol{\lambda}}, \overline{\boldsymbol{\theta}}, \underline{\boldsymbol{\theta}}; \boldsymbol{u}', x', \boldsymbol{y}') \\
= &\operatorname*{arg\,min}_{\boldsymbol{u} \in \mathbb{R}_+^{S \cdot A},\, x \in \mathbb{R}, \boldsymbol{y} \in \mathbb{R}_+^N} \mu\left(\frac{1-\omega}{N\varepsilon} \cdot \mathbf{e}^\top \boldsymbol{y} - \frac{\omega}{N} \cdot \sum_{i \in [N]} \boldsymbol{w}_i^\top \boldsymbol{F}^\top \boldsymbol{u} - (1-\omega)x\right) + \sum_{i \in [N]} \eta_i(x - \boldsymbol{w}_i^\top \boldsymbol{F}^\top \boldsymbol{u} - y_i) \\
& + \sum_{s \in \mathcal{S}}\left\{(\overline{\lambda}_s - \underline{\lambda}_s)\mathbf{e}^\top \boldsymbol{u}_s - (\overline{\boldsymbol{\theta}}_s - \underline{\boldsymbol{\theta}}_s)^\top \boldsymbol{Q}_s \boldsymbol{u}\right\} + \frac{1}{2\nu}\left(\|\boldsymbol{u} - \boldsymbol{u}'\|_2^2 + (x - x')^2 + \|\boldsymbol{y} - \boldsymbol{y}'\|_2^2\right),
\end{aligned}
\tag{5}
$$

and the dual update operator is

$$
\begin{aligned}
&\mathscr{D}(\boldsymbol{u}, x, \boldsymbol{y}; \mu', \boldsymbol{\eta}', \boldsymbol{\xi}', \overline{\boldsymbol{\lambda}}', \underline{\boldsymbol{\lambda}}', \overline{\boldsymbol{\theta}}', \underline{\boldsymbol{\theta}}') \\
= &\begin{cases}
\operatorname*{arg\,min}_{\mu, \boldsymbol{\eta}, \boldsymbol{\xi}, \overline{\boldsymbol{\lambda}}, \underline{\boldsymbol{\lambda}}, \overline{\boldsymbol{\theta}}, \underline{\boldsymbol{\theta}}} & \mu\left(-\frac{1-\omega}{N\varepsilon} \cdot \mathbf{e}^\top \boldsymbol{y} + \frac{\omega}{N} \cdot \sum_{i \in [N]} \boldsymbol{w}_i^\top (\boldsymbol{F}^\top \boldsymbol{u} - \boldsymbol{f}_{\mathrm{E}}) + (1-\omega)x - \tau\right) \\
& + \sum_{i \in [N]} \eta_i(-x + \boldsymbol{w}_i^\top (\boldsymbol{F}^\top \boldsymbol{u} - \boldsymbol{f}_{\mathrm{E}}) + y_i) + \sum_{s \in \mathcal{S}}\left\{(\overline{\lambda}_s - \underline{\lambda}_s)(d_s - \mathbf{e}^\top \boldsymbol{u}_s)\right. \\
& \left. + (\overline{\boldsymbol{\theta}}_s - \underline{\boldsymbol{\theta}}_s)^\top \boldsymbol{Q}_s \boldsymbol{u}\right\} + \frac{1}{2\sigma} \cdot \left((\mu - \mu')^2 + \|\boldsymbol{\eta} - \boldsymbol{\eta}'\|_2^2 + \|\boldsymbol{\xi} - \boldsymbol{\xi}'\|_2^2\right. \\
& \left. + \|\overline{\boldsymbol{\lambda}} - \overline{\boldsymbol{\lambda}}'\|_2^2 + \|\underline{\boldsymbol{\lambda}} - \underline{\boldsymbol{\lambda}}'\|_2^2 + \|\overline{\boldsymbol{\theta}} - \overline{\boldsymbol{\theta}}'\|_2^2 + \|\underline{\boldsymbol{\theta}} - \underline{\boldsymbol{\theta}}'\|_2^2\right) \\
\text{s.t.} & \mu \in \mathbb{R}_+, \boldsymbol{\eta} \in \mathbb{R}_+^N, \boldsymbol{\xi} \in \mathbb{R}_+^S, (\overline{\lambda}_s, \overline{\boldsymbol{\theta}}_s) \in \mathcal{L}_\infty(\xi_s), (\underline{\lambda}_s, \underline{\boldsymbol{\theta}}_s) \in \mathcal{L}_\infty(\phi_s - \xi_s) \quad \forall s \in \mathcal{S}
\end{cases}
\end{aligned}
\tag{6}
$$

for any $(\boldsymbol{u}, x, \boldsymbol{y}) \in \mathbb{R}_+^{S \cdot A} \times \mathbb{R} \times \mathbb{R}_+^N$ and $(\mu', \boldsymbol{\eta}', \boldsymbol{\xi}', \overline{\boldsymbol{\lambda}}', \underline{\boldsymbol{\lambda}}', \overline{\boldsymbol{\theta}}', \underline{\boldsymbol{\theta}}') \in \mathbb{R}_+ \times \mathbb{R}_+^N \times \mathbb{R}_+^S \times \mathbb{R}_+^S \times \mathbb{R}_+^S \times \mathbb{R}_+^{S \cdot S \cdot A \cdot S} \times \mathbb{R}_+^{S \cdot S \cdot A \cdot S}$, where $\nu, \sigma > 0$ are stepsizes of the primal and dual updates, respectively. We provide the result of the convergence rate $\mathcal{O}(1/M)$ of Algorithm 1 in Theorem 1 in Appendix C.

Comparing the primal and dual updates, we observe that the former requires solving (5) with $\mathcal{O}(SA + N)$ decision variables and $\mathcal{O}(SA + N)$ constraints, while the latter requires solving (6) with $\mathcal{O}(S^3A + N)$ decision variables and $\mathcal{O}(S^3A + N)$ constraints, impling that the bottleneck of computation time lies in the dual update. Fortunately, we can decompose (6) into $S+2$ subproblems. Two of them requires solving totally $\mathcal{O}(N)$ single-variable quadratic programs, and each allows an analytical solution. For the other $S$ ones, thanks to the choice $\ell(\boldsymbol{p}, \hat{\boldsymbol{p}}) = \|\boldsymbol{p} - \hat{\boldsymbol{p}}\|_\infty$ in SRIRL (2), each of them allows tailored algorithms and further decomposition for efficient solution, as we shall see in Section 4.2. For the primal update (5), we also provide a tailored algorithm that decomposes it into $\mathcal{O}(SA + N)$ single-variable quadratic programs that can be solved analytically in Section 4.1.

## 4.1 Tailored Algorithm for Fast Primal Update

We provide the time complexity for obtaining the analytical solution of (5) in the following proposition. The analytical solution is provided in the proof in Appendix C.

**Proposition 4** *Problem* (5) *can be solved in time* $\mathcal{O}(S^2 A + N)$.

## 4.2 Tailored Algorithms for Fast Dual Update

Note that in (6), every term in the objective function exclusively includes only one of $\mu$, $\boldsymbol{\eta}$, or $\{(\xi_{\hat{s}}, \overline{\lambda}_{\hat{s}}, \underline{\lambda}_{\hat{s}}, \overline{\boldsymbol{\theta}}_{\hat{s}}, \underline{\boldsymbol{\theta}}_{\hat{s}})\}_{\hat{s} \in \mathcal{S}}$, so does every constraint. We thus decompose (6) into $S + 2$ subproblems:

$$\mathscr{D}^{\mu}(\boldsymbol{u}, x, \boldsymbol{y}; \mu') = \underset{\mu \in \mathbb{R}_+}{\arg \min} \, \mu \left( -\frac{1-\omega}{N\varepsilon} \cdot \mathbf{e}^{\top} \boldsymbol{y} + \frac{\omega}{N} \cdot \sum_{i \in [N]} \boldsymbol{w}_i^{\top} (\boldsymbol{F}^{\top} \boldsymbol{u} - \boldsymbol{f}_{\mathrm{E}}) + (1-\omega)x - \tau \right) + \frac{1}{2\sigma} \cdot (\mu - \mu')^2,$$

$$\mathscr{D}^{\boldsymbol{\eta}}(\boldsymbol{u}, x, \boldsymbol{y}; \boldsymbol{\eta}') = \underset{\boldsymbol{\eta} \in \mathbb{R}_+^N}{\arg \min} \, \sum_{i \in [N]} \eta_i (-x + \boldsymbol{w}_i^{\top} (\boldsymbol{F}^{\top} \boldsymbol{u} - \boldsymbol{f}_{\mathrm{E}}) + y_i) + \frac{1}{2\sigma} \cdot \|\boldsymbol{\eta} - \boldsymbol{\eta}'\|_2^2,$$

and $\mathscr{D}_{\hat{s}}(\boldsymbol{u}; \xi_{\hat{s}}, \overline{\lambda}_{\hat{s}}, \underline{\lambda}_{\hat{s}}, \overline{\boldsymbol{\theta}}_{\hat{s}}, \underline{\boldsymbol{\theta}}_{\hat{s}}), \hat{s} \in \mathcal{S}$ where

$$
\begin{aligned}
\mathscr{D}_{\hat{s}}(\boldsymbol{u}; \xi', \overline{\lambda}', \underline{\lambda}', \overline{\boldsymbol{\theta}}', \underline{\boldsymbol{\theta}}') = \underset{\xi, \overline{\lambda}, \underline{\lambda}, \overline{\boldsymbol{\theta}}, \underline{\boldsymbol{\theta}}}{\arg \min} \quad & (\overline{\lambda} - \underline{\lambda})(d_{\hat{s}} - \mathbf{e}^{\top} \boldsymbol{u}_{\hat{s}}) + (\overline{\boldsymbol{\theta}} - \underline{\boldsymbol{\theta}})^{\top} \boldsymbol{Q}_{\hat{s}} \boldsymbol{u} + \frac{1}{2\sigma} \cdot \Big( (\xi - \xi')^2 \\
& + (\overline{\lambda} - \overline{\lambda}')^2 + (\underline{\lambda} - \underline{\lambda}')^2 + \|\overline{\boldsymbol{\theta}} - \overline{\boldsymbol{\theta}}'\|_2^2 + \|\underline{\boldsymbol{\theta}} - \underline{\boldsymbol{\theta}}'\|_2^2 \Big) \\
\text{s.t.} \quad & (\overline{\lambda}, \overline{\boldsymbol{\theta}}) \in \mathcal{L}_{\infty}(\xi), (\underline{\lambda}, \underline{\boldsymbol{\theta}}) \in \mathcal{L}_{\infty}(\phi_{\hat{s}} - \xi), \xi \in \mathbb{R}_+.
\end{aligned}
\tag{7}
$$

Problem $\mathscr{D}^{\mu}(\boldsymbol{u}, x, \boldsymbol{y}; \mu')$ is a single-variable quadratic program, admitting an analytical solution obtainable in time $\mathcal{O}(N + SA)$.

**Proposition 5** *Problem* $\mathscr{D}^{\mu}(\boldsymbol{u}, x, \boldsymbol{y}; \mu')$ *can be solved in time* $\mathcal{O}(N + SA)$.

Problem $\mathscr{D}^{\boldsymbol{\eta}}(\boldsymbol{u}, x, \boldsymbol{y}; \boldsymbol{\eta}')$ is decomposable into $N$ single-variable quadratic programs, each of which can be analytically solved in time $\mathcal{O}(SA)$.

**Proposition 6** *Problem* $\mathscr{D}^{\boldsymbol{\eta}}(\boldsymbol{u}, x, \boldsymbol{y}; \boldsymbol{\eta}')$ *can be solved in time* $\mathcal{O}(NSA)$.

Problem $\mathscr{D}_{\hat{s}}(\boldsymbol{u}; \xi', \overline{\lambda}', \underline{\lambda}', \overline{\boldsymbol{\theta}}', \underline{\boldsymbol{\theta}}')$ can be treated as a min-min problem where we apply golden section search (Truhar & Veselić, 2009) to locate the optimal $\xi^{\star}$ for the outer minimization. For the inner one for computing the optimal $(\overline{\lambda}^{\star}, \overline{\boldsymbol{\theta}}^{\star}, \underline{\lambda}^{\star}, \underline{\boldsymbol{\theta}}^{\star})$, we can further decompose it into two subproblems, one for locating $(\overline{\lambda}^{\star}, \overline{\boldsymbol{\theta}}^{\star})$ and the other one for $(\underline{\lambda}^{\star}, \underline{\boldsymbol{\theta}}^{\star})$. Both subproblems share the same problem structure, thus share the same efficient tailored algorithm. To solve for $(\overline{\lambda}^{\star}, \overline{\boldsymbol{\theta}}^{\star})$, we can again rewrite the corresponding subproblem as a min-min problem, where we use golden section search for computing the optimal $\overline{\lambda}^{\star}$ for the outer minimization, and we decompose the inner problem for $\overline{\boldsymbol{\theta}}^{\star}$ into $SA$ subproblem (due to the choice of an $L_{\infty}$-norm in our SRIRL (2)). For any $(s, a) \in \mathcal{S} \times \mathcal{A}$, the $(s, a)$-th subproblem can be solved in time $\mathcal{O}(S \log S)$ by our tailored algorithm (as shown in Proposition 13 in Appendix D). Due to page limit, we provide the technical details of our tailored algorithms for problem (7) in Appendix D, and we only provide the time complexity as follows.

**Proposition 7** *Problem* (7) *can be solved in time* $\mathcal{O}(S^2 A \log S (\log(\delta^{-1}))^2)$, *where* $\delta > 0$ *denotes the precision of the golden section search.*

Summarizing Propositions 5, 6, and 7, we provide the time complexity of our tailored algorithm for conducting the dual update in Algorithm 1 (*i.e.*, solving problem (6)) in the follows.

**Proposition 8** *The output of the dual update phase in Algorithm 1 is computable in time* $\mathcal{O}(S^3 A \log S (\log(\delta^{-1}))^2 + NSA)$, *where* $\delta > 0$ *denotes the precision of the golden section search.*

## 4.3 Randomized Block Coordinate Gradient Descent for Dual Update

Even when equipped with the tailored algorithm in Section 4.2, the dual update remains to be the bottleneck in computation time due to its remarkably larger number of decision variables ($\mathcal{O}(S^3 A + N)$) than that of the primal update ($\mathcal{O}(SA + N)$). Besides, the optimal policy of our SRIRL (2) depends

Figure 1: Demonstration trajectory (represented by the arrows) in the lava corridor.

only on the optimal primal solution $\boldsymbol{u}^\star$ in the minimax problem (4), not on the dual ones. Motivated by the randomized block coordinate gradient descent, we modify the dual update of Algorithm 1 and propose a variant of it called $\text{PDA}_{\text{block}}$. We divide the dual variables into $(S+1)$ groups, where the $s$-th group is $(\xi_s, \overline{\lambda}_s, \overline{\boldsymbol{\theta}}_s, \underline{\lambda}_s, \underline{\boldsymbol{\theta}}_s)$ for any $s \in \mathcal{S}$, and the $(S+1)$-th group is $(\mu, \boldsymbol{\eta})$. In each iteration of $\text{PDA}_{\text{block}}$, we only randomly choose one group to conduct the dual update. Specifically, instead of the dual update in Algorithm 1, with probability $1/S$, we sample $\hat{s} \in \mathcal{S}$ randomly and conduct $(\xi_{\hat{s}}^{k+1}, \overline{\lambda}_{\hat{s}}^{k+1}, \underline{\lambda}_{\hat{s}}^{k+1}, \overline{\boldsymbol{\theta}}_{\hat{s}}^{k+1}, \boldsymbol{\theta}_{\hat{s}}^{k+1}) \leftarrow \mathscr{D}_{\hat{s}}(2\boldsymbol{u}^{k+1} - \boldsymbol{u}^k; \xi_{\hat{s}}^k, \overline{\lambda}_{\hat{s}}^k, \underline{\lambda}_{\hat{s}}^k, \overline{\boldsymbol{\theta}}_{\hat{s}}^k, \boldsymbol{\theta}_{\hat{s}}^k)$; with probability $(S-1)/S$, we conduct $\mu^{k+1} \leftarrow \mathscr{D}^\mu(2\boldsymbol{u}^{k+1} - \boldsymbol{u}^k, 2x^{k+1} - x^k, 2\boldsymbol{y}^{k+1} - \boldsymbol{y}^k; \mu^k)$ and $\boldsymbol{\eta}^{k+1} \leftarrow \mathscr{D}^{\boldsymbol{\eta}}(2\boldsymbol{u}^{k+1} - \boldsymbol{u}^k, 2x^{k+1} - x^k, 2\boldsymbol{y}^{k+1} - \boldsymbol{y}^k; \boldsymbol{\eta}^k)$. The time complexity of the dual update, under this strategy, drops remarkably from $\mathcal{O}(S^3 A \log S(\log(\delta^{-1}))^2 + NSA)$ to either $\mathcal{O}(NSA)$ (with probability $(S-1)/S)$) or $\mathcal{O}(S^2 A \log S(\log(\delta^{-1}))^2)$ (with probability $1/S$).

## 5  Numerical Experiments

We compare the performances our SRIRL with BROIL (Brown et al., 2020b), maximum entropy IRL (MAXENT) (Ziebart et al., 2008) and linear programming apprenticeship learning (LPAL) (Syed et al., 2008) in the *lava corridor* environment (Brown et al., 2020b) (in Section 5.1) and a quadruped robot navigation application (in Section 5.2), and we provide their detailed settings in Appendices F and G, respectively. Section 5.3 compares the scalabilities of our first-order methods and Gurobi (Gurobi Optimization, LLC, 2022), with detailed setting provided in Appendix H.1. All implementations of the experiments within this section are included in `https://github.com/ICLR-2025/SRIRL.git` to facilitate the replication of experimental results.

### 5.1  Simulation: Lava Corridor

Consider a learning agent who can only access a demonstration trajectory of the expert in an MDP (Figure 1). In the trajectory, the expert demonstrates a preference for staying away from the red cells and approaching the terminal state (*i.e.*, the right bottom cell), but the agent does not know the rewards of the red and white cells, therefore does not know whether taking a shortcut by walking on the red cells is an optimal choice. We have only features "red" and "white" in the corridor. The state space consists of the locations of this corridor thus $S = 15$. We are only allowed to take actions "left", "right", "up", "down" representing the direction towards which we move. The discount factor is $\gamma = 0.99$, and the initial distribution is a discrete uniform distribution over all states. To simulate the scene where the transition kernel of the agent $\boldsymbol{p}^{\text{ag}}$ deviates from the one of the expert $\boldsymbol{p}^{\text{ex}}$ and becomes ambiguous, we obtain $\boldsymbol{p}^{\text{ag}}$ by polluting $\boldsymbol{p}^{\text{ex}}$, so that when the agent chooses to move towards a certain direction, she could possibly "slip" to the neighboring cells, and a higher pollution rate corresponds to a higher probability of slipping (see more details in Appendix F.2).

As in Brown et al. (2020b), we generate 2000 weight samples from the posterior reward weights distribution $\mathbb{P}(\mathcal{D}|\boldsymbol{w})$ via Bayesian IRL (Ramachandran & Amir, 2007) for training BROIL and SRIRL (2) (see Appendix F.1 for details). We compare SRIRL with BROIL, MAXENT, and LPAL. The support set $\mathcal{P}$ in our SRIRL is modified so that only the possible next states of the agent are allowed for nonzero transition probabilities. Note that all our theoretical results for SRIRL, as well as our tailored PDA can adapt to such a modification (see Appendix F.3). More details for our implementation of MAXENT and LPAL can be found in Appendices F.4 and F.5, respectively.

We report the results for $\omega = 0$ in Figure 2, and we relegate the results for $\omega = 0.5$ (Figure 6) and $\omega = 1$ (Figure 7) to Appendix F.6. We can observe that, BROIL and SRIRL perform better than MAXENT and LPAL under all pollution rates. Comparing BROIL with our SRIRL, we can see that BROIL performs better than SRIRL only under low pollution. Its performance degrades faster than

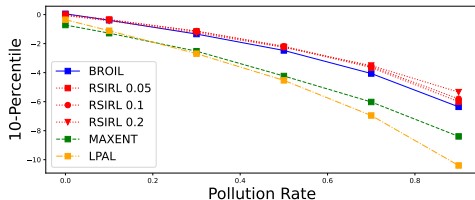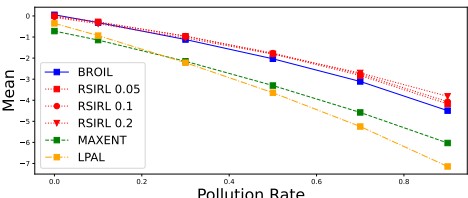

Figure 2: Percentile and average performances in the *lava corridor* application. For each nonzero pollution rate, we evaluate the models over 1000 randomly generated polluted transition kernels. For each kernel, we take the weighted average of the mean (with weight $\omega$) and CVaR (with weight $1-\omega$) of our random instances, where each instance corresponds to a weight sample generated from $\mathbb{P}(\boldsymbol{w} \mid \mathcal{D})$ and evaluates the regret (corresponding the current transition kernel and weights/rewards). Here we set $\omega = 0$, and the numbers in the legend are the difference between the optimal value of BROIL and the target parameter $\tau$ in SRIRL.

SRIRL and could quickly become notably worse than SRIRL, reflecting the stronger robustness to transition kernel ambiguity of SRIRL than BROIL. We can also observe that, by setting a lower target in SRIRL, we can trade off slightly worse performance under low pollution against notably better performance under high pollution. We also refer interested readers to Appendix F.6 where we take a closer look at the policies of SRIRL to see how soft robustness is achieved.

### 5.2 Application: Quadruped Robot Navigation

In robotics, a fundamental task is navigation, which involves controlling a robot to reach a target position. For quadruped robots, the dynamics model is overly complex, and so navigation is often handled using a two-layered framework: the upper layer is a decision-making module to compute the command speed, and the lower layer is a locomotion control module used to execute the speed commands from the upper layer and control the robot's motors. However, since it is difficult for the upper-level decisions to fully account for the complex dynamics of lower-level motion control, the navigation performance is often suboptimal. This section aims to apply BROIL and SRIRL to train the upper-level policy and then compare their policies on the robot. The environment is a 2-D plane, the state is the location of the robot (with the ranges for the x and y coordinates both being $[0, 2.5]$), and the action is the desired velocity. The goal of the robot is to reach the center of the state space (*i.e.*, location $(1.25, 1.25)$). For the lower-level motion policy, we used a neural network controller. This controller can output motor control commands based on input speed to track the robot's desired speed. Ideally, the tracking error of this neural network controller should be minimal. However, due to the facts that (*i*) the dynamics of a quadruped robot could be far more complex than 2D point-mass kinematics, and that (*ii*) the neural network motion controller may fail to perfectly achieve the desired speed, the realistic transition kernel of the robot may deviate from the empirical one. This motivates us to apply SRIRL in pursuit of robustness against such transition kernel ambiguity. See Appendix G for detailed settings of the environment.

We train BROIL and SRIRL and deploy their output policies on a Unitree A1 Robot in the PyBullet simulation environment. Results show that SRIRL can deliver performance that is more robust than that of BROIL: as shown in Figure 3, the robot under BROIL falls during its navigation, while SRIRL successfully navigates to the target point. The difference in navigation performance between SRIRL and BROIL is due to the mechanical structure of the A1 robot, which could result in a significant deviation (of the true transition kernel) from the ideal 2D point mass motion model. In particular, the optimal policy of BROIL is even unable to keep the robot balanced. In contrast, our SRIRL approach takes into account the transition kernel ambiguity, resulting in a more robust strategy.

### 5.3 Scalability of Algorithms

This section compares our tailored algorithms with Gurobi (Gurobi Optimization, LLC, 2022) in terms of computation times. Tables 1 and 3 (see Table 3 in Appendix H.2) report the computation time of Gurobi and our tailored algorithms for solving SRIRL (2) under different problem sizes, where a larger MDP size ($S$ and $A$) and a larger (reward) weight sample size ($N$) both correspond

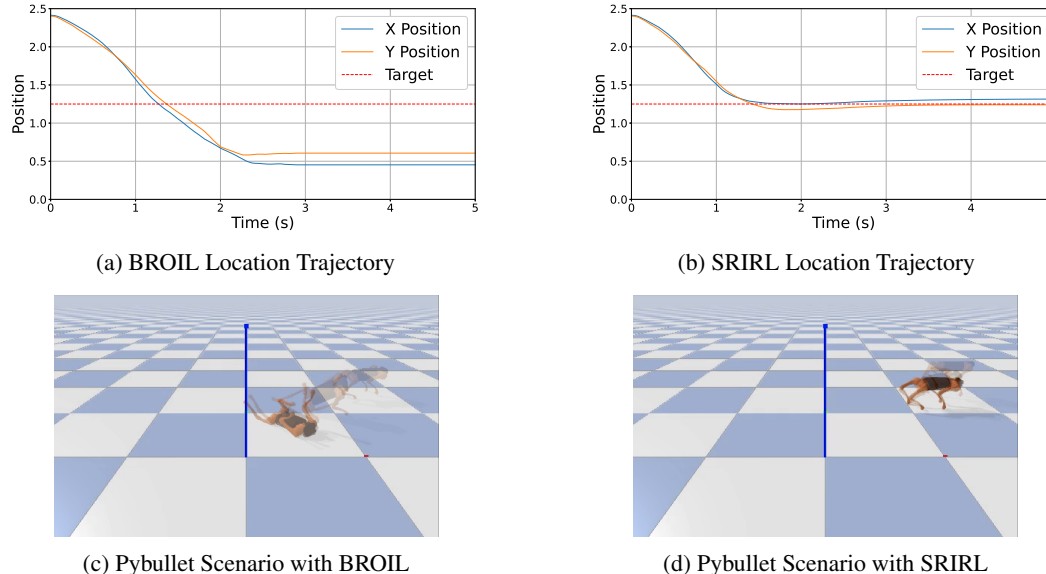

(a) BROIL Location Trajectory

(b) SRIRL Location Trajectory

(c) Pybullet Scenario with BROIL

(d) Pybullet Scenario with SRIRL

Figure 3: Experimental results of quadruped robot navigation. The initial position is $(2.4, 2.4)$.

to a larger problem size. Results shows that SRIRL is computationally challenging, where a small increase in MDP size ensues a notable increase in computation time. This is exactly the motivation behind our design of tailored first-order methods, which remains gracefully scalable under different MDP sizes (Table 1) and weight sample size (Table 3) compared to Gurobi. This matches the advantage of first-order methods that can solve problems to moderate accuracy with high efficiency.

Table 1: The average computation times (in seconds) for SRIRL under difference sizes of state $(S)$ and action $(A)$ spaces, the ratios of computation times, and the relative gaps to optimal values computed by Gurobi. The average is taken over 50 random instances. We fix $N = 10000$ throughout. The time limit for algorithms is 3600 seconds. The dash line indicates that the cell is inapplicable.

| | Computation times | | | Ratio of computation times | | Relative gaps (%) | |
|---|---|---|---|---|---|---|---|
| $S = A$ | Gurobi | PDA | PDA$_{block}$ | Gurobi/PDA | Gurobi/PDA$_{block}$ | PDA | PDA$_{block}$ |
| 15 | **19.1** | 69.8 | 168.7 | 0.27 | 0.11 | 4.3 | $< 0.1$ |
| 20 | **85.9** | 305.7 | 372.6 | 0.28 | 0.23 | 4.5 | $< 0.1$ |
| 25 | **328.1** | 704.7 | 564.1 | 0.47 | 0.58 | 4.3 | $< 0.1$ |
| 30 | 1363.6 | 1066.9 | **828.6** | 1.28 | 1.65 | 4.1 | $< 0.1$ |
| 35 | — | 2280.7 | **1512.9** | — | — | 3.7 | $< 0.1$ |

## 6 CONCLUSION

We propose target-oriented SRIRL whose output policy is soft-robust towards reward uncertainty and robust against transition kernel ambiguity in IRL. In SRIRL, the performance of the output policy is taken as a weighted sum of the average and CVaR of return, and this soft-robust performance is constrained to reach a user-specified target. This constraint is strictly imposed under the empirical transition kernel, and softly imposed under all other possible ones. The violation of the Bellman flow constraints is minimized in SRIRL, and a smaller target ensues stronger robustness of SRIRL. We reformulate SRIRL as a min-max problem where we design scalable tailored first-order methods for an efficient solution. Experiments demonstrate the soft robustness and robustness of SRIRL, as well as the strong scalability of our tailored algorithms. A promising avenue for future work would be extending SRIRL to the setting of continuous state and action spaces.

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

## A  RELATED WORK

Soft-robust MDPs are a popular line of research motivated by the notion of soft robustness that was first introduced by Ben-Tal et al. (2010). There are two different ways to model soft robustness in MDPs: (*i*) optimizing a weighted average of expected return and percentile performance (Brown et al., 2020b; Javed et al., 2021; Javed, 2022; Lobo et al., 2020). (*ii*) Optimizing the expected return with the constraint that the percentile performance is larger than some user-specified lower bound (Chow & Ghavamzadeh, 2014; Prashanth, 2014). Our SRIRL follows the first way to achieve soft robustness but is fundamentally different from existing works. First, those models that are soft-robust towards reward uncertainty (Brown et al., 2020b; Javed et al., 2021; Javed, 2022) are not robust against transition kernel ambiguity, while our SRIRL also performs robustly against transition kernel ambiguity. Second, compared to Lobo et al. (2020) where the soft-robustness is towards transition kernel uncertainty and where obtaining the exact solution is NP-hard, our SRIRL is soft-robust towards reward uncertainty and allows an exact solution via solving a conic program, and is also robust against transition kernel ambiguity via the robust satisficing framework.

There has been a surge in the study of the robust satisficing (RS) framework in recent years (Liu et al., 2023; Long et al., 2023; Ramachandra et al., 2021; Sim et al., 2022). In comparison to the robust optimization (Ben-Tal et al., 2009; Ben-Tal & Nemirovski, 2002; Bertsimas et al., 2011; Gorissen et al., 2015; Bertsimas et al., 2018) framework where the decision maker optimizes the worst-case costs by considering the worst-case realization of the uncertainty from a user-specified ambiguity set, robust satisficing is free from the specification of such a set and minimizes the constraint violation (incurred by the deviation of the uncertainty from its estimated value) directly. The robustness of the RS model is adjusted by varing the target of the costs (serving as a targeted upper bound for the costs), where a higher target comes with stronger robustness. The successful applications of the robust satisficing framework includes those in contextual Bayesian optimization (Saday et al., 2024), energy system (Keyvandarian & Saif, 2023), hub location problem (Hu et al., 2024), and resource pooling (Cui et al., 2023).

To solve nominal MDPs (Puterman, 2014), one approach is to express them as a linear program in primal and dual forms. Our SRIRL and BROIL (Brown et al., 2020b) are both based on the dual formulation. In recent years, there has been a notable increase in the development of new reinforcement learning models based on the dual formulation, driven by its interpretability. For example, in the face of reward uncertainty, Delage & Mannor (2010) optimizes the value-at-risk (VaR) of expected return, while Brown et al. (2020b) maximizes a weighted average of the conditional value-at-risk (CVaR) and expected return. Confronted with transition kernel ambiguity, Lobo et al. (2020) also optimizes a weighted average of CVaR and expected return, Ruan et al. (2023) propose a robust satisficing MDPs that hedge against the transition kernel ambiguity by applying the robust satisficing framework to the dual formulation. Our SRIRL is different from these existing models in that it not only achieves a flexible balance between risk aversion and expected return maximization but also hedges against transition kernel ambiguity, both of which are important features in IRL.

Behavior cloning directly imitates the policy of the expert, ensuing a quadratic regret (Ross & Bagnell, 2010). DAgger has a sublinear regret that is achieved by making frequent queries to the expert for her action taken at different states. Its safe variants Hoque et al. (2021); Oh & Matsubara (2024); Zhang & Cho (2016) also rely on unlimited access to extra expert demonstration. Unlike these

approaches, inverse reinforcement learning learns the expert's behavior indirectly by inferring the reward function employed by the expert (Adams et al., 2022; Arora & Doshi, 2021; Metelli et al., 2021; 2023). However, many inverse reinforcement learning methods only infer a single estimate of the demonstrator's reward function (Brown et al., 2019; 2020c; Fu et al., 2017; Ziebart et al., 2008). To hedge against reward uncertainty, a number of studies on safe imitation learning using IRL consider an adversarial reward function (Chang et al., 2021; Hadfield-Menell et al., 2017; Ho & Ermon, 2016; Huang et al., 2018; Regan & Boutilier, 2012; Syed et al., 2008), which could output overly conservative policies in practice (Brown et al., 2020b). We focus on hedging against epistemic risk coming from reward uncertainty in IRL, yet there is another line of research focusing on the aleatoric risk stemming from transition probabilities (Lacotte et al., 2019; Majumdar et al., 2017; Santara et al., 2018). Though we consider the case with demonstration provided by a Boltzmann rational expert, some previous works consider cases where the expert(s) provide suboptimal demonstration in IRL (Brown et al., 2019; Choi et al., 2019; Poiani et al., 2024; Shiarlis et al., 2016; Zheng et al., 2014).

## B    REFORMULATING BROIL AS A LINEAR PROGRAM

By making use of the expression of CVaR as a maximization problem in Section 2.3, problem (1) can be equivalently reformulated as a problem as follows:

$$\max \quad \omega \cdot \mathbb{E}_{\mathbb{P}(\boldsymbol{w} \mid \mathcal{D})}[\tilde{\boldsymbol{w}}^\top (\boldsymbol{F}^\top \boldsymbol{u} - \boldsymbol{f}_{\mathrm{E}})] + (1 - \omega) \cdot \left\{ x - (1/\varepsilon) \cdot \mathbb{E}_{\mathbb{P}(\boldsymbol{w} \mid \mathcal{D})} \left[ \left[ x - \tilde{\boldsymbol{w}}^\top \left( \boldsymbol{F}^\top \boldsymbol{u} - \boldsymbol{f}_{\mathrm{E}} \right) \right]_+ \right] \right\}$$

$$\text{s.t.} \quad \mathbf{e}^\top \boldsymbol{u}_s - \boldsymbol{p}^\top \boldsymbol{Q}_s \boldsymbol{u} - d_s = 0 \qquad\qquad\qquad \forall s \in \mathcal{S}$$

$$x \in \mathbb{R}, \boldsymbol{u} \in \mathbb{R}_+^{S \cdot A}. \tag{8}$$

By the Bayesian IRL as described in Section F.1, we have $N$ weight samples $\{\boldsymbol{w}_i\}_{i \in [N]}$ that are sampled from the posterior distribution $\mathbb{P}(\boldsymbol{w} \mid \mathcal{D})$. By substituting the sample average based on these $N$ samples to the expectation in (8), we have:

$$\max \quad \omega \cdot (1/N) \sum_{i \in [N]} \left\{ \boldsymbol{w}_i^\top (\boldsymbol{F}^\top \boldsymbol{u} - \boldsymbol{f}_{\mathrm{E}}) \right\}$$

$$+ (1 - \omega) \cdot \left\{ x - (1/\varepsilon) \cdot (1/N) \sum_{i \in [N]} \left\{ \left[ x - \boldsymbol{w}_i^\top \left( \boldsymbol{F}^\top \boldsymbol{u} - \boldsymbol{f}_{\mathrm{E}} \right) \right]_+ \right\} \right\} \tag{9}$$

$$\text{s.t.} \quad \mathbf{e}^\top \boldsymbol{u}_s - \boldsymbol{p}^\top \boldsymbol{Q}_s \boldsymbol{u} - d_s = 0 \qquad\qquad\qquad \forall s \in \mathcal{S}$$

$$x \in \mathbb{R}, \boldsymbol{u} \in \mathbb{R}_+^{S \cdot A}.$$

The following proposition provides a equivalent reformulation of (9) as a linear program, as provided in the appendix in Brown et al. (2020b). We provide it here only for ease of reference.

**Proposition 9** *Problem* (9) *allows an equivalent reformulation as a linear program as follows:*

$$\max \quad \omega \cdot (1/N) \sum_{i \in [N]} \left\{ \boldsymbol{w}_i^\top (\boldsymbol{F}^\top \boldsymbol{u} - \boldsymbol{f}_{\mathrm{E}}) \right\} + (1 - \omega) \cdot x - \left( \frac{1 - \omega}{N \varepsilon} \right) \cdot \sum_{i \in [N]} y_i$$

$$\text{s.t.} \quad y_i \geq x - \boldsymbol{w}_i^\top \left( \boldsymbol{F}^\top \boldsymbol{u} - \boldsymbol{f}_{\mathrm{E}} \right) \qquad\qquad\qquad \forall i \in [N]$$

$$\mathbf{e}^\top \boldsymbol{u}_s - \boldsymbol{p}^\top \boldsymbol{Q}_s \boldsymbol{u} - d_s = 0 \qquad\qquad\qquad \forall s \in \mathcal{S}$$

$$x \in \mathbb{R}, \boldsymbol{u} \in \mathbb{R}_+^{S \cdot A}, \boldsymbol{y} \in \mathbb{R}_+^N.$$

*Proof of Proposition 9*    Considering the epigraph form, we can equivalently reformulate (9) as:

$$\max \quad \omega \cdot (1/N) \sum_{i \in [N]} \left\{ \boldsymbol{w}_i^\top (\boldsymbol{F}^\top \boldsymbol{u} - \boldsymbol{f}_{\mathrm{E}}) \right\} + (1 - \omega) \cdot x - \left( \frac{1 - \omega}{N \varepsilon} \right) \sum_{i \in [N]} y_i$$

$$\text{s.t.} \quad y_i \geq \left[ x - \boldsymbol{w}_i^\top \left( \boldsymbol{F}^\top \boldsymbol{u} - \boldsymbol{f}_{\mathrm{E}} \right) \right]_+ \qquad\qquad\qquad \forall i \in [N]$$

$$\mathbf{e}^\top \boldsymbol{u}_s - \boldsymbol{p}^\top \boldsymbol{Q}_s \boldsymbol{u} - d_s = 0 \qquad\qquad\qquad \forall s \in \mathcal{S}$$

$$x \in \mathbb{R}, \boldsymbol{u} \in \mathbb{R}_+^{S \cdot A}, \boldsymbol{y} \in \mathbb{R}^N.$$

Our conclusion then follows by exploring the definition of the operator $[\cdot]_+$ $\qquad\qquad\qquad\square$

## C  ADDITIONAL THEORETICAL RESULTS AND PROOFS

*Proof of Proposition 1*  Considering $\boldsymbol{p} = \hat{\boldsymbol{p}}$ in the first two sets of constraints in (2), one can observe that the feasible region of (2) is a subset of the optimal solution set of (1), where our argument (*i*) follows. Our second argument follows by the fact that the maximal value that the left-hand side of the third set of constraint in (2) can possibly achieve is $\mathrm{T_B}(\hat{\boldsymbol{p}})$, implying the feasibility of the problem when setting $\tau > \mathrm{T_B}(\hat{\boldsymbol{p}})$. $\qquad\qquad\qquad\square$

*Proof of Proposition 2*  When equipped with $\mathbb{P} = \hat{\mathbb{P}}$, our SRIRL (2) becomes:

$$
\begin{aligned}
\mathrm{T_{RS}}(\hat{\boldsymbol{p}}) = \quad &\min \quad \boldsymbol{\phi}^\top \boldsymbol{k} \\
&\text{s.t.} \quad \mathbf{e}^\top \boldsymbol{u}_s - \boldsymbol{p}^\top \boldsymbol{Q}_s \boldsymbol{u} - d_s \leq k_s \cdot \ell(\boldsymbol{p}, \hat{\boldsymbol{p}}) \qquad\qquad\qquad \forall \boldsymbol{p} \in \mathcal{P}, s \in \mathcal{S} \\
&\qquad\quad \mathbf{e}^\top \boldsymbol{u}_s - \boldsymbol{p}^\top \boldsymbol{Q}_s \boldsymbol{u} - d_s \geq -k_s \cdot \ell(\boldsymbol{p}, \hat{\boldsymbol{p}}) \qquad\qquad \forall \boldsymbol{p} \in \mathcal{P}, s \in \mathcal{S} \\
&\qquad\quad \omega \cdot (1/N) \sum_{i \in [N]} \left\{ \boldsymbol{w}_i^\top (\boldsymbol{F}^\top \boldsymbol{u} - \boldsymbol{f}_\mathrm{E}) \right\} \\
&\qquad\quad + (1-\omega) \cdot \left\{ x - (1/\varepsilon) \cdot (1/N) \sum_{i \in [N]} \left\{ \left[ x - \boldsymbol{w}_i^\top \left( \boldsymbol{F}^\top \boldsymbol{u} - \boldsymbol{f}_\mathrm{E} \right) \right]_+ \right\} \right\} \geq \tau \\
&\qquad\quad \boldsymbol{u} \in \mathbb{R}_+^{S \cdot A}, \boldsymbol{k} \in \mathbb{R}_+^S, x \in \mathbb{R}.
\end{aligned}
\tag{10}
$$

For the first set of constraints of (10), notice that the $s$-th one is equivalent to:

$$
\inf_{\boldsymbol{p} \in \mathcal{P}} k_s \cdot \|\boldsymbol{p} - \hat{\boldsymbol{p}}\| + \boldsymbol{p}^\top \boldsymbol{Q}_s \boldsymbol{u} \geq \mathbf{e}^\top \boldsymbol{u}_s - d_s,
$$

The dual of the left-hand minimization problem is:

$$
\max_{\overline{\boldsymbol{\beta}} \in \mathbb{R}_+^{S \cdot A \cdot S}, \overline{\boldsymbol{\alpha}} \in \mathbb{R}^{S \cdot A}} \inf_{\boldsymbol{p} \in \mathbb{R}_+^{S \cdot A \cdot S}} k_s \cdot \|\boldsymbol{p} - \hat{\boldsymbol{p}}\| + \boldsymbol{p}^\top \boldsymbol{Q}_s \boldsymbol{u} + \overline{\boldsymbol{\alpha}}^\top (\boldsymbol{B}\boldsymbol{p} - \mathbf{e}) - \overline{\boldsymbol{\beta}}^\top \boldsymbol{p}
$$

$$
\Leftrightarrow \quad \begin{cases} \max \quad -\overline{\boldsymbol{\alpha}}^\top \mathbf{e} - \hat{\boldsymbol{p}}^\top (\overline{\boldsymbol{\beta}} - \boldsymbol{Q}_s \boldsymbol{u} - \boldsymbol{B}^\top \overline{\boldsymbol{\alpha}}) \\ \text{s.t.} \quad \|\overline{\boldsymbol{\beta}} - \boldsymbol{Q}_s \boldsymbol{u} - \boldsymbol{B}^\top \overline{\boldsymbol{\alpha}}\|_* \leq k_s \\ \qquad \overline{\boldsymbol{\alpha}} \in \mathbb{R}^{S \cdot A}, \overline{\boldsymbol{\beta}} \in \mathbb{R}_+^{S \cdot A \cdot S}. \end{cases}
$$

where $\boldsymbol{B} = \mathrm{diag}(\mathbf{e}^\top, \cdots, \mathbf{e}^\top) \in \mathbb{R}^{S \cdot A \times S \cdot A \cdot S}$ and $\mathbf{e} \in \mathbb{R}^S$ so that

$$
\boldsymbol{B}\boldsymbol{p} = \mathbf{e} \quad \Leftrightarrow \quad \mathbf{e}^\top \boldsymbol{p}_{s,a} = 1 \ \forall s \in \mathcal{S}, a \in \mathcal{A}.
$$

Here strong duality holds because Slater's condition is satisfied by the feasible solution $\boldsymbol{p} = (1/S) \cdot \mathbf{e}$. Therefore, the first set of constraints of (10) is equivalent to:

$$
\forall s \in \mathcal{S} : \begin{cases} \exists \, \overline{\boldsymbol{\alpha}}_s \in \mathbb{R}^{S \cdot A}, \overline{\boldsymbol{\beta}}_s \in \mathbb{R}_+^{S \cdot A \cdot S} : \\ \mathbf{e}^\top \boldsymbol{u}_s - d_s \leq -\overline{\boldsymbol{\alpha}}_s^\top \mathbf{e} - \hat{\boldsymbol{p}}^\top (\overline{\boldsymbol{\beta}}_s - \boldsymbol{Q}_s \boldsymbol{u} - \boldsymbol{B}^\top \overline{\boldsymbol{\alpha}}_s) \\ \|\overline{\boldsymbol{\beta}}_s - \boldsymbol{Q}_s \boldsymbol{u} - \boldsymbol{B}^\top \overline{\boldsymbol{\alpha}}_s\|_* \leq k_s. \end{cases}
\tag{11}
$$

Applying similar techniques to the second set of constraints leads to their equivalent set of constraints as follows:

$$
\forall s \in \mathcal{S} : \begin{cases} \exists \, \underline{\boldsymbol{\alpha}}_s \in \mathbb{R}^{S \cdot A}, \underline{\boldsymbol{\beta}}_s \in \mathbb{R}_+^{S \cdot A \cdot S} : \\ -\mathbf{e}^\top \boldsymbol{u}_s + d_s \leq -\underline{\boldsymbol{\alpha}}_s^\top \mathbf{e} - \hat{\boldsymbol{p}}^\top (\underline{\boldsymbol{\beta}}_s + \boldsymbol{Q}_s \boldsymbol{u} - \boldsymbol{B}^\top \underline{\boldsymbol{\alpha}}_s) \\ \|\underline{\boldsymbol{\beta}}_s + \boldsymbol{Q}_s \boldsymbol{u} - \boldsymbol{B}^\top \underline{\boldsymbol{\alpha}}_s\|_* \leq k_s. \end{cases}
\tag{12}
$$

By introducing auxiliary decision variables $\boldsymbol{y} \in \mathbb{R}^N$, the third set of constraint of (10) is equivalent to:

$$
\begin{cases} \exists \, \boldsymbol{y} \in \mathbb{R}_+^N : \\ \dfrac{\omega}{N} \sum_{i \in [N]} \boldsymbol{w}_i^\top (\boldsymbol{F}^\top \boldsymbol{u} - \boldsymbol{f}_\mathrm{E}) + (1-\omega)x - \tau \geq \dfrac{1-\omega}{N\varepsilon} \cdot \sum_{i \in [N]} y_i \\ y_i \geq x - \boldsymbol{w}_i^\top (\boldsymbol{F}^\top \boldsymbol{u} - \boldsymbol{f}_\mathrm{E}) \qquad\qquad\qquad\qquad \forall i \in [N]. \end{cases}
\tag{13}
$$

Plugging (11), (12), and (13) into (10) concludes our proof. $\qquad\square$

**Lemma 1** *Equipped with $w \in \mathbb{R}_{++}$, strong duality holds for the problem as follows:*

$$\max \quad \overline{\lambda}(\mathbf{e}^\top \boldsymbol{u} - d) + \underline{\lambda}(-\mathbf{e}^\top \boldsymbol{u} + d) - \overline{\boldsymbol{\theta}}^\top \boldsymbol{Q}\boldsymbol{u} + \underline{\boldsymbol{\theta}}^\top \boldsymbol{Q}\boldsymbol{u}$$

$$\begin{aligned}
\text{s.t.} \quad & \overline{\xi} + \underline{\xi} \le w \\
& \boldsymbol{B}\overline{\boldsymbol{\theta}} = \overline{\lambda} \cdot \mathbf{e} \\
& \boldsymbol{B}\underline{\boldsymbol{\theta}} = \underline{\lambda} \cdot \mathbf{e} \\
& \|\overline{\lambda} \cdot \hat{\boldsymbol{p}} - \overline{\boldsymbol{\theta}}\| \le \overline{\xi} \\
& \|\underline{\lambda} \cdot \hat{\boldsymbol{p}} - \underline{\boldsymbol{\theta}}\| \le \underline{\xi} \\
& \overline{\xi}, \underline{\xi}, \overline{\lambda}, \underline{\lambda} \in \mathbb{R}_+, \overline{\boldsymbol{\theta}}, \underline{\boldsymbol{\theta}} \in \mathbb{R}_+^{S \cdot A \cdot S},
\end{aligned} \tag{14}$$

*where $\boldsymbol{u} \in \mathbb{R}^{S \cdot A}$, $d \in \mathbb{R}$, $\boldsymbol{Q} \in \mathbb{R}^{S \cdot A \cdot S \times S \cdot A}$ are arbitrary constants.*

*Proof of Lemma 1* A strictly feasible solution to (14) would be sufficient for the proof. Considering $\overline{\xi} = \underline{\xi} = w/3$, then it suffices to construct solution for the system

$$\begin{cases}
\|\boldsymbol{\theta} - \lambda \cdot \hat{\boldsymbol{p}}\| \le w/3 \\
\lambda \cdot \mathbf{e} = \boldsymbol{B}\boldsymbol{\theta} \\
\lambda \in \mathbb{R}_+, \boldsymbol{\theta} \in \mathbb{R}_+^{S \cdot A \cdot S}
\end{cases} \tag{15}$$

that is strictly feasible (following from which $(\overline{\xi}, \underline{\xi}, \overline{\lambda}, \underline{\lambda}, \overline{\boldsymbol{\theta}}, \underline{\boldsymbol{\theta}}) = (w/3, w/3, \lambda, \lambda, \boldsymbol{\theta}, \boldsymbol{\theta})$ is strictly feasible for (14)). To achieve this, notice that

$$\lambda \cdot \mathbf{e} = \boldsymbol{B}\boldsymbol{\theta} \iff \mathbf{e}^\top \boldsymbol{\theta}_{s,a} = \lambda \ \forall s \in \mathcal{S}, a \in \mathcal{A},$$

implying that $(\lambda, \boldsymbol{\theta}) : \boldsymbol{\theta} = \lambda \cdot \hat{\boldsymbol{p}}$ is feasible to (15) for arbitrary $\lambda > 0$. Such a solution is already strictly feasible to (15), thus by Slater's condition, strong duality holds for (14). Otherwise, we will construct a feasible solution for (15) by $(\lambda, \boldsymbol{\theta}) : \boldsymbol{\theta} = \lambda \cdot \hat{\boldsymbol{p}}$ for the case with $\hat{\boldsymbol{p}} \ge \mathbf{0}$ with at least one zero entry. Without loss of generality, suppose $\hat{p}_{\overline{s}, \overline{a}, \overline{s}'} = 0$. Since $\hat{\boldsymbol{p}} \in (\Delta^S)^{S \cdot A}$, there must be some strictly positive entry $\theta_{\overline{s}, \overline{a}, \overline{s}''} > 0$ ($\overline{s}' \ne \overline{s}''$) of $\boldsymbol{\theta} = \lambda \cdot \hat{\boldsymbol{p}}$. Let

$$\varepsilon = \min\{\theta_{\overline{s}, \overline{a}, \overline{s}''}/2, w/(6 \cdot \|\mathbf{e}_{\overline{s}, \overline{a}, \overline{s}'} - \mathbf{e}_{\overline{s}, \overline{a}, \overline{s}''}\|)\},$$

where $\mathbf{e}_{\overline{s}, \overline{a}, \overline{s}'}$ and $\mathbf{e}_{\overline{s}, \overline{a}, \overline{s}'}$ are standard bases of $\mathbb{R}^{S \cdot A \cdot S}$. It then follows that $(\lambda, \boldsymbol{\theta}')$ with $\boldsymbol{\theta}' = \boldsymbol{\theta} + \varepsilon \cdot (\mathbf{e}_{\overline{s}, \overline{a}, \overline{s}'} - \mathbf{e}_{\overline{s}, \overline{a}, \overline{s}''})$ is also feasible in (15), which is strictly feasible for the first inequality constraint in (15), with $\theta'_{\overline{s}, \overline{a}, \overline{s}'}, \theta'_{\overline{s}, \overline{a}, \overline{s}''} > 0$. Going through a similar procedure, one can eventually construct a feasible solution with all entries being strictly positive, constituting a strictly feasible solution of (15). $\qquad\square$

*Proof of Proposition 3* By considering the constraints corresponding to $\boldsymbol{p} = \hat{\boldsymbol{p}}$ in the first two set of constraints in (2), we have

$$\mathbf{e}^\top \boldsymbol{u}_s - \hat{\boldsymbol{p}}^\top \boldsymbol{Q}_s \boldsymbol{u} - d_s = 0 \ \forall s \in \mathcal{S}.$$

Therefore, by Proposition 2, our SRIRL (2) is equivalent to:

$$\begin{aligned}
\mathrm{T}_{\mathrm{RS}}(\hat{\boldsymbol{p}}) = \quad \min \quad & \boldsymbol{\phi}^\top \boldsymbol{k} \\
\text{s.t.} \quad & \mathbf{e}^\top \boldsymbol{u}_s - d_s \le -\hat{\boldsymbol{p}}^\top \overline{\boldsymbol{\beta}}_s + \hat{\boldsymbol{p}}^\top \boldsymbol{Q}_s \boldsymbol{u} && \forall s \in \mathcal{S} \\
& -\mathbf{e}^\top \boldsymbol{u}_s + d_s \le -\hat{\boldsymbol{p}}^\top \underline{\boldsymbol{\beta}}_s - \hat{\boldsymbol{p}}^\top \boldsymbol{Q}_s \boldsymbol{u} && \forall s \in \mathcal{S} \\
& \mathbf{e}^\top \boldsymbol{u}_s - \hat{\boldsymbol{p}}^\top \boldsymbol{Q}_s \boldsymbol{u} - d_s = 0 && \forall s \in \mathcal{S} \\
& \|\overline{\boldsymbol{\beta}}_s - \boldsymbol{Q}_s \boldsymbol{u} - \boldsymbol{B}^\top \overline{\boldsymbol{\alpha}}_s\|_* \le k_s && \forall s \in \mathcal{S} \\
& \|\underline{\boldsymbol{\beta}}_s + \boldsymbol{Q}_s \boldsymbol{u} - \boldsymbol{B}^\top \underline{\boldsymbol{\alpha}}_s\|_* \le k_s && \forall s \in \mathcal{S} \\
& \frac{\omega}{N} \sum_{i \in [N]} \boldsymbol{w}_i^\top (\boldsymbol{F}^\top \boldsymbol{u} - \boldsymbol{f}_{\mathrm{E}}) + (1 - \omega)x - \tau \ge \frac{1 - \omega}{N\varepsilon} \cdot \sum_{i \in [N]} y_i \\
& y_i \ge x - \boldsymbol{w}_i^\top (\boldsymbol{F}^\top \boldsymbol{u} - \boldsymbol{f}_{\mathrm{E}}) && \forall i \in [N] \\
& \boldsymbol{u} \in \mathbb{R}_+^{S \cdot A}, \boldsymbol{k} \in \mathbb{R}_+^S, x \in \mathbb{R}, \boldsymbol{y} \in \mathbb{R}_+^N \\
& \overline{\boldsymbol{\alpha}}_s \in \mathbb{R}^{S \cdot A}, \underline{\boldsymbol{\alpha}}_s \in \mathbb{R}^{S \cdot A}, \overline{\boldsymbol{\beta}}_s \in \mathbb{R}_+^{S \cdot A \cdot S}, \underline{\boldsymbol{\beta}}_s \in \mathbb{R}_+^{S \cdot A \cdot S} && \forall s \in \mathcal{S}.
\end{aligned}$$

This problem can be re-expressed as an equivalent min-min problem as follows:

$$
\begin{aligned}
&\mathrm{T_{RS}}(\hat{\boldsymbol{p}}) \\
&= \min_{\boldsymbol{u},x,\boldsymbol{y}} \ \min_{\boldsymbol{k},\overline{\boldsymbol{\alpha}},\underline{\boldsymbol{\alpha}},\overline{\boldsymbol{\beta}},\underline{\boldsymbol{\beta}}} \quad \boldsymbol{\phi}^{\top}\boldsymbol{k} \\
&\qquad\ \text{s.t.} \quad \boldsymbol{e}^{\top}\boldsymbol{u}_s - d_s \leq -\hat{\boldsymbol{p}}^{\top}\overline{\boldsymbol{\beta}}_s + \hat{\boldsymbol{p}}^{\top}\boldsymbol{Q}_s\boldsymbol{u} && \forall s \in \mathcal{S} \\
&\qquad\qquad\quad -\boldsymbol{e}^{\top}\boldsymbol{u}_s + d_s \leq -\hat{\boldsymbol{p}}^{\top}\underline{\boldsymbol{\beta}}_s - \hat{\boldsymbol{p}}^{\top}\boldsymbol{Q}_s\boldsymbol{u} && \forall s \in \mathcal{S} \\
&\qquad\qquad\quad \boldsymbol{e}^{\top}\boldsymbol{u}_s - \hat{\boldsymbol{p}}^{\top}\boldsymbol{Q}_s\boldsymbol{u} - d_s = 0 && \forall s \in \mathcal{S}. \\
&\qquad\qquad\quad \|\overline{\boldsymbol{\beta}}_s - \boldsymbol{Q}_s\boldsymbol{u} - \boldsymbol{B}^{\top}\overline{\boldsymbol{\alpha}}_s\|_* \leq k_s && \forall s \in \mathcal{S} \\
&\qquad\qquad\quad \|\underline{\boldsymbol{\beta}}_s + \boldsymbol{Q}_s\boldsymbol{u} - \boldsymbol{B}^{\top}\underline{\boldsymbol{\alpha}}_s\|_* \leq k_s && \forall s \in \mathcal{S} \\
&\qquad\qquad\quad \frac{\omega}{N}\sum_{i\in[N]}\boldsymbol{w}_i^{\top}(\boldsymbol{F}^{\top}\boldsymbol{u}-\boldsymbol{f}_{\mathrm{E}}) + (1-\omega)x - \tau \geq \frac{1-\omega}{N\varepsilon}\cdot\sum_{i\in[N]}y_i \\
&\qquad\qquad\quad y_i \geq x - \boldsymbol{w}_i^{\top}(\boldsymbol{F}^{\top}\boldsymbol{u}-\boldsymbol{f}_{\mathrm{E}}) && \forall i \in [N] \\
&\qquad\qquad\quad \boldsymbol{u} \in \mathbb{R}_+^{S\cdot A},\ \boldsymbol{k} \in \mathbb{R}_+^{S},\ x \in \mathbb{R},\ \boldsymbol{y} \in \mathbb{R}_+^{N} \\
&\qquad\qquad\quad \overline{\boldsymbol{\alpha}}_s \in \mathbb{R}^{S\cdot A},\ \underline{\boldsymbol{\alpha}}_s \in \mathbb{R}^{S\cdot A},\ \overline{\boldsymbol{\beta}}_s \in \mathbb{R}_+^{S\cdot A\cdot S},\ \underline{\boldsymbol{\beta}}_s \in \mathbb{R}_+^{S\cdot A\cdot S} && \forall s \in \mathcal{S}.
\end{aligned}
\tag{16}
$$

Here the equivalence follows by noting that, for any feasible solution $(\boldsymbol{u},x,\boldsymbol{y})$ for the outer minimization problem, that is, for any $(\boldsymbol{u},x,\boldsymbol{y}) \in \mathbb{R}_+^{S\cdot A}\times\mathbb{R}\times\mathbb{R}_+^{N}$ that satisfies

$$
\begin{cases}
\dfrac{\omega}{N}\sum_{i\in[N]}\boldsymbol{w}_i^{\top}(\boldsymbol{F}^{\top}\boldsymbol{u}-\boldsymbol{f}_{\mathrm{E}}) + (1-\omega)x - \tau \geq \dfrac{1-\omega}{N\varepsilon}\cdot\sum_{i\in[N]}y_i \\
y_i \geq x - \boldsymbol{w}_i^{\top}(\boldsymbol{F}^{\top}\boldsymbol{u}-\boldsymbol{f}_{\mathrm{E}}) \qquad\qquad\qquad \forall i \in [N],
\end{cases}
\tag{17}
$$

the inner minimization problem is feasible with a feasible solution

$$
(k_s,\overline{\boldsymbol{\alpha}}_s,\underline{\boldsymbol{\alpha}}_s,\overline{\boldsymbol{\beta}}_s,\underline{\boldsymbol{\beta}}_s) = (\|\boldsymbol{Q}_s\boldsymbol{u}\|_*,\boldsymbol{0},\boldsymbol{0},\boldsymbol{0},\boldsymbol{0}) \quad \forall s \in \mathcal{S}.
$$

Problem (16) is further equivalent to:

$$
\begin{aligned}
&\mathrm{T_{RS}}(\hat{\boldsymbol{p}}) \\
&= \min_{\boldsymbol{u},x,\boldsymbol{y}} \ \min_{\boldsymbol{k},\overline{\boldsymbol{\alpha}},\underline{\boldsymbol{\alpha}},\overline{\boldsymbol{\beta}},\underline{\boldsymbol{\beta}},\overline{\boldsymbol{y}},\underline{\boldsymbol{y}}} \quad \boldsymbol{\phi}^{\top}\boldsymbol{k} \\
&\qquad\ \text{s.t.} \quad \boldsymbol{e}^{\top}\boldsymbol{u}_s - d_s \leq -\overline{\boldsymbol{\alpha}}_s^{\top}\boldsymbol{e} - \hat{\boldsymbol{p}}^{\top}\overline{\boldsymbol{y}}_s && \forall s \in \mathcal{S} \\
&\qquad\qquad\quad -\boldsymbol{e}^{\top}\boldsymbol{u}_s + d_s \leq -\underline{\boldsymbol{\alpha}}_s^{\top}\boldsymbol{e} - \hat{\boldsymbol{p}}^{\top}\underline{\boldsymbol{y}}_s && \forall s \in \mathcal{S} \\
&\qquad\qquad\quad \boldsymbol{e}^{\top}\boldsymbol{u}_s - \hat{\boldsymbol{p}}^{\top}\boldsymbol{Q}_s\boldsymbol{u} - d_s = 0 && \forall s \in \mathcal{S} \\
&\qquad\qquad\quad \overline{\boldsymbol{y}}_s = \overline{\boldsymbol{\beta}}_s - \boldsymbol{Q}_s\boldsymbol{u} - \boldsymbol{B}^{\top}\overline{\boldsymbol{\alpha}}_s && \forall s \in \mathcal{S} \\
&\qquad\qquad\quad \underline{\boldsymbol{y}}_s = \underline{\boldsymbol{\beta}}_s + \boldsymbol{Q}_s\boldsymbol{u} - \boldsymbol{B}^{\top}\underline{\boldsymbol{\alpha}}_s && \forall s \in \mathcal{S} \\
&\qquad\qquad\quad \|\overline{\boldsymbol{y}}_s\|_* \leq k_s && \forall s \in \mathcal{S} \\
&\qquad\qquad\quad \|\underline{\boldsymbol{y}}_s\|_* \leq k_s && \forall s \in \mathcal{S} \\
&\qquad\qquad\quad \frac{\omega}{N}\sum_{i\in[N]}\boldsymbol{w}_i^{\top}(\boldsymbol{F}^{\top}\boldsymbol{u}-\boldsymbol{f}_{\mathrm{E}}) + (1-\omega)x - \tau \geq \frac{1-\omega}{N\varepsilon}\cdot\sum_{i\in[N]}y_i \\
&\qquad\qquad\quad y_i \geq x - \boldsymbol{w}_i^{\top}(\boldsymbol{F}^{\top}\boldsymbol{u}-\boldsymbol{f}_{\mathrm{E}}) && \forall i \in [N] \\
&\qquad\qquad\quad \boldsymbol{u} \in \mathbb{R}_+^{S\cdot A},\ \boldsymbol{k} \in \mathbb{R}_+^{S},\ x \in \mathbb{R},\ \boldsymbol{y} \in \mathbb{R}_+^{N} \\
&\qquad\qquad\quad \overline{\boldsymbol{\alpha}}_s,\ \underline{\boldsymbol{\alpha}}_s \in \mathbb{R}^{S\cdot A},\ \overline{\boldsymbol{\beta}}_s,\ \underline{\boldsymbol{\beta}}_s \in \mathbb{R}_+^{S\cdot A\cdot S},\ \overline{\boldsymbol{y}}_s,\ \underline{\boldsymbol{y}}_s \in \mathbb{R}^{S\cdot A\cdot S} && \forall s \in \mathcal{S},
\end{aligned}
$$

where the dual problem of the inner minimization problem is:

$$\max_{\substack{\overline{\lambda},\underline{\lambda}\in\mathbb{R}^S_+,\\ \overline{\theta},\underline{\theta}\in\mathbb{R}^{S\cdot S\cdot A\cdot S}\\ \overline{\xi},\underline{\xi}\in\mathbb{R}^S_+}} \min_{\substack{k\in\mathbb{R}^S_+,\overline{\alpha},\underline{\alpha}\in\mathbb{R}^{S\cdot S\cdot A},\\ \overline{\beta},\underline{\beta}\in\mathbb{R}^{S\cdot S\cdot A\cdot S}_+,\\ \overline{y},\underline{y}\in\mathbb{R}^{S\cdot S\cdot A\cdot S}}} \phi^\top k + \sum_{s\in\mathcal{S}}\big\{\overline{\lambda}_s(\mathbf{e}^\top u_s - d_s + \overline{\alpha}_s^\top\mathbf{e} + \hat{p}^\top\overline{y}_s)$$

$$+\underline{\lambda}_s(-\mathbf{e}^\top u_s + d_s + \underline{\alpha}_s^\top\mathbf{e} + \hat{p}^\top\underline{y}_s)$$

$$+\overline{\theta}_s^\top(\overline{\beta}_s - Q_s u - B^\top\overline{\alpha}_s - \overline{y}_s)$$

$$+\underline{\theta}_s^\top(\underline{\beta}_s + Q_s u - B^\top\underline{\alpha}_s - \underline{y}_s)$$

$$+\overline{\xi}_s(\|\overline{y}_s\|_* - k_s) + \underline{\xi}_s(\|\underline{y}_s\|_* - k_s)\big\},$$

which is equivalent to

$$\max \quad \sum_{s\in\mathcal{S}}\Big\{\overline{\lambda}_s(\mathbf{e}^\top u_s - d_s) + \underline{\lambda}_s(-\mathbf{e}^\top u_s + d_s) - \overline{\theta}_s^\top Q_s u + \underline{\theta}_s^\top Q_s u\Big\}$$

$$\text{s.t.} \quad \overline{\xi} + \underline{\xi} \le \phi$$

$$B\overline{\theta}_s = \overline{\lambda}_s\cdot\mathbf{e} \qquad\qquad\qquad \forall s\in\mathcal{S}$$

$$B\underline{\theta}_s = \underline{\lambda}_s\cdot\mathbf{e} \qquad\qquad\qquad \forall s\in\mathcal{S}$$

$$\|\overline{\lambda}_s\cdot\hat{p} - \overline{\theta}_s\| \le \overline{\xi}_s \qquad\qquad \forall s\in\mathcal{S}$$

$$\|\underline{\lambda}_s\cdot\hat{p} - \underline{\theta}_s\| \le \underline{\xi}_s \qquad\qquad \forall s\in\mathcal{S}$$

$$\overline{\xi},\ \underline{\xi},\ \overline{\lambda},\ \underline{\lambda}\in\mathbb{R}^S_+,\ \overline{\theta},\ \underline{\theta}\in\mathbb{R}^{S\cdot S\cdot A\cdot S}_+$$

since

$$\min_{k\in\mathbb{R}^S_+}(\phi - \overline{\xi} - \underline{\xi})^\top k = \begin{cases} 0 & \phi - \overline{\xi} - \underline{\xi} \ge \mathbf{0} \\ -\infty & \text{otherwise,} \end{cases}$$

$$\min_{\overline{\alpha}\in\mathbb{R}^{S\cdot S\cdot A}}\sum_{s\in\mathcal{S}}\overline{\alpha}_s^\top(\overline{\lambda}_s\cdot\mathbf{e} - B\overline{\theta}_s) = \begin{cases} 0 & \overline{\lambda}_s\cdot\mathbf{e} - B\overline{\theta}_s = \mathbf{0}\ \forall s\in\mathcal{S} \\ -\infty & \text{otherwise,} \end{cases}$$

$$\min_{\underline{\alpha}\in\mathbb{R}^{S\cdot S\cdot A}}\sum_{s\in\mathcal{S}}\underline{\alpha}_s^\top(\underline{\lambda}_s\cdot\mathbf{e} - B\underline{\theta}_s) = \begin{cases} 0 & \underline{\lambda}_s\cdot\mathbf{e} - B\underline{\theta}_s = \mathbf{0}\ \forall s\in\mathcal{S} \\ -\infty & \text{otherwise,} \end{cases}$$

$$\min_{\overline{\beta}\in\mathbb{R}^{S\cdot S\cdot A\cdot S}_+}\sum_{s\in\mathcal{S}}\overline{\theta}_s^\top\overline{\beta}_s = \begin{cases} 0 & \overline{\theta}_s \ge \mathbf{0}\ \forall s\in\mathcal{S} \\ -\infty & \text{otherwise,} \end{cases}$$

$$\min_{\underline{\beta}\in\mathbb{R}^{S\cdot S\cdot A\cdot S}_+}\sum_{s\in\mathcal{S}}\underline{\theta}_s^\top\underline{\beta}_s = \begin{cases} 0 & \underline{\theta}_s \ge \mathbf{0}\ \forall s\in\mathcal{S} \\ -\infty & \text{otherwise,} \end{cases}$$

$$\min_{\overline{y}\in\mathbb{R}^{S\cdot S\cdot A\cdot S}}\sum_{s\in\mathcal{S}}\big\{\overline{\xi}_s\cdot\|\overline{y}_s\|_* + \overline{y}_s^\top(\overline{\lambda}_s\cdot\hat{p} - \overline{\theta}_s)\big\} = \begin{cases} 0 & \|\overline{\theta}_s - \overline{\lambda}_s\cdot\hat{p}\| \le \overline{\xi}_s\ \forall s\in\mathcal{S} \\ -\infty & \text{otherwise,} \end{cases}$$

and

$$\min_{\underline{y}\in\mathbb{R}^{S\cdot S\cdot A\cdot S}}\sum_{s\in\mathcal{S}}\big\{\underline{\xi}_s\cdot\|\underline{y}_s\|_* + \underline{y}_s^\top(\underline{\lambda}_s\cdot\hat{p} - \underline{\theta}_s)\big\} = \begin{cases} 0 & \|\underline{\theta}_s - \underline{\lambda}_s\cdot\hat{p}\| \le \underline{\xi}_s\ \forall s\in\mathcal{S} \\ -\infty & \text{otherwise.} \end{cases}$$

Here strong duality holds by Lemma 1. Therefore, problem (16) is equivalent to

$$
\min_{\boldsymbol{u},x,\boldsymbol{y}} \max_{\overline{\boldsymbol{\xi}},\underline{\boldsymbol{\xi}},\overline{\boldsymbol{\lambda}},\underline{\boldsymbol{\lambda}},\overline{\boldsymbol{\theta}},\underline{\boldsymbol{\theta}}} \sum_{s\in\mathcal{S}}\left\{\overline{\lambda}_s(\mathbf{e}^\top\boldsymbol{u}_s - d_s) + \underline{\lambda}_s(-\mathbf{e}^\top\boldsymbol{u}_s + d_s) - \overline{\boldsymbol{\theta}}_s^\top \boldsymbol{Q}_s\boldsymbol{u} + \underline{\boldsymbol{\theta}}_s^\top \boldsymbol{Q}_s\boldsymbol{u}\right\}
$$

$$
\begin{aligned}
\text{s.t.}\quad & \overline{\boldsymbol{\xi}} + \underline{\boldsymbol{\xi}} \leq \boldsymbol{\phi} \\
& \boldsymbol{B}\overline{\boldsymbol{\theta}}_s = \overline{\lambda}_s \cdot \mathbf{e} && \forall s\in\mathcal{S} \\
& \boldsymbol{B}\underline{\boldsymbol{\theta}}_s = \underline{\lambda}_s \cdot \mathbf{e} && \forall s\in\mathcal{S} \\
& \|\overline{\lambda}_s \cdot \hat{\boldsymbol{p}} - \overline{\boldsymbol{\theta}}_s\| \leq \overline{\xi}_s && \forall s\in\mathcal{S} \\
& \|\underline{\lambda}_s \cdot \hat{\boldsymbol{p}} - \underline{\boldsymbol{\theta}}_s\| \leq \underline{\xi}_s && \forall s\in\mathcal{S} \\
& \mathbf{e}^\top\boldsymbol{u}_s - \boldsymbol{p}^\top\boldsymbol{Q}_s\boldsymbol{u} - d_s = 0 && \forall s\in\mathcal{S} \\
& \frac{\omega}{N}\sum_{i\in[N]}\boldsymbol{w}_i^\top(\boldsymbol{F}^\top\boldsymbol{u} - \boldsymbol{f}_{\mathrm{E}}) + (1-\omega)x - \tau \geq \frac{1-\omega}{N\varepsilon}\cdot\sum_{i\in[N]}y_i \\
& y_i \geq x - \boldsymbol{w}_i^\top(\boldsymbol{F}^\top\boldsymbol{u} - \boldsymbol{f}_{\mathrm{E}}) && \forall i\in[N] \\
& \boldsymbol{u}\in\mathbb{R}_+^{S\cdot A},\ x\in\mathbb{R},\ \boldsymbol{y}\in\mathbb{R}_+^N,\ \overline{\boldsymbol{\xi}},\ \underline{\boldsymbol{\xi}},\ \overline{\boldsymbol{\lambda}},\ \underline{\boldsymbol{\lambda}}\in\mathbb{R}_+^S,\ \overline{\boldsymbol{\theta}},\ \underline{\boldsymbol{\theta}}\in\mathbb{R}_+^{S\cdot S\cdot A\cdot S}.
\end{aligned}
$$

We can indeed remove the sixth set of constraints and obtain an equivalent problem to this minimax problem. This is because for any feasible solution $(\boldsymbol{u},x,\boldsymbol{y})$ for the outer minimization problem, if $\mathbf{e}^\top\boldsymbol{u}_s - \boldsymbol{p}^\top\boldsymbol{Q}_s\boldsymbol{u} - d_s > 0$ (resp., $\mathbf{e}^\top\boldsymbol{u}_s - \boldsymbol{p}^\top\boldsymbol{Q}_s\boldsymbol{u} - d_s < 0$) for some $s\in\mathcal{S}$, then by $\overline{\boldsymbol{\theta}}_s = \overline{\lambda}_s \cdot \hat{\boldsymbol{p}}$ (resp., $\underline{\boldsymbol{\theta}}_s = \underline{\lambda}_s \cdot \hat{\boldsymbol{p}}$), the objective value can be arbitrarily large by considering $\overline{\lambda}_s \to \infty$ (resp., $\underline{\lambda}_s \to \infty$).

We then need to show that

$$
\min_{\boldsymbol{u},x,\boldsymbol{y}} \max_{\overline{\boldsymbol{\xi}},\underline{\boldsymbol{\xi}},\overline{\boldsymbol{\lambda}},\underline{\boldsymbol{\lambda}},\overline{\boldsymbol{\theta}},\underline{\boldsymbol{\theta}}} \sum_{s\in\mathcal{S}}\left\{\overline{\lambda}_s(\mathbf{e}^\top\boldsymbol{u}_s - d_s) + \underline{\lambda}_s(-\mathbf{e}^\top\boldsymbol{u}_s + d_s) - \overline{\boldsymbol{\theta}}_s^\top \boldsymbol{Q}_s\boldsymbol{u} + \underline{\boldsymbol{\theta}}_s^\top \boldsymbol{Q}_s\boldsymbol{u}\right\}
$$

$$
\begin{aligned}
\text{s.t.}\quad & \overline{\boldsymbol{\xi}} + \underline{\boldsymbol{\xi}} \leq \boldsymbol{\phi} \\
& \boldsymbol{B}\overline{\boldsymbol{\theta}}_s = \overline{\lambda}_s \cdot \mathbf{e} && \forall s\in\mathcal{S} \\
& \boldsymbol{B}\underline{\boldsymbol{\theta}}_s = \underline{\lambda}_s \cdot \mathbf{e} && \forall s\in\mathcal{S} \\
& \|\overline{\lambda}_s \cdot \hat{\boldsymbol{p}} - \overline{\boldsymbol{\theta}}_s\| \leq \overline{\xi}_s && \forall s\in\mathcal{S} \\
& \|\underline{\lambda}_s \cdot \hat{\boldsymbol{p}} - \underline{\boldsymbol{\theta}}_s\| \leq \underline{\xi}_s && \forall s\in\mathcal{S} \\
& \frac{\omega}{N}\sum_{i\in[N]}\boldsymbol{w}_i^\top(\boldsymbol{F}^\top\boldsymbol{u} - \boldsymbol{f}_{\mathrm{E}}) + (1-\omega)x - \tau \geq \frac{1-\omega}{N\varepsilon}\cdot\sum_{i\in[N]}y_i \\
& y_i \geq x - \boldsymbol{w}_i^\top(\boldsymbol{F}^\top\boldsymbol{u} - \boldsymbol{f}_{\mathrm{E}}) && \forall i\in[N] \\
& \boldsymbol{u}\in\mathbb{R}_+^{S\cdot A},\ x\in\mathbb{R},\ \boldsymbol{y}\in\mathbb{R}_+^N,\ \overline{\boldsymbol{\xi}},\ \underline{\boldsymbol{\xi}},\ \overline{\boldsymbol{\lambda}},\ \underline{\boldsymbol{\lambda}}\in\mathbb{R}_+^S,\ \overline{\boldsymbol{\theta}},\ \underline{\boldsymbol{\theta}}\in\mathbb{R}_+^{S\cdot S\cdot A\cdot S}
\end{aligned}
\tag{18}
$$

is equivalent to

$$
\min_{\boldsymbol{u},x,\boldsymbol{y}} \max_{\boldsymbol{\xi},\overline{\boldsymbol{\lambda}},\underline{\boldsymbol{\lambda}},\overline{\boldsymbol{\theta}},\underline{\boldsymbol{\theta}}} \sum_{s\in\mathcal{S}}\left\{\overline{\lambda}_s(\mathbf{e}^\top\boldsymbol{u}_s - d_s) + \underline{\lambda}_s(-\mathbf{e}^\top\boldsymbol{u}_s + d_s) - \overline{\boldsymbol{\theta}}_s^\top \boldsymbol{Q}_s\boldsymbol{u} + \underline{\boldsymbol{\theta}}_s^\top \boldsymbol{Q}_s\boldsymbol{u}\right\}
$$

$$
\begin{aligned}
\text{s.t.}\quad & \boldsymbol{B}\overline{\boldsymbol{\theta}}_s = \overline{\lambda}_s \cdot \mathbf{e} && \forall s\in\mathcal{S} \\
& \boldsymbol{B}\underline{\boldsymbol{\theta}}_s = \underline{\lambda}_s \cdot \mathbf{e} && \forall s\in\mathcal{S} \\
& \|\overline{\lambda}_s \cdot \hat{\boldsymbol{p}} - \overline{\boldsymbol{\theta}}_s\| \leq \xi_s && \forall s\in\mathcal{S} \\
& \|\underline{\lambda}_s \cdot \hat{\boldsymbol{p}} - \underline{\boldsymbol{\theta}}_s\| \leq \phi_s - \xi_s && \forall s\in\mathcal{S} \\
& \frac{\omega}{N}\sum_{i\in[N]}\boldsymbol{w}_i^\top(\boldsymbol{F}^\top\boldsymbol{u} - \boldsymbol{f}_{\mathrm{E}}) + (1-\omega)x - \tau \geq \frac{1-\omega}{N\varepsilon}\cdot\sum_{i\in[N]}y_i \\
& y_i \geq x - \boldsymbol{w}_i^\top(\boldsymbol{F}^\top\boldsymbol{u} - \boldsymbol{f}_{\mathrm{E}}) && \forall i\in[N] \\
& \boldsymbol{u}\in\mathbb{R}_+^{S\cdot A},\ x\in\mathbb{R},\ \boldsymbol{y}\in\mathbb{R}_+^N,\ \boldsymbol{\xi},\ \overline{\boldsymbol{\lambda}},\ \underline{\boldsymbol{\lambda}}\in\mathbb{R}_+^S,\ \overline{\boldsymbol{\theta}},\ \underline{\boldsymbol{\theta}}\in\mathbb{R}_+^{S\cdot S\cdot A\cdot S},
\end{aligned}
\tag{19}
$$

that is, the optimal solution sets for the outer minimization problems of these two problems are the same. To prove this, it suffices to show that for any feasible solution $(\boldsymbol{u}, x, \boldsymbol{y}) \in \mathbb{R}_+^{S \cdot A} \times \mathbb{R} \times \mathbb{R}_+^N$ that satisfies (17), the corresponding optimal values of the inner maximization problems of (18) and (19) are equal. The former is no smaller than the latter because the feasible region for $(\overline{\boldsymbol{\lambda}}, \underline{\boldsymbol{\lambda}}, \overline{\boldsymbol{\theta}}, \underline{\boldsymbol{\theta}})$ is no smaller than the one for the latter. The latter is no smaller than the former because for an arbitrary optimal solution $(\overline{\boldsymbol{\xi}}^\star, \underline{\boldsymbol{\xi}}^\star, \overline{\boldsymbol{\lambda}}^\star, \underline{\boldsymbol{\lambda}}^\star, \overline{\boldsymbol{\theta}}^\star, \underline{\boldsymbol{\theta}}^\star)$ of the corresponding inner maximization problem of (18), $(\overline{\boldsymbol{\xi}}^\star, \overline{\boldsymbol{\lambda}}^\star, \underline{\boldsymbol{\lambda}}^\star, \overline{\boldsymbol{\theta}}^\star, \underline{\boldsymbol{\theta}}^\star)$ is a feasible solution of the inner maximization problem of (19) that has the same objective value. We then note that (19) is equivalent to

$$
\min_{\boldsymbol{u},x,\boldsymbol{y}} \max_{\mu,\boldsymbol{\eta}} \quad \mu\left(\frac{1-\omega}{N\varepsilon} \cdot \mathbf{e}^\top \boldsymbol{y} - \frac{\omega}{N} \cdot \sum_{i\in[N]} \boldsymbol{w}_i^\top(\boldsymbol{F}^\top\boldsymbol{u} - \boldsymbol{f}_{\mathrm{E}}) - (1-\omega)x + \tau\right)
$$
$$
+ \sum_{i\in[N]} \eta_i(x - \boldsymbol{w}_i^\top(\boldsymbol{F}^\top\boldsymbol{u} - \boldsymbol{f}_{\mathrm{E}}) - y_i)
$$
$$
+ \max_{\overline{\boldsymbol{\xi}},\underline{\boldsymbol{\xi}},\overline{\boldsymbol{\lambda}},\underline{\boldsymbol{\lambda}},\overline{\boldsymbol{\theta}},\underline{\boldsymbol{\theta}}} \sum_{s\in\mathcal{S}} \left\{\overline{\lambda}_s(\mathbf{e}^\top\boldsymbol{u}_s - d_s) + \underline{\lambda}_s(-\mathbf{e}^\top\boldsymbol{u}_s + d_s) - \overline{\boldsymbol{\theta}}_s^\top\boldsymbol{Q}_s\boldsymbol{u} + \underline{\boldsymbol{\theta}}_s^\top\boldsymbol{Q}_s\boldsymbol{u}\right\}
$$
$$
\text{s.t.} \quad \boldsymbol{B}\overline{\boldsymbol{\theta}}_s = \overline{\lambda}_s \cdot \mathbf{e} \qquad\qquad\qquad\qquad \forall s\in\mathcal{S}
$$
$$
\boldsymbol{B}\underline{\boldsymbol{\theta}}_s = \underline{\lambda}_s \cdot \mathbf{e} \qquad\qquad\qquad\qquad \forall s\in\mathcal{S}
$$
$$
\|\overline{\lambda}_s \cdot \hat{\boldsymbol{p}} - \overline{\boldsymbol{\theta}}_s\| \le \xi_s \qquad\qquad\qquad \forall s\in\mathcal{S}
$$
$$
\|\underline{\lambda}_s \cdot \hat{\boldsymbol{p}} - \underline{\boldsymbol{\theta}}_s\| \le \phi_s - \xi_s \qquad\qquad\quad \forall s\in\mathcal{S}
$$
$$
\boldsymbol{u}\in\mathbb{R}_+^{S\cdot A},\ x\in\mathbb{R},\ \boldsymbol{y}\in\mathbb{R}_+^N,\ \boldsymbol{\xi},\ \overline{\boldsymbol{\lambda}},\ \underline{\boldsymbol{\lambda}}\in\mathbb{R}_+^S,\ \overline{\boldsymbol{\theta}},\ \underline{\boldsymbol{\theta}}\in\mathbb{R}_+^{S\cdot S\cdot A\cdot S},\ \mu\in\mathbb{R}_+,\ \boldsymbol{\eta}\in\mathbb{R}_+^N
$$
$$(20)$$

because

$$
\max_{\mu\in\mathbb{R}_+} \mu\left(\frac{1-\omega}{N\varepsilon} \cdot \mathbf{e}^\top\boldsymbol{y} - \frac{\omega}{N} \cdot \sum_{i\in[N]} \boldsymbol{w}_i^\top(\boldsymbol{F}^\top\boldsymbol{u} - \boldsymbol{f}_{\mathrm{E}}) - (1-\omega)x + \tau\right)
$$
$$
= \begin{cases} 0 & \text{if } \dfrac{1-\omega}{N\varepsilon} \cdot \mathbf{e}^\top\boldsymbol{y} - \dfrac{\omega}{N} \cdot \sum_{i\in[N]} \boldsymbol{w}_i^\top(\boldsymbol{F}^\top\boldsymbol{u} - \boldsymbol{f}_{\mathrm{E}}) - (1-\omega)x + \tau \le 0 \\[1em] \infty & \text{otherwise,} \end{cases}
$$

and

$$
\max_{\boldsymbol{\eta}\in\mathbb{R}_+^N} \sum_{i\in[N]} \eta_i(x - \boldsymbol{w}_i^\top(\boldsymbol{F}^\top\boldsymbol{u} - \boldsymbol{f}_{\mathrm{E}}) - y_i)
$$
$$
= \begin{cases} 0 & \text{if } x - \boldsymbol{w}_i^\top(\boldsymbol{F}^\top\boldsymbol{u} - \boldsymbol{f}_{\mathrm{E}}) - y_i \le 0 \quad \forall i\in[N] \\[0.5em] \infty & \text{otherwise.} \end{cases}
$$

Our conclusion then follows by aggregating the two inner maximizations in (20). $\qquad\square$

**Theorem 1** *Let $\{(\boldsymbol{u}^k, x^k, \boldsymbol{y}^k, \mu^k, \boldsymbol{\eta}^k, \boldsymbol{\xi}^k, \overline{\boldsymbol{\lambda}}^k, \underline{\boldsymbol{\lambda}}^k, \overline{\boldsymbol{\theta}}^k, \underline{\boldsymbol{\theta}}^k)\}_{k=0}^M$ be the sequence of output of Algorithm 1. When the stepsizes $\nu, \sigma$ satisfy*

$$
\frac{1}{2\nu} \cdot \left\|\begin{matrix} \boldsymbol{u} - \boldsymbol{u}' \\ x - x' \\ \boldsymbol{y} - \boldsymbol{y}' \end{matrix}\right\|_2^2 + \frac{1}{2\sigma} \cdot \left\|\begin{matrix} \mu - \mu' \\ \boldsymbol{\eta} - \boldsymbol{\eta}' \\ \boldsymbol{\xi} - \boldsymbol{\xi}' \\ \overline{\boldsymbol{\lambda}} - \overline{\boldsymbol{\lambda}}' \\ \underline{\boldsymbol{\lambda}} - \underline{\boldsymbol{\lambda}}' \\ \overline{\boldsymbol{\theta}} - \overline{\boldsymbol{\theta}}' \\ \underline{\boldsymbol{\theta}} - \underline{\boldsymbol{\theta}}' \end{matrix}\right\|_2^2 - \langle \boldsymbol{D}((\boldsymbol{u} - \boldsymbol{u}')^\top, x - x', (\boldsymbol{y} - \boldsymbol{y}')^\top)^\top,
$$
$$
(\mu - \mu', (\boldsymbol{\eta} - \boldsymbol{\eta}')^\top, (\boldsymbol{\xi} - \boldsymbol{\xi}')^\top, (\overline{\boldsymbol{\lambda}} - \overline{\boldsymbol{\lambda}}')^\top, (\underline{\boldsymbol{\lambda}} - \underline{\boldsymbol{\lambda}}')^\top, (\overline{\boldsymbol{\theta}} - \overline{\boldsymbol{\theta}}')^\top, (\underline{\boldsymbol{\theta}} - \underline{\boldsymbol{\theta}}')^\top)^\top \rangle \ge 0
$$

*for any $(\boldsymbol{u}, x, \boldsymbol{y})$, $(\boldsymbol{u}', x', \boldsymbol{y}') \in \mathbb{R}^{S \cdot A} \times \mathbb{R} \times \mathbb{R}^N$ and $(\mu, \boldsymbol{\eta}, \boldsymbol{\xi}, \overline{\boldsymbol{\lambda}}, \underline{\boldsymbol{\lambda}}, \overline{\boldsymbol{\theta}}, \underline{\boldsymbol{\theta}})$, $(\mu', \boldsymbol{\eta}', \boldsymbol{\xi}', \overline{\boldsymbol{\lambda}}', \underline{\boldsymbol{\lambda}}', \overline{\boldsymbol{\theta}}', \underline{\boldsymbol{\theta}}') \in \mathbb{R} \times \mathbb{R}^N \times \mathbb{R}^S \times \mathbb{R}^S \times \mathbb{R}^S \times \mathbb{R}^{S \cdot S \cdot A \cdot S} \times \mathbb{R}^{S \cdot S \cdot A \cdot S}$, then for any feasible solution of (4), it holds that*

$$g(\boldsymbol{u}^{\text{avg}}, x^{\text{avg}}, \boldsymbol{y}^{\text{avg}}, \mu, \boldsymbol{\eta}, \boldsymbol{\xi}, \overline{\boldsymbol{\lambda}}, \underline{\boldsymbol{\lambda}}, \overline{\boldsymbol{\theta}}, \underline{\boldsymbol{\theta}}) - g(\boldsymbol{u}, x, \boldsymbol{y}, \mu^{\text{avg}}, \boldsymbol{\eta}^{\text{avg}}, \boldsymbol{\xi}^{\text{avg}}, \overline{\boldsymbol{\lambda}}^{\text{avg}}, \underline{\boldsymbol{\lambda}}^{\text{avg}}, \overline{\boldsymbol{\theta}}^{\text{avg}}, \underline{\boldsymbol{\theta}}^{\text{avg}}) = \mathcal{O}(1/M),$$

*where* $\boldsymbol{u}^{\text{avg}} = \frac{1}{M}\sum_{k\in[M]} \boldsymbol{u}^k$, $x^{\text{avg}} = \frac{1}{M}\sum_{k\in[M]} x^k$, $\boldsymbol{y}^{\text{avg}} = \frac{1}{M}\sum_{k\in[M]} \boldsymbol{y}^k$, $\mu^{\text{avg}} = \frac{1}{M}\sum_{k\in[M]} \mu^k$, $\boldsymbol{\eta}^{\text{avg}} = \frac{1}{M}\sum_{k\in[M]} \boldsymbol{\eta}^k$, $\boldsymbol{\xi}^{\text{avg}} = \frac{1}{M}\sum_{k\in[M]} \boldsymbol{\xi}^k$, $\overline{\boldsymbol{\lambda}}^{\text{avg}} = \frac{1}{M}\sum_{k\in[M]} \overline{\boldsymbol{\lambda}}^k$, $\underline{\boldsymbol{\lambda}}^{\text{avg}} = \frac{1}{M}\sum_{k\in[M]} \underline{\boldsymbol{\lambda}}^k$, $\overline{\boldsymbol{\theta}}^{\text{avg}} = \frac{1}{M}\sum_{k\in[M]} \overline{\boldsymbol{\theta}}^k$, *and* $\underline{\boldsymbol{\theta}}^{\text{avg}} = \frac{1}{M}\sum_{k\in[M]} \underline{\boldsymbol{\theta}}^k$, *and we express the objective function of problem (4) as a function of its decision variables as* $g(\boldsymbol{u}, x, \boldsymbol{y}, \mu, \boldsymbol{\eta}, \boldsymbol{\xi}, \overline{\boldsymbol{\lambda}}, \underline{\boldsymbol{\lambda}}, \overline{\boldsymbol{\theta}}, \underline{\boldsymbol{\theta}})$.

Note that the convergence rate in Theorem 1 can be achieved by stepsizes satisfying $\nu\sigma \leq (1/G^2)$, where $G = \|\boldsymbol{D}\|_{\text{Op}}$ with $\|\cdot\|_{\text{Op}}$ being the operator norm and with the coefficient matrix $\boldsymbol{D} \in \mathbb{R}^{(1+N+3S+2S\cdot S\cdot A\cdot S)\times(S\cdot A+1+N)}$ satisfying that the objective function of (4) can be rewritten $\langle \boldsymbol{D}(\boldsymbol{u}^\top, x, \boldsymbol{y}^\top)^\top, (\mu, \boldsymbol{\eta}^\top, \boldsymbol{\xi}^\top, \overline{\boldsymbol{\lambda}}^\top, \underline{\boldsymbol{\lambda}}^\top, \overline{\boldsymbol{\theta}}^\top, \underline{\boldsymbol{\theta}}^\top)^\top \rangle$ (Chambolle & Pock, 2016).

*Proof of Proposition 4*  For ease of description, let

$$\boldsymbol{a} = -\frac{\mu\omega}{N} \cdot \sum_{i\in[N]} \boldsymbol{F}\boldsymbol{w}_i - \sum_{i\in[N]} \eta_i \boldsymbol{F}\boldsymbol{w}_i + \boldsymbol{\lambda} - \sum_{s\in\mathcal{S}} \boldsymbol{Q}_s^\top(\overline{\boldsymbol{\theta}}_s - \underline{\boldsymbol{\theta}}_s),$$

where

$$\boldsymbol{\lambda} = (\underbrace{\overline{\lambda}_1 - \underline{\lambda}_1, \dots, \overline{\lambda}_1 - \underline{\lambda}_1}_{A}, \dots, \underbrace{\overline{\lambda}_S - \underline{\lambda}_S, \dots, \overline{\lambda}_S - \underline{\lambda}_S}_{A}) \in \mathbb{R}^{S\cdot A},$$

and let

$$b = -\mu(1-\omega) + \sum_{i\in[N]} \eta_i$$

and

$$\boldsymbol{c} = \frac{\mu(1-\omega)}{N\varepsilon} \cdot \mathbf{e} - \boldsymbol{\eta}.$$

It is then sufficient to solve the problem

$$\min_{\boldsymbol{u}\in\mathbb{R}_+^{S\cdot A}} \boldsymbol{a}^\top\boldsymbol{u} + \frac{1}{2\nu}\cdot\|\boldsymbol{u} - \boldsymbol{u}'\|_2^2 \tag{21}$$

for the optimal $\boldsymbol{u}^\star$,

$$\min_{x\in\mathbb{R}} bx + \frac{1}{2\nu}(x - x')^2 \tag{22}$$

for the optimal $x^\star$, and

$$\min_{\boldsymbol{y}\in\mathbb{R}_+^N} \boldsymbol{c}^\top\boldsymbol{y} + \frac{1}{2\nu}\cdot\|\boldsymbol{y} - \boldsymbol{y}'\|_2^2 \tag{23}$$

for the optimal $\boldsymbol{y}^\star$. Problem (21) can be decomposed into $SA$ subproblems, where for every $s \in \mathcal{S}, a \in \mathcal{A}$, the $sa$-th one is a single-variable quadratic program as follows:

$$\min_{u\in\mathbb{R}_+} a_{s,a}u + \frac{1}{2\nu}(u - u'_{s,a})^2.$$

Therefore, we have $\boldsymbol{u}^\star = [\boldsymbol{u}' - \nu \cdot \boldsymbol{a}]_+$. Similarly, we have $x^\star = x' - \nu b$ and $\boldsymbol{y}^\star = [\boldsymbol{y}' - \nu \cdot \boldsymbol{c}]_+$. The time complexity of computing $\boldsymbol{a}$ is $\mathcal{O}(S^2A)$, and the on of computing $b$ and $\boldsymbol{c}$ are both $\mathcal{O}(N)$, leading to our result. □

*Proof of Proposition 5*  The optimal solution for problem $\mathscr{D}^\mu(\boldsymbol{u}, x, \boldsymbol{y}; \mu')$ is

$$\mu^\star = \left[\mu' - \sigma\left(-\frac{1-\omega}{N\varepsilon}\cdot\mathbf{e}^\top\boldsymbol{y} + \frac{\omega}{N}\cdot\sum_{i\in[N]} \boldsymbol{w}_i^\top(\boldsymbol{F}^\top\boldsymbol{u} - \boldsymbol{f}_{\text{E}}) + (1-\omega)x - \tau\right)\right]_+.$$

The time complexity of computing $\mathbf{e}^\top\boldsymbol{y}$ is $\mathcal{O}(N)$, and hte one for computing $\sum_{i\in[N]}(\boldsymbol{F}\boldsymbol{w}_i)^\top\boldsymbol{u}$ is $\mathcal{O}(SA)$, leading to our result. □

*Proof of Proposition 6* Problem $\mathscr{D}^{\boldsymbol{\eta}}(\boldsymbol{u}, x, \boldsymbol{y}; \boldsymbol{\eta}')$ can be decomposed in to $N$ subproblems, where for every $i \in [N]$, the $i$-th one is a single-variable quadratic program as follows:

$$
\min_{\eta \in \mathbb{R}_+} \eta(-x + \boldsymbol{w}_i^\top (\boldsymbol{F}^\top \boldsymbol{u} - \boldsymbol{f}_{\mathrm{E}}) + y_i) + \frac{1}{2\sigma} \cdot (\eta - \eta_i')^2.
$$

The optimal solution of this problem is $[\eta_i' - \sigma(-x + \boldsymbol{w}_i^\top (\boldsymbol{F}^\top \boldsymbol{u} - \boldsymbol{f}_{\mathrm{E}}) + y_i)]_+$. Our conclusion follows by the fact that the computation of $(\boldsymbol{F}\boldsymbol{w}_i)^\top \boldsymbol{u}$ takes time $\mathcal{O}(SA)$ for every $i \in [N]$. $\square$

**Lemma 2** *Let the vector* $\boldsymbol{\zeta} \in \mathbb{R}^{2+2S \cdot A \cdot S}$ *and the positive definite matrix* $\boldsymbol{A} \in \mathbb{R}^{(2+2S \cdot A \cdot S) \times (2+2S \cdot A \cdot S)}$ *be arbitrarily taken. It holds that:*

$$
\boldsymbol{\zeta}^\top \boldsymbol{A} \boldsymbol{\zeta} \geq \frac{\sigma_{\min} \cdot \|\boldsymbol{\zeta}\|_1^2}{2 + 2SAS},
$$

*where* $\sigma_{\min} > 0$ *is the smallest eigenvalue of* $\boldsymbol{A}$.

*Proof of Lemma 2* Conducting the eigenvalue decomposition of $\boldsymbol{A}$ as $\boldsymbol{A} = \boldsymbol{U}^\top \boldsymbol{\Lambda} \boldsymbol{U}$, we then have

$$
\begin{aligned}
\boldsymbol{\zeta}^\top \boldsymbol{A} \boldsymbol{\zeta} =\ & (\boldsymbol{U}\boldsymbol{\zeta})^\top \boldsymbol{\Lambda} (\boldsymbol{U}\boldsymbol{\zeta}) \\
=\ & |\boldsymbol{U}\boldsymbol{\zeta}|^\top \boldsymbol{\Lambda} |\boldsymbol{U}\boldsymbol{\zeta}| \\
\geq\ & \sigma_{\min} \cdot |\boldsymbol{U}\boldsymbol{\zeta}|^\top |\boldsymbol{U}\boldsymbol{\zeta}| \\
=\ & \sigma_{\min} \cdot \|\boldsymbol{U}\boldsymbol{\zeta}\|_2^2 \\
=\ & \sigma_{\min} \cdot \|\boldsymbol{\zeta}\|_2^2 \\
\geq\ & \frac{\sigma_{\min} \cdot \|\boldsymbol{\zeta}\|_1^2}{2 + 2SAS},
\end{aligned}
$$

where the second equality holds because $\boldsymbol{A}$ is positive definite, and the last one is because the matrix $\boldsymbol{U}$ is orthogonal. The Cauchy-Schwarz inequality

$$
\sqrt{2 + 2SAS} \cdot \|\boldsymbol{\zeta}\|_2 \geq \|\boldsymbol{\zeta}\|_1
$$

leads to the last inequality. $\square$

**Lemma 3** *Let*

$$
\mathcal{K}_s(\xi) = \left\{ (\overline{\lambda}, \underline{\lambda}, \overline{\boldsymbol{\theta}}, \underline{\boldsymbol{\theta}}) \in \mathbb{R}_+ \times \mathbb{R}_+ \times \mathbb{R}_+^{S \cdot A \cdot S} \times \mathbb{R}_+^{S \cdot A \cdot S} \left| \begin{array}{l} \|\overline{\boldsymbol{\theta}} - \overline{\lambda} \cdot \hat{\boldsymbol{p}}\|_\infty \leq \xi \\ \|\underline{\boldsymbol{\theta}} - \underline{\lambda} \cdot \hat{\boldsymbol{p}}\|_\infty \leq \phi_s - \xi \\ \overline{\lambda} \cdot \mathbf{e} = \boldsymbol{B}\overline{\boldsymbol{\theta}} \\ \underline{\lambda} \cdot \mathbf{e} = \boldsymbol{B}\underline{\boldsymbol{\theta}} \end{array} \right. \right\}.
$$

*For any $\rho'' \in \mathbb{R}_+$, the problem:*

$$
\begin{aligned}
\min_{(\overline{\lambda}, \underline{\lambda}, \overline{\boldsymbol{\theta}}, \underline{\boldsymbol{\theta}}) \in \mathcal{K}_s(\xi)} \quad & (\overline{\lambda} - \underline{\lambda})(d_s - \mathbf{e}^\top \boldsymbol{u}_s) + (\overline{\boldsymbol{\theta}} - \underline{\boldsymbol{\theta}})^\top \boldsymbol{Q}_s \boldsymbol{u} + \frac{1}{2\sigma} \Big( (\xi - \xi')^2 + \\
& (\overline{\lambda} - \overline{\lambda}')^2 + (\underline{\lambda} - \underline{\lambda}')^2 + \|\overline{\boldsymbol{\theta}} - \overline{\boldsymbol{\theta}}'\|_2^2 + \|\underline{\boldsymbol{\theta}} - \underline{\boldsymbol{\theta}}'\|_2^2 \Big) \\
\text{s.t.} \quad & \|(\overline{\lambda}, \underline{\lambda}, \overline{\boldsymbol{\theta}}, \underline{\boldsymbol{\theta}})\|_1 = \rho''
\end{aligned} \tag{24}
$$

*can attain its optimal value. Moreover, for any $L < \infty$, there exists $\rho' > 0$ such that the optimal value of problem* (24) *equipped with any $\rho'' \geq \rho'$ is strictly larger than $L$.*

*Proof of Lemma 3* Let

$$
\boldsymbol{z} = \begin{bmatrix} d_s - \mathbf{e}^\top \boldsymbol{u}_s - (1/\sigma)\overline{\lambda}' \\ -d_s + \mathbf{e}^\top \boldsymbol{u}_s - (1/\sigma)\underline{\lambda}' \\ \boldsymbol{Q}_s \boldsymbol{u} - (1/\sigma) \cdot \overline{\boldsymbol{\theta}}' \\ -\boldsymbol{Q}_s \boldsymbol{u} - (1/\sigma) \cdot \underline{\boldsymbol{\theta}}' \end{bmatrix}
$$

and

$$\boldsymbol{A} = \frac{1}{2\sigma} \cdot \boldsymbol{I}$$

where $\boldsymbol{I} \in \mathbb{R}^{(2+2S \cdot A \cdot S) \times (2+2S \cdot A \cdot S)}$ is an identity matrix. We then can re-write (24) in a simplified form as follows:

$$\min_{\substack{(\overline{\lambda}, \underline{\lambda}, \overline{\boldsymbol{\theta}}, \underline{\boldsymbol{\theta}}) \in \mathcal{K}_s(\xi): \\ \|(\overline{\lambda}, \underline{\lambda}, \overline{\boldsymbol{\theta}}, \underline{\boldsymbol{\theta}})\|_1 = \rho''}} \boldsymbol{z}^\top (\overline{\lambda}, \underline{\lambda}, \overline{\boldsymbol{\theta}}^\top, \underline{\boldsymbol{\theta}}^\top)^\top + (\overline{\lambda}, \underline{\lambda}, \overline{\boldsymbol{\theta}}^\top, \underline{\boldsymbol{\theta}}^\top) \boldsymbol{A} (\overline{\lambda}, \underline{\lambda}, \overline{\boldsymbol{\theta}}^\top, \underline{\boldsymbol{\theta}}^\top)^\top.$$

We have Note that

$$\min_{\substack{(\overline{\lambda}, \underline{\lambda}, \overline{\boldsymbol{\theta}}, \underline{\boldsymbol{\theta}}) \in \mathcal{K}_s(\xi): \\ \|(\overline{\lambda}, \underline{\lambda}, \overline{\boldsymbol{\theta}}, \underline{\boldsymbol{\theta}})\|_1 = \rho''}} \boldsymbol{z}^\top (\overline{\lambda}, \underline{\lambda}, \overline{\boldsymbol{\theta}}^\top, \underline{\boldsymbol{\theta}}^\top)^\top + (\overline{\lambda}, \underline{\lambda}, \overline{\boldsymbol{\theta}}^\top, \underline{\boldsymbol{\theta}}^\top) \boldsymbol{A} (\overline{\lambda}, \underline{\lambda}, \overline{\boldsymbol{\theta}}^\top, \underline{\boldsymbol{\theta}}^\top)^\top.$$

$$\geq \min_{\substack{(\overline{\lambda}, \underline{\lambda}, \overline{\boldsymbol{\theta}}, \underline{\boldsymbol{\theta}}) \in \mathcal{K}_s(\xi): \\ \|(\overline{\lambda}, \underline{\lambda}, \overline{\boldsymbol{\theta}}, \underline{\boldsymbol{\theta}})\|_1 = \rho''}} \boldsymbol{z}^\top (\overline{\lambda}, \underline{\lambda}, \overline{\boldsymbol{\theta}}^\top, \underline{\boldsymbol{\theta}}^\top)^\top + \frac{\sigma_{\min} \cdot \rho''^2}{2 + 2SAS}$$

$$\geq \min_{\substack{(\overline{\lambda}, \underline{\lambda}, \overline{\boldsymbol{\theta}}, \underline{\boldsymbol{\theta}}) \in \mathbb{R}_+ \times \mathbb{R}_+ \times \mathbb{R}_+^{S \cdot A \cdot S} \times \mathbb{R}_+^{S \cdot A \cdot S}: \\ \|(\overline{\lambda}, \underline{\lambda}, \overline{\boldsymbol{\theta}}, \underline{\boldsymbol{\theta}})\|_1 = \rho''}} \boldsymbol{z}^\top (\overline{\lambda}, \underline{\lambda}, \overline{\boldsymbol{\theta}}^\top, \underline{\boldsymbol{\theta}}^\top)^\top + \frac{\sigma_{\min} \cdot \rho''^2}{2 + 2SAS} \qquad (25)$$

$$= \rho'' \cdot \left\{ \min_{\substack{(\overline{\lambda}, \underline{\lambda}, \overline{\boldsymbol{\theta}}, \underline{\boldsymbol{\theta}}) \in \mathbb{R}_+ \times \mathbb{R}_+ \times \mathbb{R}_+^{S \cdot A \cdot S} \times \mathbb{R}_+^{S \cdot A \cdot S}: \\ \|(\overline{\lambda}, \underline{\lambda}, \overline{\boldsymbol{\theta}}, \underline{\boldsymbol{\theta}})\|_1 = 1}} \boldsymbol{z}^\top (\overline{\lambda}, \underline{\lambda}, \overline{\boldsymbol{\theta}}^\top, \underline{\boldsymbol{\theta}}^\top)^\top \right\} + \frac{\sigma_{\min} \cdot \rho''^2}{2 + 2SAS}.$$

We argue that, all four minimization problems in (25) can attain its optimal value. To observe this, note that

$$\begin{cases} \|(\overline{\lambda}, \underline{\lambda}, \overline{\boldsymbol{\theta}}, \underline{\boldsymbol{\theta}})\|_1 = \rho'' \\ (\overline{\lambda}, \underline{\lambda}, \overline{\boldsymbol{\theta}}, \underline{\boldsymbol{\theta}}) \in \mathbb{R}_+ \times \mathbb{R}_+ \times \mathbb{R}_+^{S \cdot A \cdot S} \times \mathbb{R}_+^{S \cdot A \cdot S} \end{cases} \iff \begin{cases} \overline{\lambda} + \underline{\lambda} + \mathbf{e}^\top \overline{\boldsymbol{\theta}} + \mathbf{e}^\top \underline{\boldsymbol{\theta}} = \rho'' \\ (\overline{\lambda}, \underline{\lambda}, \overline{\boldsymbol{\theta}}, \underline{\boldsymbol{\theta}}) \in \mathbb{R}_+ \times \mathbb{R}_+ \times \mathbb{R}_+^{S \cdot A \cdot S} \times \mathbb{R}_+^{S \cdot A \cdot S} \end{cases}$$
$$(26)$$

is true for any $\rho'' \in \mathbb{R}_+$. Conduct this constraint substitution to the optimization problems in (25), one can then observe that each one of them becomes a convex optimization problem where the objective function is continuous, and the feasible region is non-empty, closed, and bounded. A feasible solution to the first three convex optimization problems is $(\overline{\lambda}, \underline{\lambda}, \overline{\boldsymbol{\theta}}, \underline{\boldsymbol{\theta}}) = (\rho''/(2 \cdot \|(1, \hat{\boldsymbol{p}})\|_1)) \cdot (1, 1, \hat{\boldsymbol{p}}, \hat{\boldsymbol{p}})$, and the one to the last convex optimization problem could be $(\overline{\lambda}, \underline{\lambda}, \overline{\boldsymbol{\theta}}, \underline{\boldsymbol{\theta}}) = (1/(2 \cdot \|(1, \hat{\boldsymbol{p}})\|_1)) \cdot (1, 1, \hat{\boldsymbol{p}}, \hat{\boldsymbol{p}})$. The boundedness of these convex optimization problems is implied the by constraints in the right-hand side of (26). Therefore, all four convex optimization problems can attain their optimal values by the Weierstrass theorem, thus the four minimization problems in (25) can also attain their optimality. In problem (25), the first inequality follows from Lemma 2, and the last one holds due to the fact that the minimization problem on the right-hand side is a relaxation of left-hand one. The equality simply follows by considering the variable substitution $(\overline{\lambda}, \underline{\lambda}, \overline{\boldsymbol{\theta}}, \underline{\boldsymbol{\theta}}) \leftarrow \rho'' \cdot (\overline{\lambda}, \underline{\lambda}, \overline{\boldsymbol{\theta}}, \underline{\boldsymbol{\theta}})$. The right-hand side of the equality in (25) is a single-variable quadratic function of $\rho''$ with strictly positive coefficient for the quadratic term, whose value could be arbitrarily large with $\rho'' \to \infty$. Hence, with $\rho'' \to \infty$, the optimal value of (24) will tends to infinity by (25). $\qquad \square$

**Proposition 10** *For any $\xi \in [0, \phi_{\hat{s}}]$, problem (33) is well-defined.*

*Proof of Proposition 10*  Let the feasible region of problem (33) be denoted as

$$\mathcal{K}_{\hat{s}}(\xi) = \left\{ (\overline{\lambda}, \underline{\lambda}, \overline{\boldsymbol{\theta}}, \underline{\boldsymbol{\theta}}) \in \mathbb{R}_+ \times \mathbb{R}_+ \times \mathbb{R}_+^{S \cdot A \cdot S} \times \mathbb{R}_+^{S \cdot A \cdot S} \,\middle|\, \begin{aligned} &\|\overline{\boldsymbol{\theta}} - \overline{\lambda} \cdot \hat{\boldsymbol{p}}\|_\infty \leq \xi \\ &\|\underline{\boldsymbol{\theta}} - \underline{\lambda} \cdot \hat{\boldsymbol{p}}\|_\infty \leq \phi_{\hat{s}} - \xi \\ &\overline{\lambda} \cdot \mathbf{e} = \boldsymbol{B}\overline{\boldsymbol{\theta}} \\ &\underline{\lambda} \cdot \mathbf{e} = \boldsymbol{B}\underline{\boldsymbol{\theta}} \end{aligned} \right\}.$$

For ease of exposition, let

$$\boldsymbol{z} = \begin{bmatrix} d_{\hat{s}} - \mathbf{e}^\top \boldsymbol{u}_{\hat{s}} - (1/\sigma)\overline{\lambda}' \\ -d_{\hat{s}} + \mathbf{e}^\top \boldsymbol{u}_{\hat{s}} - (1/\sigma)\underline{\lambda}' \\ \boldsymbol{Q}_{\hat{s}} \boldsymbol{u} - (1/\sigma) \cdot \overline{\boldsymbol{\theta}}' \\ -\boldsymbol{Q}_{\hat{s}} \boldsymbol{u} - (1/\sigma) \cdot \underline{\boldsymbol{\theta}}' \end{bmatrix}$$

and

$$\boldsymbol{A} = \frac{1}{2\sigma} \cdot \boldsymbol{I},$$

where $\boldsymbol{I} \in \mathbb{R}^{(2+2S\cdot A\cdot S)\times(2+2S\cdot A\cdot S)}$ is an identity matrix. It then suffices to prove that the problem

$$\min_{(\overline{\lambda},\underline{\lambda},\overline{\boldsymbol{\theta}},\underline{\boldsymbol{\theta}})\in\mathcal{K}_{\hat{s}}(\xi)} \boldsymbol{z}^\top(\overline{\lambda},\underline{\lambda},\overline{\boldsymbol{\theta}}^\top,\underline{\boldsymbol{\theta}}^\top)^\top + (\overline{\lambda},\underline{\lambda},\overline{\boldsymbol{\theta}}^\top,\underline{\boldsymbol{\theta}}^\top)\boldsymbol{A}(\overline{\lambda},\underline{\lambda},\overline{\boldsymbol{\theta}}^\top,\underline{\boldsymbol{\theta}}^\top)^\top$$

can attain its minimal value for any $\xi \in [0, \phi_{\hat{s}}]$.

Let us arbitrarily fix $\rho \in \mathbb{R}_{++}$ and $\xi \in [0, \phi_{\hat{s}}]$. The optimization problem

$$\min_{(\overline{\lambda},\underline{\lambda},\overline{\boldsymbol{\theta}},\underline{\boldsymbol{\theta}})\in\mathcal{K}_{\hat{s}}(\xi)} \boldsymbol{z}^\top(\overline{\lambda},\underline{\lambda},\overline{\boldsymbol{\theta}}^\top,\underline{\boldsymbol{\theta}}^\top)^\top + (\overline{\lambda},\underline{\lambda},\overline{\boldsymbol{\theta}}^\top,\underline{\boldsymbol{\theta}}^\top)\boldsymbol{A}(\overline{\lambda},\underline{\lambda},\overline{\boldsymbol{\theta}}^\top,\underline{\boldsymbol{\theta}}^\top)^\top \tag{27}$$
$$\text{s.t.} \qquad \|(\overline{\lambda},\underline{\lambda},\overline{\boldsymbol{\theta}},\underline{\boldsymbol{\theta}})\|_1 \le \rho$$

has a feasible region that is non-empty (where $(\overline{\lambda},\underline{\lambda},\overline{\boldsymbol{\theta}},\underline{\boldsymbol{\theta}}) = \boldsymbol{0}$ is a feasible solution), bounded (that is implied by the inequality constraint with an $\ell_1$-norm) and closed, and a continuous objective function. Hence, this problem can attain its optimality by the Weierstrass theorem. Let $L$ denote the optimal value of this problem. Lemma 3 ensures the existence of some $\rho' < \infty$ such that for all $\rho'' > \rho'$, problem

$$\min_{\substack{(\overline{\lambda},\underline{\lambda},\overline{\boldsymbol{\theta}},\underline{\boldsymbol{\theta}})\in\mathcal{K}_{\hat{s}}(\xi): \\ \|(\overline{\lambda},\underline{\lambda},\overline{\boldsymbol{\theta}},\underline{\boldsymbol{\theta}})\|_1=\rho''}} \boldsymbol{z}^\top(\overline{\lambda},\underline{\lambda},\overline{\boldsymbol{\theta}}^\top,\underline{\boldsymbol{\theta}}^\top)^\top + (\overline{\lambda},\underline{\lambda},\overline{\boldsymbol{\theta}}^\top,\underline{\boldsymbol{\theta}}^\top)\boldsymbol{A}(\overline{\lambda},\underline{\lambda},\overline{\boldsymbol{\theta}}^\top,\underline{\boldsymbol{\theta}}^\top)^\top$$

can attain its optimality, and the optimal value is strictly larger than $L$. This fact, and the fact that (27) is a restriction of problem (33) together, implies that the optimal solution $(\overline{\lambda}^\star, \underline{\lambda}^\star, \overline{\boldsymbol{\theta}}^\star, \underline{\boldsymbol{\theta}}^\star)$ of problem (33) satisfies $\|(\overline{\lambda}^\star, \underline{\lambda}^\star, \overline{\boldsymbol{\theta}}^\star, \underline{\boldsymbol{\theta}}^\star)\|_1 \ne \rho'' \; \forall \rho'' > \rho'$. Therefore, problem (33) is equivalent to the problem

$$\min_{(\overline{\lambda},\underline{\lambda},\overline{\boldsymbol{\theta}},\underline{\boldsymbol{\theta}})\in\mathcal{K}_{\hat{s}}(\xi)} \boldsymbol{z}^\top(\overline{\lambda},\underline{\lambda},\overline{\boldsymbol{\theta}}^\top,\underline{\boldsymbol{\theta}}^\top)^\top + (\overline{\lambda},\underline{\lambda},\overline{\boldsymbol{\theta}}^\top,\underline{\boldsymbol{\theta}}^\top)\boldsymbol{A}(\overline{\lambda},\underline{\lambda},\overline{\boldsymbol{\theta}}^\top,\underline{\boldsymbol{\theta}}^\top)^\top \tag{28}$$
$$\text{s.t.} \qquad \|(\overline{\lambda},\underline{\lambda},\overline{\boldsymbol{\theta}},\underline{\boldsymbol{\theta}})\|_1 \le \rho',$$

which can attain its optimality following from a similar argument for (27). Therefore, (33) can also attain its minimum. $\qquad\square$

**Lemma 4** *Let $a \in \mathbb{R}_{++}, \boldsymbol{b} \in \mathbb{R}^S, \boldsymbol{x}' \in \mathbb{R}^S, \underline{\boldsymbol{x}}, \overline{\boldsymbol{x}} \in \mathbb{R}_+^S : \underline{\boldsymbol{x}} \le \overline{\boldsymbol{x}}$ be arbitrarily taken. It holds that*

$$a \cdot \overline{\boldsymbol{x}}^\top \overline{\boldsymbol{x}} + |\boldsymbol{b} - 2a \cdot \boldsymbol{x}'|^\top \overline{\boldsymbol{x}} + a \cdot \boldsymbol{x}'^\top \boldsymbol{x}' \ge \max_{\boldsymbol{x}\in\mathbb{R}^S:\underline{\boldsymbol{x}}\le\boldsymbol{x}\le\overline{\boldsymbol{x}}} a \cdot \|\boldsymbol{x} - \boldsymbol{x}'\|_2^2 + \boldsymbol{b}^\top \boldsymbol{x}.$$

*Proof of Lemma 4* Arbitrarily fix a feasible solution $\boldsymbol{x}$ of the maximization problem in our argument. It holds that:

$$a \cdot \|\boldsymbol{x} - \boldsymbol{x}'\|_2^2 + \boldsymbol{b}^\top \boldsymbol{x}$$
$$= \sum_{s\in[S]} \left\{ a(x_s - x_s')^2 + b_s x_s \right\}$$
$$= \sum_{s\in[S]} \left\{ a \cdot x_s^2 + (b_s - 2a \cdot x_s')x_s \right\} + a \cdot \boldsymbol{x}'^\top \boldsymbol{x}'$$
$$\le \sum_{s\in[S]} \left\{ a \cdot \overline{x}_s^2 + |b_s - 2a \cdot x_s'| \cdot \overline{x}_s \right\} + a \cdot \boldsymbol{x}'^\top \boldsymbol{x}'$$
$$= a \cdot \overline{\boldsymbol{x}}^\top \overline{\boldsymbol{x}} + |\boldsymbol{b} - 2a \cdot \boldsymbol{x}'|^\top \overline{\boldsymbol{x}} + a \cdot \boldsymbol{x}'^\top \boldsymbol{x}',$$

where the inequality holds because $\boldsymbol{0} \le \underline{\boldsymbol{x}} \le \boldsymbol{x} \le \overline{\boldsymbol{x}}$ and $a > 0$. $\qquad\square$

**Proposition 11** *Let the optimal solution of problem $\overline{K}_{\hat{s}}(\xi)$ with $\xi > 0$ be $(\overline{\lambda}^\star, \overline{\boldsymbol{\theta}}^\star)$. It holds that*

$$
\begin{aligned}
\overline{\lambda}^\star \leq \ & \overline{\lambda}' - \left(\sigma(d_{\hat{s}} - \mathbf{e}^\top \boldsymbol{u}_{\hat{s}})\right) + \left[ \left(\sigma(d_{\hat{s}} - \mathbf{e}^\top \boldsymbol{u}_{\hat{s}})\right)^2 + \left\| \begin{array}{c} \sigma \cdot \boldsymbol{Q}_{\hat{s}} \boldsymbol{u} - \overline{\boldsymbol{\theta}}' \\ \overline{\lambda}' \cdot \hat{\boldsymbol{p}} + \xi \cdot \mathbf{e} \end{array} \right\|_2^2 + \right. \\
& 2 \cdot \left| \sigma \cdot \boldsymbol{Q}_{\hat{s}} \boldsymbol{u} - \overline{\boldsymbol{\theta}}' \right|^\top \left. \left( \overline{\lambda}' \cdot \hat{\boldsymbol{p}} + \xi \cdot \mathbf{e} \right) \right]^{1/2}.
\end{aligned}
$$

*Proof of Proposition 11* Arbitrarily fix a feasible solution $(\overline{\lambda}, \overline{\boldsymbol{\theta}})$ of problem $\overline{K}_{\hat{s}}(\xi)$. By plugging $(\overline{\lambda}, \overline{\boldsymbol{\theta}})$ in the objective function of problem $\overline{K}_{\hat{s}}(\xi)$, we obtain an upper bound of its optimal value:

$$
\begin{aligned}
& \overline{\lambda}(d_{\hat{s}} - \mathbf{e}^\top \boldsymbol{u}_{\hat{s}}) + \overline{\boldsymbol{\theta}}^\top \boldsymbol{Q}_{\hat{s}} \boldsymbol{u} + \tfrac{1}{2\sigma} \left( (\overline{\lambda} - \overline{\lambda}')^2 + \|\overline{\boldsymbol{\theta}} - \overline{\boldsymbol{\theta}}'\|_2^2 \right) \\
\leq \ & \overline{\lambda}(d_{\hat{s}} - \mathbf{e}^\top \boldsymbol{u}_{\hat{s}}) + \tfrac{1}{2\sigma} \left( \left(\overline{\lambda} - \overline{\lambda}'\right)^2 + \left\|\overline{\lambda} \cdot \hat{\boldsymbol{p}} + \xi \cdot \mathbf{e}\right\|_2^2 + \overline{\boldsymbol{\theta}}'^\top \overline{\boldsymbol{\theta}}' \right. \\
& + \left| 2\sigma \cdot \boldsymbol{Q}_{\hat{s}} \boldsymbol{u} - 2 \cdot \overline{\boldsymbol{\theta}}' \right|^\top \left. \left( \overline{\lambda} \cdot \hat{\boldsymbol{p}} + \xi \cdot \mathbf{e} \right) \right).
\end{aligned}
\tag{29}
$$

The inequality here follows from the fact that $\left[\overline{\lambda} \cdot \hat{\boldsymbol{p}} - \xi \cdot \mathbf{e}\right]_+ \leq \overline{\boldsymbol{\theta}} \leq \overline{\lambda} \cdot \hat{\boldsymbol{p}} + \xi \cdot \mathbf{e}$ and Lemma 4. Let $(\overline{\lambda}^\star, \overline{\boldsymbol{\theta}}^\star)$ be an optimal solution of problem $\overline{K}_{\hat{s}}(\xi)$, and let $\overline{\lambda}^\star = \overline{\lambda} + \Delta\overline{\lambda}$. The problem

$$
\begin{aligned}
\min \quad & (\overline{\lambda} + \Delta\overline{\lambda})(d_{\hat{s}} - \mathbf{e}^\top \boldsymbol{u}_{\hat{s}}) + \boldsymbol{\theta}^\top \boldsymbol{Q}_{\hat{s}} \boldsymbol{u} + \frac{1}{2\sigma} \left( \left(\overline{\lambda} + \Delta\overline{\lambda} - \overline{\lambda}'\right)^2 + \|\boldsymbol{\theta} - \overline{\boldsymbol{\theta}}'\|_2^2 \right) \\
\text{s.t.} \quad & \theta_{s,a,s'} \geq (\overline{\lambda} + \Delta\overline{\lambda})\, \hat{p}_{s,a,s'} - \xi && \forall s \in \mathcal{S},\ a \in \mathcal{A},\ s' \in \mathcal{S} \\
& \theta_{s,a,s'} \leq (\overline{\lambda} + \Delta\overline{\lambda})\, \hat{p}_{s,a,s'} + \xi && \forall s \in \mathcal{S},\ a \in \mathcal{A},\ s' \in \mathcal{S} \\
& \mathbf{e}^\top \boldsymbol{\theta}_{s,a} = \overline{\lambda} + \Delta\overline{\lambda} && \forall s \in \mathcal{S},\ a \in \mathcal{A} \\
& \boldsymbol{\theta} \in \mathbb{R}_+^{S \cdot A \cdot S}
\end{aligned}
\tag{30}
$$

and problem $\overline{K}_{\hat{s}}(\xi)$ then share an equal optimal value. Taking the dual of (30), we have

$$
\begin{aligned}
\max_{\substack{\boldsymbol{\chi} \in \mathbb{R}_+^{S \cdot A \cdot S},\ \boldsymbol{\psi} \in \mathbb{R}_+^{S \cdot A \cdot S}, \\ \boldsymbol{\varrho} \in \mathbb{R}^{S \cdot A}, \\ \boldsymbol{\mu} \in \mathbb{R}_+^{S \cdot A \cdot S}}} \min_{\boldsymbol{\theta} \in \mathbb{R}_+^{S \cdot A \cdot S}} \quad & (\overline{\lambda} + \Delta\overline{\lambda})(d_{\hat{s}} - \mathbf{e}^\top \boldsymbol{u}_{\hat{s}}) + \boldsymbol{\theta}^\top \boldsymbol{Q}_{\hat{s}} \boldsymbol{u} + \frac{1}{2\sigma} \left( \left(\overline{\lambda} + \Delta\overline{\lambda} - \overline{\lambda}'\right)^2 + \|\boldsymbol{\theta} - \overline{\boldsymbol{\theta}}'\|_2^2 \right) \\
& + \sum_{(s,a,s') \in \mathcal{S} \times \mathcal{A} \times \mathcal{S}} \chi_{s,a,s'} \cdot \left( (\overline{\lambda} + \Delta\overline{\lambda})\, \hat{p}_{s,a,s'} - \xi - \theta_{s,a,s'} \right) \\
& - \sum_{(s,a,s') \in \mathcal{S} \times \mathcal{A} \times \mathcal{S}} \psi_{s,a,s'} \cdot \left( (\overline{\lambda} + \Delta\overline{\lambda})\, \hat{p}_{s,a,s'} + \xi - \theta_{s,a,s'} \right) \\
& + \sum_{(s,a) \in \mathcal{S} \times \mathcal{A}} \varrho_{s,a} \cdot \left( \mathbf{e}^\top \boldsymbol{\theta}_{s,a} - \overline{\lambda} - \Delta\overline{\lambda} \right) - \boldsymbol{\mu}^\top \boldsymbol{\theta},
\end{aligned}
$$

or equivalently,

$$
\begin{aligned}
\max \quad & (d_{\hat{s}} - \mathbf{e}^\top \boldsymbol{u}_{\hat{s}}) \cdot (\overline{\lambda} + \Delta\overline{\lambda}) + \frac{1}{2\sigma} \cdot \left( \left(\overline{\lambda} + \Delta\overline{\lambda} - \overline{\lambda}'\right)^2 + \overline{\boldsymbol{\theta}}'^\top \overline{\boldsymbol{\theta}}' \right) \\
& + \sum_{(s,a,s') \in \mathcal{S} \cdot \mathcal{A} \cdot \mathcal{S}} \chi_{s,a,s'} \cdot \left( (\overline{\lambda} + \Delta\overline{\lambda})\, \hat{p}_{s,a,s'} - \xi \right) \\
& - \sum_{(s,a,s') \in \mathcal{S} \times \mathcal{A} \times \mathcal{S}} \psi_{s,a,s'} \cdot \left( (\overline{\lambda} + \Delta\overline{\lambda})\, \hat{p}_{s,a,s'} + \xi \right) \\
& - \sum_{(s,a) \in \mathcal{S} \cdot \mathcal{A}} \varrho_{s,a} \cdot (\overline{\lambda} + \Delta\overline{\lambda}) - \frac{\sigma}{2} \left\| \boldsymbol{Q}_{\hat{s}} \boldsymbol{u} - \frac{1}{\sigma} \overline{\boldsymbol{\theta}}' - \boldsymbol{\chi} + \boldsymbol{\psi} + \boldsymbol{B}^\top \boldsymbol{\varrho} - \boldsymbol{\mu} \right\|_2^2 \\
\text{s.t.} \quad & \boldsymbol{\chi} \in \mathbb{R}_+^{S \cdot A \cdot S},\ \boldsymbol{\psi} \in \mathbb{R}_+^{S \cdot A \cdot S},\ \boldsymbol{\varrho} \in \mathbb{R}^{S \cdot A},\ \boldsymbol{\mu} \in \mathbb{R}_+^{S \cdot A \cdot S}.
\end{aligned}
\tag{31}
$$

By weak duality, the objective value of (31) achieved by the feasible solution $(\boldsymbol{\chi}, \boldsymbol{\psi}, \boldsymbol{\varrho}, \boldsymbol{\mu}) = \mathbf{0}$ provides a lower bound of the optimal value of problem $\overline{K}_{\hat{s}}(\xi)$ as

$$(d_{\hat{s}} - \mathbf{e}^\top \boldsymbol{u}_{\hat{s}}) \cdot (\overline{\lambda} + \Delta\overline{\lambda}) + \frac{1}{2\sigma} \cdot \left( \left( \overline{\lambda} + \Delta\overline{\lambda} - \overline{\lambda}' \right)^2 + \overline{\boldsymbol{\theta}}'^\top \overline{\boldsymbol{\theta}}' \right) - \frac{\sigma}{2} \cdot \left\| \boldsymbol{Q}_{\hat{s}}\boldsymbol{u} - \frac{1}{\sigma} \cdot \overline{\boldsymbol{\theta}}' \right\|_2^2. \quad (32)$$

Hence, by (29) and (32), it holds that:

$$(d_{\hat{s}} - \mathbf{e}^\top \boldsymbol{u}_{\hat{s}}) \cdot (\overline{\lambda} + \Delta\overline{\lambda}) + \frac{1}{2\sigma} \cdot \left( \left( \overline{\lambda} + \Delta\overline{\lambda} - \overline{\lambda}' \right)^2 + \overline{\boldsymbol{\theta}}'^\top \overline{\boldsymbol{\theta}}' \right) - \frac{\sigma}{2} \cdot \left\| \boldsymbol{Q}_{\hat{s}}\boldsymbol{u} - \frac{1}{\sigma} \cdot \overline{\boldsymbol{\theta}}' \right\|_2^2$$
$$\leq \quad \overline{\lambda}(d_{\hat{s}} - \mathbf{e}^\top \boldsymbol{u}_{\hat{s}}) + \frac{1}{2\sigma} \left( \left( \overline{\lambda} - \overline{\lambda}' \right)^2 + \left\| \overline{\lambda} \cdot \hat{\boldsymbol{p}} + \xi \cdot \mathbf{e} \right\|_2^2 + \overline{\boldsymbol{\theta}}'^\top \overline{\boldsymbol{\theta}}' + \left| 2\sigma \cdot \boldsymbol{Q}_{\hat{s}}\boldsymbol{u} - 2 \cdot \overline{\boldsymbol{\theta}}' \right|^\top \left( \overline{\lambda} \cdot \hat{\boldsymbol{p}} + \xi \cdot \mathbf{e} \right) \right),$$

which is equivalent to

$$(\Delta\overline{\lambda})^2 + 2 \left( \sigma(d_{\hat{s}} - \mathbf{e}^\top \boldsymbol{u}_{\hat{s}}) + \overline{\lambda} - \overline{\lambda}' \right)(\Delta\overline{\lambda}) - \left\| \begin{matrix} \sigma \cdot \boldsymbol{Q}_{\hat{s}}\boldsymbol{u} - \overline{\boldsymbol{\theta}}' \\ \overline{\lambda} \cdot \hat{\boldsymbol{p}} + \xi \cdot \mathbf{e} \end{matrix} \right\|_2^2 - 2 \cdot \left| \sigma \cdot \boldsymbol{Q}_{\hat{s}}\boldsymbol{u} - \overline{\boldsymbol{\theta}}' \right|^\top \left( \overline{\lambda} \cdot \hat{\boldsymbol{p}} + \xi \cdot \mathbf{e} \right) \leq 0,$$

yielding an upper bound

$$\Delta\overline{\lambda} \leq \quad - \left( \sigma(d_{\hat{s}} - \mathbf{e}^\top \boldsymbol{u}_{\hat{s}}) + \overline{\lambda} - \overline{\lambda}' \right) + \left[ \left( \sigma(d_{\hat{s}} - \mathbf{e}^\top \boldsymbol{u}_{\hat{s}}) + \overline{\lambda} - \overline{\lambda}' \right)^2 + \left\| \begin{matrix} \sigma \cdot \boldsymbol{Q}_{\hat{s}}\boldsymbol{u} - \overline{\boldsymbol{\theta}}' \\ \overline{\lambda} \cdot \hat{\boldsymbol{p}} + \xi \cdot \mathbf{e} \end{matrix} \right\|_2^2 + \right.$$
$$\left. 2 \cdot \left| \sigma \cdot \boldsymbol{Q}_{\hat{s}}\boldsymbol{u} - \overline{\boldsymbol{\theta}}' \right|^\top \left( \overline{\lambda} \cdot \hat{\boldsymbol{p}} + \xi \cdot \mathbf{e} \right) \right]^{1/2}.$$

This inequality further leads to

$$\overline{\lambda}^\star \leq \quad \overline{\lambda} - \left( \sigma(d_{\hat{s}} - \mathbf{e}^\top \boldsymbol{u}_{\hat{s}}) + \overline{\lambda} - \overline{\lambda}' \right) + \left[ \left( \sigma(d_{\hat{s}} - \mathbf{e}^\top \boldsymbol{u}_{\hat{s}}) + \overline{\lambda} - \overline{\lambda}' \right)^2 + \left\| \begin{matrix} \sigma \cdot \boldsymbol{Q}_{\hat{s}}\boldsymbol{u} - \overline{\boldsymbol{\theta}}' \\ \overline{\lambda} \cdot \hat{\boldsymbol{p}} + \xi \cdot \mathbf{e} \end{matrix} \right\|_2^2 + \right.$$
$$\left. 2 \cdot \left| \sigma \cdot \boldsymbol{Q}_{\hat{s}}\boldsymbol{u} - \overline{\boldsymbol{\theta}}' \right|^\top \left( \overline{\lambda} \cdot \hat{\boldsymbol{p}} + \xi \cdot \mathbf{e} \right) \right]^{1/2}.$$

Our conclusion then follows by taking $(\overline{\lambda}, \overline{\boldsymbol{\theta}}) = (\overline{\lambda}', \overline{\lambda}' \cdot \hat{\boldsymbol{p}})$ since they are taken arbitrarily. $\qquad\square$

**Lemma 5** *The optimal value of problem $\overline{f}_{\hat{s}}(\overline{\lambda})$ can be attained for any $\overline{\lambda} \in \mathbb{R}_+$.*

*Proof of Lemma 5* Arbitrarily fix $\overline{\lambda} \in \mathbb{R}_+$. The second collection of constraints in $\overline{f}_{\hat{s}}(\overline{\lambda})$, by definition of $\boldsymbol{B}$, allow the equivalence:

$$\overline{\lambda} \cdot \mathbf{e} = \boldsymbol{B}\overline{\boldsymbol{\theta}} \iff \mathbf{e}^\top \overline{\boldsymbol{\theta}}_{s,a} = \overline{\lambda} \quad \forall s \in \mathcal{S}, a \in \mathcal{A}.$$

This observation and the constraints $\overline{\boldsymbol{\theta}} \in \mathbb{R}_+^{S \cdot A \cdot S}$ together implies that problem $\overline{f}_{\hat{s}}(\overline{\lambda})$ has a bounded feasible region. Hence, problem $\overline{f}_{\hat{s}}(\overline{\lambda})$ has a continuous objective function and a non-empty, closed, and bounded feasible region, where $\overline{\boldsymbol{\theta}} = \overline{\lambda} \cdot \hat{\boldsymbol{p}}$ is a feasible solution. $\qquad\square$

*Proof of Proposition 7* The details of our tailored algorithm for solving (7) can be found in Appendix D. Problem (7) can be treated as $\min_{\xi \in [0, \phi_{\hat{s}}]} K_{\hat{s}}(\xi)$ that is solved via golden section search. We then decompose problem $K_{\hat{s}}(\xi)$ into problems $\overline{K}_{\hat{s}}(\xi)$ and $\underline{K}_{\hat{s}}(\xi)$ that share the same tailored algorithm. Problem $\overline{K}_{\hat{s}}(\xi)$ is equivalent to problem $\min_{\overline{\lambda} \in [0, \overline{\lambda}^{\text{up}}]} \overline{f}_{\hat{s}}(\overline{\lambda})$ that we again solve by golden section search. Problem $\overline{f}_{\hat{s}}(\overline{\lambda})$ is decomposed into $SA$ subproblems, and we solve each of them in time $\mathcal{O}(S \log S)$ by Algorithm 2. $\qquad\square$

*Proof of Proposition 8* Our conclusion follows immediately from Propositions 5, 6, and 7. $\qquad\square$

## D  TAILORED ALGORITHM FOR PROBLEM (7)

To solve problem (7), we can express it as a min-min problem $\min_{\xi \in [0, \phi_{\hat{s}}]} K_{\hat{s}}(\xi)$ that is solved by golden section search. Problem $K_{\hat{s}}(\xi)$ is decomposable into two subproblems $\overline{K}_{\hat{s}}(\xi)$ and $\underline{K}_{\hat{s}}(\xi)$ that share the exactly same tailored algorithm. To solve problem $\overline{K}_{\hat{s}}(\xi)$, we again use golden section search to solve its equivalent min-min problem $\min_{\overline{\lambda} \in [0, \overline{\lambda}^{\mathrm{up}}]} \overline{f}_{\hat{s}}(\overline{\lambda})$, where we provide the upper bound $\overline{\lambda}^{\mathrm{up}}$ for the search in Appendix C. Problem $\overline{f}_{\hat{s}}(\overline{\lambda})$ can be decomposed into $SA$ subproblems, and we solve each of them by our tailored algorithm that will be provided soon.

Problem (7) can be re-expressed as a min-min problem $\min_{\xi \in [0, \phi_{\hat{s}}]} K_{\hat{s}}(\xi)$ with

$$
\begin{aligned}
K_{\hat{s}}(\xi) = \min_{(\overline{\lambda}, \overline{\theta}) \in \mathcal{L}_{\infty}(\xi), (\underline{\lambda}, \underline{\theta}) \in \mathcal{L}_{\infty}(\phi_{\hat{s}} - \xi)} \quad & (\overline{\lambda} - \underline{\lambda})(d_{\hat{s}} - \mathbf{e}^{\top} \boldsymbol{u}_{\hat{s}}) + (\overline{\boldsymbol{\theta}} - \underline{\boldsymbol{\theta}})^{\top} \boldsymbol{Q}_{\hat{s}} \boldsymbol{u} + \frac{1}{2\sigma} \Big( (\xi - \xi')^2 + \\
& (\overline{\lambda} - \overline{\lambda}')^2 + (\underline{\lambda} - \underline{\lambda}')^2 + \|\overline{\boldsymbol{\theta}} - \overline{\boldsymbol{\theta}}'\|_2^2 + \|\underline{\boldsymbol{\theta}} - \underline{\boldsymbol{\theta}}'\|_2^2 \Big)
\end{aligned}
\tag{33}
$$

because $(\overline{\lambda}, \underline{\lambda}, \overline{\boldsymbol{\theta}}, \underline{\boldsymbol{\theta}}) = \mathbf{0}$ is a feasible solution for (33) for any $\xi \in [0, \phi_{\hat{s}}]$. Problem $\min_{\xi \in [0, \phi_{\hat{s}}]} K_{\hat{s}}(\xi)$ can be solved by golden section search (see, *e.g.*, Truhar & Veselić (2009)). The golden section search requires the function $K_{\hat{s}}(\xi)$ to be well-defined for any $\xi \in [0, \phi_{\hat{s}}]$. We prove this result in Proposition 10, and we relegate this result and its two preceding lemmas, Lemmas 2 and 3, to Appendix C. In general, the golden section search returns a suboptimal solution. Fortunately, it is an optimal solution due to the convexity of $K_{\hat{s}}(\xi)$ that we prove in the following lemma.

**Lemma 6** *Function $K_{\hat{s}}(\xi)$ is convex on $[0, \phi_{\hat{s}}]$.*

*Proof of Lemma 6* Arbitrarily fix $\xi, \xi' \in [0, \phi_{\hat{s}}]$. Let $(\overline{\lambda}, \underline{\lambda}, \overline{\boldsymbol{\theta}}, \underline{\boldsymbol{\theta}})$ and $(\overline{\lambda}', \underline{\lambda}', \overline{\boldsymbol{\theta}}', \underline{\boldsymbol{\theta}}')$ be the optimal solutions of problems $K_{\hat{s}}(\xi)$ and $K_{\hat{s}}(\xi')$, respectively. Let $\overline{K}_{\hat{s}}(\xi, \overline{\lambda}, \underline{\lambda}, \overline{\boldsymbol{\theta}}, \underline{\boldsymbol{\theta}})$ be the objective function of problem $K_{\hat{s}}(\xi)$, and $\mathcal{K}_{\hat{s}}(\xi)$ be its feasible region. The convexity of the feasible region of (6) implies $(1 - \omega) \cdot (\overline{\lambda}, \underline{\lambda}, \overline{\boldsymbol{\theta}}, \underline{\boldsymbol{\theta}}) + \omega \cdot (\overline{\lambda}', \underline{\lambda}', \overline{\boldsymbol{\theta}}', \underline{\boldsymbol{\theta}}') \in \mathcal{K}_{\hat{s}}((1 - \omega)\xi + \omega \xi')$ for any $\omega \in [0, 1]$, followed by which

$$
\begin{aligned}
K_{\hat{s}}((1 - \omega)\xi + \omega \xi') \leq \quad & \overline{K}_{\hat{s}} \Big( (1 - \omega) \cdot (\xi, \overline{\lambda}, \underline{\lambda}, \overline{\boldsymbol{\theta}}, \underline{\boldsymbol{\theta}}) + \omega \cdot (\xi', \overline{\lambda}', \underline{\lambda}', \overline{\boldsymbol{\theta}}', \underline{\boldsymbol{\theta}}') \Big) \\
\leq \quad & (1 - \omega) \cdot \overline{K}_{\hat{s}}(\xi, \overline{\lambda}, \underline{\lambda}, \overline{\boldsymbol{\theta}}, \underline{\boldsymbol{\theta}}) + \omega \cdot \overline{K}_{\hat{s}}(\xi', \overline{\lambda}', \underline{\lambda}', \overline{\boldsymbol{\theta}}', \underline{\boldsymbol{\theta}}') \\
= \quad & (1 - \omega) \cdot K_{\hat{s}}(\xi) + \omega \cdot K_{\hat{s}}(\xi').
\end{aligned}
$$

Here, the first inequality follows by the definition of $K_{\hat{s}}$, the second one is because $\overline{K}_{\hat{s}}$ is convex. $\square$

In each iteration of the above golden section search, problem (33) (with a different $\xi$) is solved. The efficiency of our dual update thus highly depends on the computation time of solving (33). To solve this problem, notice that for any fixed $\xi \in [0, \phi_{\hat{s}}]$, every constraint is exclusively for either $(\overline{\lambda}, \overline{\boldsymbol{\theta}})$ or $(\underline{\lambda}, \underline{\boldsymbol{\theta}})$, so is every term in the objective function. Therefore, we can decompose (33) into subproblems

$$
\overline{K}_{\hat{s}}(\xi) = \min_{(\overline{\lambda}, \overline{\boldsymbol{\theta}}) \in \mathcal{L}_{\infty}(\xi)} \overline{\lambda}(d_{\hat{s}} - \mathbf{e}^{\top} \boldsymbol{u}_{\hat{s}}) + \overline{\boldsymbol{\theta}}^{\top} \boldsymbol{Q}_{\hat{s}} \boldsymbol{u} + \frac{1}{2\sigma} \Big( (\overline{\lambda} - \overline{\lambda}')^2 + \|\overline{\boldsymbol{\theta}} - \overline{\boldsymbol{\theta}}'\|_2^2 \Big)
\tag{34}
$$

and

$$
\underline{K}_{\hat{s}}(\xi) = \min_{(\underline{\lambda}, \underline{\boldsymbol{\theta}}) \in \mathcal{L}_{\infty}(\phi_{\hat{s}} - \xi)} -\underline{\lambda}(d_{\hat{s}} - \mathbf{e}^{\top} \boldsymbol{u}_{\hat{s}}) - \underline{\boldsymbol{\theta}}^{\top} \boldsymbol{Q}_{\hat{s}} \boldsymbol{u} + \frac{1}{2\sigma} \Big( (\underline{\lambda} - \underline{\lambda}')^2 + \|\underline{\boldsymbol{\theta}} - \underline{\boldsymbol{\theta}}'\|_2^2 \Big).
\tag{35}
$$

The structures of these two problems are exactly the same. Therefore, we only introduce our tailored algorithm for (34), and we relegate the one for (35) to Appendix E. Here we remark that both problems (34) and (35) can attain their optimality because problem (33) can (by Proposition 10 in Appendix C), and they are the two subproblems of (33).

We consider two cases $\xi = 0$ and $\xi > 0$ for (34). In the former case, (34) reduces to a single-variable quadratic program that allows an analytical solution.

**Proposition 12** *For the optimal solution $(\overline{\lambda}^\star, \overline{\boldsymbol{\theta}}^\star)$ of problem $\overline{K}_{\hat{s}}(0)$, it holds that $\overline{\boldsymbol{\theta}}^\star = \overline{\lambda}^\star \cdot \hat{\boldsymbol{p}}$ and*

$$
\overline{\lambda}^\star = \begin{cases} -\frac{\sigma \cdot (d_{\hat{s}} - \mathbf{e}^\top \boldsymbol{u}_{\hat{s}} + \hat{\boldsymbol{p}}^\top \boldsymbol{Q}_{\hat{s}} \boldsymbol{u}) - \overline{\lambda}' - \hat{\boldsymbol{p}}^\top \overline{\boldsymbol{\theta}}'}{\hat{\boldsymbol{p}}^\top \hat{\boldsymbol{p}} + 1} & \text{if } \sigma \cdot (d_{\hat{s}} - \mathbf{e}^\top \boldsymbol{u}_{\hat{s}} + \hat{\boldsymbol{p}}^\top \boldsymbol{Q}_{\hat{s}} \boldsymbol{u}) - \overline{\lambda}' - \hat{\boldsymbol{p}}^\top \overline{\boldsymbol{\theta}}' \leq 0 \\ 0 & \text{otherwise.} \end{cases}
$$

*The optimal value of problem $\overline{K}_{\hat{s}}(0)$ is*

$$
\overline{K}_{\hat{s}}(0) = \begin{cases} \frac{1}{2\sigma} \cdot \left( \overline{\lambda}'^2 + \overline{\boldsymbol{\theta}}'^\top \overline{\boldsymbol{\theta}}' - \frac{\left( \sigma \cdot (d_{\hat{s}} - \mathbf{e}^\top \boldsymbol{u}_{\hat{s}} + \hat{\boldsymbol{p}}^\top \boldsymbol{Q}_{\hat{s}} \boldsymbol{u}) - \overline{\lambda}' - \hat{\boldsymbol{p}}^\top \overline{\boldsymbol{\theta}}' \right)^2}{\hat{\boldsymbol{p}}^\top \hat{\boldsymbol{p}} + 1} \right) & \text{if } \sigma \cdot (d_{\hat{s}} - \mathbf{e}^\top \boldsymbol{u}_{\hat{s}} + \hat{\boldsymbol{p}}^\top \boldsymbol{Q}_{\hat{s}} \boldsymbol{u}) \\ & \quad -(\overline{\lambda}' + \hat{\boldsymbol{p}}^\top \overline{\boldsymbol{\theta}}') \leq 0 \\ \frac{\overline{\lambda}'^2 + \overline{\boldsymbol{\theta}}'^\top \overline{\boldsymbol{\theta}}'}{2\sigma} & \text{otherwise.} \end{cases}
$$

*Proof of Proposition 12* The equality $\overline{\boldsymbol{\theta}}^\star = \overline{\lambda}^\star \cdot \hat{\boldsymbol{p}}$ follows immediately from the first constraint in problem $\overline{K}_{\hat{s}}(0)$. We then can reduce problem $\overline{K}_{\hat{s}}(0)$ to a single-variable quadratic program

$$
\min_{\overline{\lambda} \in \mathbb{R}_+} \frac{1}{2\sigma}(\hat{\boldsymbol{p}}^\top \hat{\boldsymbol{p}} + 1)\overline{\lambda}^2 + \left( d_{\hat{s}} - \mathbf{e}^\top \boldsymbol{u}_{\hat{s}} + \hat{\boldsymbol{p}}^\top \boldsymbol{Q}_{\hat{s}} \boldsymbol{u} - (1/\sigma) \left( \overline{\lambda}' + \hat{\boldsymbol{p}}^\top \overline{\boldsymbol{\theta}}' \right) \right) \overline{\lambda} + \frac{1}{2\sigma} \left( \overline{\lambda}'^2 + \overline{\boldsymbol{\theta}}'^\top \overline{\boldsymbol{\theta}}' \right),
$$

by substituting the equality $\overline{\boldsymbol{\theta}}^\star = \overline{\lambda}^\star \cdot \hat{\boldsymbol{p}}$ to the objective function of problem $\overline{K}_{\hat{s}}(0)$. $\quad\square$

For $\xi > 0$, note that (34) is expressable as an equivalent min-min problem $\min_{\overline{\lambda} \in \mathbb{R}_+} \overline{f}_{\hat{s}}(\overline{\lambda})$ where

$$
\overline{f}_{\hat{s}}(\overline{\lambda}) = \min_{\overline{\boldsymbol{\theta}} : (\overline{\lambda}, \overline{\boldsymbol{\theta}}) \in \mathcal{L}_\infty(\overline{\lambda})} \overline{\lambda}(d_{\hat{s}} - \mathbf{e}^\top \boldsymbol{u}_{\hat{s}}) + \overline{\boldsymbol{\theta}}^\top \boldsymbol{Q}_{\hat{s}} \boldsymbol{u} + \frac{1}{2\sigma} \left( (\overline{\lambda} - \overline{\lambda}')^2 + \|\overline{\boldsymbol{\theta}} - \overline{\boldsymbol{\theta}}'\|_2^2 \right). \tag{36}
$$

The equivalence here holds because $\overline{\boldsymbol{\theta}} = \overline{\lambda} \cdot \hat{\boldsymbol{p}}$ is a feasible solution of (36) for any $\overline{\lambda} \in \mathbb{R}_+$. We again apply the golden section search for locating the optimal $\overline{\lambda}^\star \in \mathbb{R}_+$ for the outer minimization problem for this min-min problem. Since problem (34) can attain its optimality as described above, it is then natural to find the upper and lower bounds for $\overline{\lambda}^\star$ and then conduct the search on the interval between these bounds. While 0 is a natural lower bound for the search, we will provide an upper bound $\overline{\lambda}^{\text{up}}$ in Proposition 11 in Appendix C. We also provide its preceding lemma, Lemma 4 in Appendix C. Similarly, to ensure that the golden section search can be conducted, we need the function $\overline{f}_{\hat{s}}(\overline{\lambda})$ to be well-defined (*i.e.*, problem $\overline{f}_{\hat{s}}(\overline{\lambda})$ can obtain its optimal value) for all $\overline{\lambda} \in [0, \overline{\lambda}^{\text{up}}]$. This is guaranteed by Lemma 5 in Appendix C. The golden section search here again returns a global optimal solution because $\overline{f}_{\hat{s}}(\cdot)$ is convex.

**Lemma 7** *The function $\overline{f}_{\hat{s}}(\cdot)$ is convex on $[0, \overline{\lambda}^{\text{up}}]$*

*Proof of Lemma 7* Let $\overline{\lambda}, \overline{\lambda}' \in [0, \overline{\lambda}^{\text{up}}]$ and $\kappa \in [0, 1]$ be arbitrarily fixed. Let $\overline{\boldsymbol{\theta}} \in \mathcal{D}_{\hat{s}}(\overline{\lambda})$ and $\overline{\boldsymbol{\theta}}' \in \mathcal{D}_{\hat{s}}(\overline{\lambda}')$ such that $h_{\hat{s}}(\overline{\lambda}, \overline{\boldsymbol{\theta}}) = \overline{f}_{\hat{s}}(\overline{\lambda})$ and $h_{\hat{s}}(\overline{\lambda}', \overline{\boldsymbol{\theta}}') = \overline{f}_{\hat{s}}(\overline{\lambda}')$. Here

$$
\mathcal{D}_{\hat{s}}(\overline{\lambda}) = \left\{ \overline{\boldsymbol{\theta}} \in \mathbb{R}_+^{S \cdot A \cdot S} \left| \begin{array}{ll} \|\overline{\boldsymbol{\theta}} - \overline{\lambda} \cdot \hat{\boldsymbol{p}}\|_\infty \leq \xi \\ \mathbf{e}^\top \overline{\boldsymbol{\theta}}_{s,a} = \overline{\lambda} & \forall (s, a) \in \mathcal{S} \times \mathcal{A} \end{array} \right. \right\}
$$

is the feasible region of problem $\overline{f}_{\hat{s}}(\overline{\lambda})$ and

$$
h_{\hat{s}}(\overline{\lambda}, \overline{\boldsymbol{\theta}}) = \overline{\lambda}(d_{\hat{s}} - \mathbf{e}^\top \boldsymbol{u}_{\hat{s}}) + \overline{\boldsymbol{\theta}}^\top \boldsymbol{Q}_{\hat{s}} \boldsymbol{u} + \frac{1}{2\sigma} \cdot \left\| \begin{array}{c} \overline{\lambda} - \overline{\lambda}' \\ \overline{\boldsymbol{\theta}} - \overline{\boldsymbol{\theta}}' \end{array} \right\|_2^2
$$

is its objective function. Note that the solution $((1 - \kappa) \cdot \overline{\lambda} + \kappa \overline{\lambda}', (1 - \kappa) \cdot \overline{\boldsymbol{\theta}} + \kappa \cdot \overline{\boldsymbol{\theta}}')$ is feasible to problem $\overline{K}_{\hat{s}}(\xi)$ because it has a convex feasible region. It then follows that $(1 - \kappa) \cdot \overline{\boldsymbol{\theta}} + \kappa \cdot \overline{\boldsymbol{\theta}}' \in \mathcal{D}_{\hat{s}}((1 - \kappa) \cdot \overline{\lambda} + \kappa \overline{\lambda}')$, and

$$
\begin{aligned} \overline{f}_{\hat{s}}((1 - \kappa) \cdot \overline{\lambda} + \kappa \overline{\lambda}') \leq & \ h_{\hat{s}}((1 - \kappa) \cdot \overline{\lambda} + \kappa \overline{\lambda}', (1 - \kappa) \cdot \overline{\boldsymbol{\theta}} + \kappa \cdot \overline{\boldsymbol{\theta}}') \\ \leq & \ (1 - \kappa) \cdot h_{\hat{s}}(\overline{\lambda}, \overline{\boldsymbol{\theta}}) + \kappa \cdot h_{\hat{s}}(\overline{\lambda}', \overline{\boldsymbol{\theta}}') \\ = & \ (1 - \kappa) \cdot \overline{f}_{\hat{s}}(\overline{\lambda}) + \kappa \cdot \overline{f}_{\hat{s}}(\overline{\lambda}'), \end{aligned}
$$

where the first inequality follows by the definition of the function $\overline{f}_{\hat{s}}$, the second inequality is because the function $h_{\hat{s}}$ is convex. Since $\overline{\lambda}$, $\overline{\lambda}'$ and $\kappa$ are all arbitrary, the convexity of $\overline{f}_{\hat{s}}$ is proved.
$\square$

Up to now, we have utilized the golden section search multiple times to address problem (6). The efficiency of the searches heavily relies on how to efficiently solve the subproblems encountered during each iteration, which ultimately hinges on the speed of our algorithm for solving problem (36). Observe that in this problem, for any $\overline{\lambda} \in \mathbb{R}_+$, each constraint is exclusively related to only one of the decision variables among $\{\boldsymbol{\theta}_{s,a}\}_{(s,a)\in\mathcal{S}\times\mathcal{A}}$, so is each term among the $SA$ terms in the objective function $\sum_{s\in\mathcal{S}}\sum_{a\in\mathcal{A}}\left\{\frac{1}{2\sigma}\cdot\overline{\boldsymbol{\theta}}_{s,a}^{\top}\overline{\boldsymbol{\theta}}_{s,a} + \overline{\boldsymbol{\theta}}_{s,a}^{\top}\left(\boldsymbol{x}_{\hat{s},s,a} - \frac{1}{\sigma}\cdot\overline{\boldsymbol{\theta}}'_{s,a}\right)\right\}$ with $\boldsymbol{x}_{\hat{s}} = \boldsymbol{Q}_{\hat{s}}\boldsymbol{u} \in \mathbb{R}^{S\cdot A\cdot S}$. Therefore, we can decompose problem (36) into $SA$ subproblems, and the $(s,a)$-th one is

$$\min_{\substack{\overline{\boldsymbol{\theta}}\in\mathbb{R}^{S}:\\ \mathbf{e}^{\top}\overline{\boldsymbol{\theta}}=\overline{\lambda}}} \frac{1}{2\sigma}\cdot\overline{\boldsymbol{\theta}}^{\top}\overline{\boldsymbol{\theta}} + \overline{\boldsymbol{\theta}}^{\top}\left(\boldsymbol{x}_{\hat{s},s,a} - \frac{1}{\sigma}\cdot\overline{\boldsymbol{\theta}}'_{s,a}\right) : [\overline{\lambda}\hat{p}_{s,a,s'} - \xi]_+ \leq \overline{\theta}_{s'} \leq \overline{\lambda}\hat{p}_{s,a,s'} + \xi \quad \forall s' \in \mathcal{S} \quad (37)$$

for all $(s,a) \in \mathcal{S}\times\mathcal{A}$. Observe that this is a quadratic program with no cross term in the objective function, and with only one linear constraint (in addition to some box constraints where we specify lower and upper bounds for decision variables). By exploring the KKT conditions, we design a tailored algorithm via which we reduce problem (37) to computing the root of a non-decreasing piecewise linear function, solvable in time $\mathcal{O}(S\log(S))$.

**Proposition 13** *Problem* (37) *is solvable in time* $\mathcal{O}(S\log S)$.

*Proof of Proposition 13.* Let $\boldsymbol{\eta}$, $\boldsymbol{\varphi} \in \mathbb{R}_+^S$ and $\rho \in \mathbb{R}$ be the dual variables of problem (37). We then can express the Lagrangian function of problem (37) as follows:

$$L(\overline{\boldsymbol{\theta}}_{s,a},\boldsymbol{\eta},\boldsymbol{\varphi},\rho) = \frac{1}{2\sigma}\cdot\overline{\boldsymbol{\theta}}_{s,a}^{\top}\overline{\boldsymbol{\theta}}_{s,a} + \overline{\boldsymbol{\theta}}_{s,a}^{\top}\left(\boldsymbol{x}_{\hat{s},s,a} - \frac{1}{\sigma}\cdot\overline{\boldsymbol{\theta}}'_{s,a}\right) + \sum_{s'\in\mathcal{S}}\eta_{s'}\cdot([\overline{\lambda}\hat{p}_{s,a,s'} - \xi]_+ - \overline{\theta}_{s,a,s'})$$

$$+ \sum_{s'\in\mathcal{S}}\varphi_{s'}\cdot(\overline{\theta}_{s,a,s'} - (\overline{\lambda}\hat{p}_{s,a,s'} + \xi)) + \rho\cdot(\overline{\lambda} - \mathbf{e}^{\top}\overline{\boldsymbol{\theta}}_{s,a}).$$

For the convex optimization problem (37), KKT conditions are sufficient and necessary for its optimality:

$$\begin{cases} \overline{\theta}_{s,a,s'} \geq [\overline{\lambda}\hat{p}_{s,a,s'} - \xi]_+ & \forall s' \in \mathcal{S} \\ \overline{\theta}_{s,a,s'} \leq \overline{\lambda}\hat{p}_{s,a,s'} + \xi & \forall s' \in \mathcal{S} \\ \mathbf{e}^{\top}\overline{\boldsymbol{\theta}}_{s,a} = \overline{\lambda} \\ \boldsymbol{\eta} \geq \mathbf{0} \\ \boldsymbol{\varphi} \geq \mathbf{0} \\ \eta_{s'}\cdot([\overline{\lambda}\hat{p}_{s,a,s'} - \xi]_+ - \overline{\theta}_{s,a,s'}) = 0 & \forall s' \in \mathcal{S} \\ \varphi_{s'}\cdot(\overline{\theta}_{s,a,s'} - (\overline{\lambda}\hat{p}_{s,a,s'} + \xi)) = 0 & \forall s' \in \mathcal{S} \\ \nabla_{\overline{\boldsymbol{\theta}}_{s,a}}L(\overline{\boldsymbol{\theta}}_{s,a},\boldsymbol{\eta},\boldsymbol{\varphi},\rho) = \frac{1}{\sigma}\cdot\overline{\boldsymbol{\theta}}_{s,a} + \left(\boldsymbol{x}_{\hat{s},s,a} - \frac{1}{\sigma}\cdot\overline{\boldsymbol{\theta}}'_{s,a}\right) - \boldsymbol{\eta} + \boldsymbol{\varphi} - \rho\cdot\mathbf{e} = \mathbf{0}, \end{cases}$$

where

$$\overline{\theta}_{s,a,s'} = \begin{cases} \overline{\lambda}\hat{p}_{s,a,s'} + \xi & \forall s' \in \mathcal{S} : \varphi_{s'} \neq 0 \\ \sigma\cdot\left(\rho + \frac{1}{\sigma}\overline{\theta}'_{s,a,s'} - x_{\hat{s},s,a,s'}\right) & \forall s' \in \mathcal{S} : \eta_{s'} = 0 \text{ and } \varphi_{s'} = 0 \\ [\overline{\lambda}\hat{p}_{s,a,s'} - \xi]_+ & \forall s' \in \mathcal{S} : \eta_{s'} \neq 0 \end{cases}$$

---

**Algorithm 2** Interval-Searching Algorithm for Problem (37)

---

Compute all the upper breakpoints $\overline{\rho}_{s'} \leftarrow \frac{1}{\sigma}(\overline{\lambda}\hat{p}_{s,a,s'} + \xi) - \frac{1}{\sigma}\overline{\theta}'_{s,a,s'} + x_{\hat{s},s,a,s'}$ $\forall s' \in \mathcal{S}$ and lower

breakpoints $\underline{\rho}_{s'} \leftarrow \frac{1}{\sigma}[\overline{\lambda}\hat{p}_{s,a,s'} - \xi]_+ - \frac{1}{\sigma}\overline{\theta}'_{s,a,s'} + x_{\hat{s},s,a,s'}$ $\forall s' \in \mathcal{S}$

Sort the breakpoints in an ascending order as $\rho_1 \leq \cdots \leq \rho_{2S}$

Initialize $\chi \leftarrow \sigma$ and $\psi \leftarrow \sum_{s' \in \mathcal{S}: s' \neq p_1(1)}[\overline{\lambda}\hat{p}_{s,a,s'} - \xi]_+ + \sigma \cdot (\frac{1}{\sigma}\overline{\theta}'_{s,a,p_1(1)} - x_{\hat{s},s,a,p_1(1)})$

Initialize the index set for the upper breakpoints $\mathcal{U} \leftarrow \emptyset$ and the one for the lower breakpoints
$\mathcal{L} \leftarrow \mathcal{S} \setminus p_1(1)$

**for** $k = 1, \cdots, 2S - 1$ **do**

  **if** $\chi \cdot \rho_{k+1} + \psi \geq \overline{\lambda}$ **then**

    $\rho^\star \leftarrow \frac{\overline{\lambda} - \psi}{\chi}$

    **for** $s' = 1, \cdots, S$ **do**

$$
\overline{\theta}^\star_{s,a,s'} \leftarrow
\begin{cases}
\overline{\lambda}\hat{p}_{s,a,s'} + \xi & \forall s' \in \mathcal{U} \\
[\overline{\lambda}\hat{p}_{s,a,s'} - \xi]_+ & \forall s' \in \mathcal{L} \\
\sigma \cdot (\rho^\star + \frac{1}{\sigma}\overline{\theta}'_{s,a,s'} - x_{\hat{s},s,a,s'}) & \forall s' \in \mathcal{S} \setminus (\mathcal{U} \cup \mathcal{L});
\end{cases}
$$

    **end for**

  **else if** $p_2(k+1) = $ "upper" **then**

    $\chi \leftarrow \chi - \sigma$

    $\psi \leftarrow \psi - \sigma \cdot (\frac{1}{\sigma}\overline{\theta}'_{s,a,p_1(k+1)} - x_{\hat{s},s,a,p_1(k+1)}) + \overline{\lambda}\hat{p}_{s,a,p_1(k+1)} + \xi$

  **else**

    $\chi \leftarrow \chi + \sigma$

    $\psi \leftarrow \psi + \sigma \cdot (\frac{1}{\sigma}\overline{\theta}'_{s,a,p_1(k+1)} - x_{\hat{s},s,a,p_1(k+1)}) - [\overline{\lambda}\hat{p}_{s,a,p_1(k+1)} - \xi]_+$

  **end if**

**end for**

**Output:** Solution $\overline{\theta}^\star_{s,a}$

---

follows. It then suffices to solve for the optimal solution $\rho^\star$ of the equation $H_{s,a}(\rho) = \overline{\lambda}$, after which we can have $\overline{\theta}^\star_{s,a,s'} = H_{s,a,s'}(\rho^\star)$ $\forall s \in \mathcal{S}$, where $H_{s,a}(\rho) = \sum_{s' \in \mathcal{S}} H_{s,a,s'}(\rho)$ and

$$
H_{s,a,s'}(\rho) =
\begin{cases}
\overline{\lambda}\hat{p}_{s,a,s'} + \xi & \text{if } \rho \geq \frac{1}{\sigma} \cdot (\overline{\lambda}\hat{p}_{s,a,s'} + \xi) + x_{\hat{s},s,a,s'} - \frac{1}{\sigma}\overline{\theta}'_{s,a,s'} \\
[\overline{\lambda}\hat{p}_{s,a,s'} - \xi]_+ & \text{if } \rho < \frac{1}{\sigma} \cdot [\overline{\lambda}\hat{p}_{s,a,s'} - \xi]_+ + x_{\hat{s},s,a,s'} - \frac{1}{\sigma}\overline{\theta}'_{s,a,s'} \\
\sigma \cdot \left(\rho + \frac{1}{\sigma}\overline{\theta}'_{s,a,s'} - x_{\hat{s},s,a,s'}\right) & \text{otherwise}
\end{cases}
$$

for all $s' \in \mathcal{S}$. As the sum of $S$ piecewise linear and non-decreasing functions, the function $H_{s,a} = \sum_{s' \in \mathcal{S}} H_{s,a,s'}$ is also piecewise linear and non-decreasing, who has $2S$ break-points: $\frac{1}{\sigma} \cdot (\overline{\lambda}\hat{p}_{s,a,s'} + \xi) - \frac{1}{\sigma}\overline{\theta}'_{s,a,s'} + x_{\hat{s},s,a,s'}$, $s' \in \mathcal{S}$ (that we call "upper breakpoints") and $\frac{1}{\sigma} \cdot [\overline{\lambda}\hat{p}_{s,a,s'} - \xi]_+ - \frac{1}{\sigma}\overline{\theta}'_{s,a,s'} + x_{\hat{s},s,a,s'}$, $s' \in \mathcal{S}$ (that we call "lower breakpoints"). After sorting $2S$ breakpoints in an ascending order $[\rho_1, \rho_2], [\rho_2, \rho_3], \ldots, [\rho_{2S-1}, \rho_{2S}]$, we can sequentially search the intervals $[\rho_1, \rho_2], [\rho_2, \rho_3], \ldots, [\rho_{2S-1}, \rho_{2S}]$, obtain the optimal $\rho^\star$, and finally obtain $\overline{\theta}^\star_{s,a,s'} = H_{s,a,s'}(\rho^\star)$ $\forall s' \in \mathcal{S}$.

The time complexity of the above process is $\mathcal{O}(S \log S)$ required by sorting the breakpoints. $\qquad\square$

The pseudocode for the tailored algorithm for problem (37) as described in the proof of Proposition 13 is provided in Algorithm 2. In the pseudocode, the functions $p_1(\cdot) : [2S] \mapsto \mathcal{S}$ and $p_2(\cdot) : [2S] \mapsto \{$ "lower", "upper" $\}$ map the indices of the non-decreasing breakpoint sequence to the indices and types of lower/upper breakpoints, respectively; *e.g.*, if $\rho_4$ corresponds to $\overline{\rho}_6$, then we have $p_1(4) = 6$ and $p_2(4) = $ "upper".

## E  TAILORED ALGORITHM FOR SUBPROBLEM IN DUAL UPDATE

We consider two cases $\xi = \phi_{\hat{s}}$ and $\xi < \phi_{\hat{s}}$ for (35). In the former case, (35) reduces to a single-variable quadratic program that allows an analytical solution.

**Proposition 14** *Let* $(\underline{\lambda}^{\star}, \underline{\theta}^{\star})$ *denote the optimal solution of problem* $\underline{K}_{\hat{s}}(\phi_{\hat{s}})$. *It holds that* $\underline{\theta}^{\star} = \underline{\lambda}^{\star} \cdot \hat{p}$,

$$\underline{\lambda}^{\star} = \begin{cases} -\frac{\sigma \cdot (-d_{\hat{s}} + \mathbf{e}^{\top} \boldsymbol{u}_{\hat{s}} - \hat{\boldsymbol{p}}^{\top} \boldsymbol{Q}_{\hat{s}} \boldsymbol{u}) - \underline{\lambda}' - \hat{\boldsymbol{p}}^{\top} \underline{\boldsymbol{\theta}}'}{\hat{\boldsymbol{p}}^{\top} \hat{\boldsymbol{p}} + 1} & \text{if } \sigma \cdot (-d_{\hat{s}} + \mathbf{e}^{\top} \boldsymbol{u}_{\hat{s}} - \hat{\boldsymbol{p}}^{\top} \boldsymbol{Q}_{\hat{s}} \boldsymbol{u}) - \underline{\lambda}' - \hat{\boldsymbol{p}}^{\top} \underline{\boldsymbol{\theta}}' \leq 0 \\ 0 & \text{otherwise.} \end{cases}$$

*The optimal value of problem* $\underline{K}_{\hat{s}}(\phi_{\hat{s}})$ *is as follows:*

$$\underline{K}_{\hat{s}}(\phi_{\hat{s}}) = \begin{cases} \frac{1}{2\sigma} \cdot \left( \underline{\lambda}'^2 + \underline{\boldsymbol{\theta}}'^{\top} \underline{\boldsymbol{\theta}}' - \frac{(\sigma \cdot (-d_{\hat{s}} + \mathbf{e}^{\top} \boldsymbol{u}_{\hat{s}} - \hat{\boldsymbol{p}}^{\top} \boldsymbol{Q}_{\hat{s}} \boldsymbol{u}) - \underline{\lambda}' - \hat{\boldsymbol{p}}^{\top} \underline{\boldsymbol{\theta}}')^2}{\hat{\boldsymbol{p}}^{\top} \hat{\boldsymbol{p}} + 1} \right) & \text{if } \sigma \cdot (-d_{\hat{s}} + \mathbf{e}^{\top} \boldsymbol{u}_{\hat{s}} - \hat{\boldsymbol{p}}^{\top} \boldsymbol{Q}_{\hat{s}} \boldsymbol{u}) \\ & \quad -(\underline{\lambda}' + \hat{\boldsymbol{p}}^{\top} \underline{\boldsymbol{\theta}}') \leq 0 \\ \frac{\underline{\lambda}'^2 + \underline{\boldsymbol{\theta}}'^{\top} \underline{\boldsymbol{\theta}}'}{2\sigma} & \text{otherwise.} \end{cases}$$

*Proof of Proposition 14*  The equality $\underline{\boldsymbol{\theta}}^{\star} = \underline{\lambda}^{\star} \cdot \hat{\boldsymbol{p}}$ can be realized by looking at the first constraint in problem $\underline{K}_{\hat{s}}(\phi_{\hat{s}})$. Substituting this equality to the objective function of problem $\underline{K}_{\hat{s}}(\phi_{\hat{s}})$, we then reduce this problem to a single-variable quadratic program as follows:

$$\min_{\underline{\lambda} \in \mathbb{R}_+} \frac{1}{2\sigma} (\hat{\boldsymbol{p}}^{\top} \hat{\boldsymbol{p}} + 1) \underline{\lambda}^2 + \left( -d_{\hat{s}} + \mathbf{e}^{\top} \boldsymbol{u}_{\hat{s}} - \hat{\boldsymbol{p}}^{\top} \boldsymbol{Q}_{\hat{s}} \boldsymbol{u} - (1/\sigma) \left( \underline{\lambda}' + \hat{\boldsymbol{p}}^{\top} \underline{\boldsymbol{\theta}}' \right) \right) \underline{\lambda} + \frac{1}{2\sigma} \left( \underline{\lambda}'^2 + \underline{\boldsymbol{\theta}}'^{\top} \underline{\boldsymbol{\theta}}' \right),$$

where our conclusion follows immediately. $\qquad \square$

In the case $\underline{\xi} < \phi_{\hat{s}}$, problem $\underline{K}_{\hat{s}}(\xi)$ is treated as a min-min problem $\min_{\underline{\lambda} \in \mathbb{R}_+} \underline{f}_{\hat{s}}(\underline{\lambda})$ with

$$\underline{f}_{\hat{s}}(\underline{\lambda}) = \quad \min \quad -\underline{\lambda}(d_{\hat{s}} - \mathbf{e}^{\top} \boldsymbol{u}_{\hat{s}}) - \underline{\boldsymbol{\theta}}^{\top} \boldsymbol{Q}_{\hat{s}} \boldsymbol{u} + \frac{1}{2\sigma} \left( (\underline{\lambda} - \underline{\lambda}')^2 + \|\underline{\boldsymbol{\theta}} - \underline{\boldsymbol{\theta}}'\|_2^2 \right)$$

$$\text{s.t.} \quad \|\underline{\boldsymbol{\theta}} - \underline{\lambda} \cdot \hat{\boldsymbol{p}}\|_{\infty} \leq \phi_{\hat{s}} - \xi$$

$$\underline{\lambda} \cdot \mathbf{e} = \boldsymbol{B} \underline{\boldsymbol{\theta}}$$

$$\underline{\boldsymbol{\theta}} \in \mathbb{R}_+^{S \cdot A \cdot S}.$$

This equivalence is allowed because $\underline{\boldsymbol{\theta}} = \underline{\lambda} \cdot \hat{\boldsymbol{p}}$ is a feasible solution to problem $\underline{f}_{\hat{s}}(\underline{\lambda})$ for any $\underline{\lambda} \in \mathbb{R}_+$. As we note in Appendix D, problem (33) can attain its optimal value by Proposition 10. Therefore, its subproblem $\underline{K}_{\hat{s}}(\xi)$ can also attain its optimal value. For the problem $\min_{\underline{\lambda} \in \mathbb{R}_+} \underline{f}_{\hat{s}}(\underline{\lambda})$, we compute the optimal $\underline{\lambda}^{\star} \in \mathbb{R}$ for its outer minimization problem via golden section search on the interval $[0, \underline{\lambda}^{\text{up}}]$. A choice for $\underline{\lambda}^{\text{up}}$ is provided as in the following lemma.

**Lemma 8** *Let* $(\underline{\lambda}^{\star}, \underline{\boldsymbol{\theta}}^{\star})$ *be the optimal solution to problem* $\underline{K}_{\hat{s}}(\xi)$, *where* $\xi < \phi_{\hat{s}}$. *It holds that*

$$\underline{\lambda}^{\star} \leq \quad \underline{\lambda}' - \left( \sigma(-d_{\hat{s}} + \mathbf{e}^{\top} \boldsymbol{u}_{\hat{s}}) \right) + \left[ \left( \sigma(-d_{\hat{s}} + \mathbf{e}^{\top} \boldsymbol{u}_{\hat{s}}) \right)^2 + \left\| \begin{array}{c} \sigma \cdot \boldsymbol{Q}_{\hat{s}} \boldsymbol{u} + \underline{\boldsymbol{\theta}}' \\ \underline{\lambda}' \cdot \hat{\boldsymbol{p}} + (\phi_{\hat{s}} - \xi) \cdot \mathbf{e} \end{array} \right\|_2^2 + \right.$$

$$\left. 2 \cdot \left| \sigma \cdot \boldsymbol{Q}_{\hat{s}} \boldsymbol{u} + \underline{\boldsymbol{\theta}}' \right|^{\top} \left( \underline{\lambda}' \cdot \hat{\boldsymbol{p}} + (\phi_{\hat{s}} - \xi) \cdot \mathbf{e} \right) \right]^{1/2}.$$

*Proof of Lemma 8*  Arbitrarily take a feasible solution $(\underline{\lambda}, \underline{\boldsymbol{\theta}})$ of problem $\underline{K}_{\hat{s}}(\xi)$. By plugging it into the objective function of problem $\underline{K}_{\hat{s}}(\xi)$, we obtain an upper bound of the optimal value of this problem as follows:

$$\underline{\lambda}(-d_{\hat{s}} + \mathbf{e}^{\top} \boldsymbol{u}_{\hat{s}}) - \underline{\boldsymbol{\theta}}^{\top} \boldsymbol{Q}_{\hat{s}} \boldsymbol{u} + \frac{1}{2\sigma} \left( (\underline{\lambda} - \underline{\lambda}')^2 + \|\underline{\boldsymbol{\theta}} - \underline{\boldsymbol{\theta}}'\|_2^2 \right)$$

$$\leq \quad \underline{\lambda}(-d_{\hat{s}} + \mathbf{e}^{\top} \boldsymbol{u}_{\hat{s}}) + \frac{1}{2\sigma} \cdot \left( \left( \underline{\lambda} - \underline{\lambda}' \right)^2 + \|\underline{\lambda} \cdot \hat{\boldsymbol{p}} + (\phi_{\hat{s}} - \xi) \cdot \mathbf{e}\|_2^2 + \underline{\boldsymbol{\theta}}'^{\top} \underline{\boldsymbol{\theta}}' \right. \tag{38}$$

$$\left. +2 \cdot \left| \sigma \cdot \boldsymbol{Q}_{\hat{s}} \boldsymbol{u} + \underline{\boldsymbol{\theta}}' \right|^{\top} (\underline{\lambda} \cdot \hat{\boldsymbol{p}} + (\phi_{\hat{s}} - \xi) \cdot \mathbf{e}) \right).$$

The inequality here holds by Lemma 4 and the fact that $[\underline{\lambda} \cdot \hat{p} - (\phi_{\hat{s}} - \xi) \cdot \mathbf{e}]_+ \le \underline{\theta} \le \underline{\lambda} \cdot \hat{p} + (\phi_{\hat{s}} - \xi) \cdot \mathbf{e}$.

Let $(\underline{\lambda}^\star, \underline{\theta}^\star)$ be the optimal solution to problem $\underline{K}_{\hat{s}}(\xi)$, and let $\underline{\lambda}^\star = \underline{\lambda} + \Delta\underline{\lambda}$. Then problem

$$
\begin{aligned}
\min \quad & (\underline{\lambda} + \Delta\underline{\lambda})\left(-d_{\hat{s}} + \mathbf{e}^\top \boldsymbol{u}_{\hat{s}}\right) - \boldsymbol{\theta}^\top \boldsymbol{Q}_{\hat{s}} \boldsymbol{u} + \frac{1}{2\sigma}\left(\left(\underline{\lambda} + \Delta\underline{\lambda} - \underline{\lambda}'\right)^2 + \|\boldsymbol{\theta} - \underline{\theta}'\|_2^2\right) \\
\text{s.t.} \quad & \theta_{s,a,s'} \ge (\underline{\lambda} + \Delta\underline{\lambda})\,\hat{p}_{s,a,s'} - (\phi_{\hat{s}} - \xi) && \forall s \in \mathcal{S},\ a \in \mathcal{A},\ s' \in \mathcal{S} \\
& \theta_{s,a,s'} \le (\underline{\lambda} + \Delta\underline{\lambda})\,\hat{p}_{s,a,s'} + (\phi_{\hat{s}} - \xi) && \forall s \in \mathcal{S},\ a \in \mathcal{A},\ s' \in \mathcal{S} \\
& \mathbf{e}^\top \boldsymbol{\theta}_{s,a} = \underline{\lambda} + \Delta\underline{\lambda} && \forall s \in \mathcal{S},\ a \in \mathcal{A} \\
& \boldsymbol{\theta} \in \mathbb{R}_+^{S \cdot A \cdot S},
\end{aligned}
\tag{39}
$$

and problem $\underline{K}_{\hat{s}}(\xi)$ have an equal optimal value. Introducing dual variables $\boldsymbol{\chi} \in \mathbb{R}_+^{S \cdot A \cdot S}, \boldsymbol{\psi} \in \mathbb{R}_+^{S \cdot A \cdot S}, \boldsymbol{\varrho} \in \mathbb{R}^{S \cdot A}$ and $\boldsymbol{\mu} \in \mathbb{R}_+^{S \cdot A \cdot S}$, we take the dual of (39) as

$$
\begin{aligned}
\max \quad & (-d_{\hat{s}} + \mathbf{e}^\top \boldsymbol{u}_{\hat{s}}) \cdot (\underline{\lambda} + \Delta\underline{\lambda}) + \frac{1}{2\sigma} \cdot \left(\left(\underline{\lambda} + \Delta\underline{\lambda} - \underline{\lambda}'\right)^2 + \underline{\theta}'^\top \underline{\theta}'\right) \\
& + \sum_{(s,a,s') \in \mathcal{S} \cdot \mathcal{A} \cdot \mathcal{S}} \chi_{s,a,s'} \cdot ((\underline{\lambda} + \Delta\underline{\lambda})\,\hat{p}_{s,a,s'} - (\phi_{\hat{s}} - \xi)) \\
& - \sum_{(s,a,s') \in \mathcal{S} \times \mathcal{A} \times \mathcal{S}} \psi_{s,a,s'} \cdot ((\underline{\lambda} + \Delta\underline{\lambda})\,\hat{p}_{s,a,s'} + (\phi_{\hat{s}} - \xi)) \\
& - \sum_{(s,a) \in \mathcal{S} \cdot \mathcal{A}} \varrho_{s,a} \cdot (\underline{\lambda} + \Delta\underline{\lambda}) - \frac{\sigma}{2} \left\| -\boldsymbol{Q}_{\hat{s}} \boldsymbol{u} - \frac{1}{\sigma}\underline{\theta}' - \boldsymbol{\chi} + \boldsymbol{\psi} + \boldsymbol{B}^\top \boldsymbol{\varrho} - \boldsymbol{\mu} \right\|_2^2 \\
\text{s.t.} \quad & \boldsymbol{\chi} \in \mathbb{R}_+^{S \cdot A \cdot S},\ \boldsymbol{\psi} \in \mathbb{R}_+^{S \cdot A \cdot S},\ \boldsymbol{\varrho} \in \mathbb{R}^{S \cdot A},\ \boldsymbol{\mu} \in \mathbb{R}_+^{S \cdot A \cdot S}.
\end{aligned}
\tag{40}
$$

Consider a feasible solution $(\boldsymbol{\chi}, \boldsymbol{\psi}, \boldsymbol{\varrho}, \boldsymbol{\mu}) = \boldsymbol{0}$ to (40). By weak duality, it gives a lower bound of the optimal value of problem $\underline{K}_{\hat{s}}(\xi)$ as

$$
(-d_{\hat{s}} + \mathbf{e}^\top \boldsymbol{u}_{\hat{s}}) \cdot (\underline{\lambda} + \Delta\underline{\lambda}) + \frac{1}{2\sigma} \cdot \left(\left(\underline{\lambda} + \Delta\underline{\lambda} - \underline{\lambda}'\right)^2 + \underline{\theta}'^\top \underline{\theta}'\right) - \frac{\sigma}{2} \cdot \left\| -\boldsymbol{Q}_{\hat{s}} \boldsymbol{u} - \frac{1}{\sigma} \cdot \underline{\theta}' \right\|_2^2. \tag{41}
$$

We then obtain the following inequality by (38) and (41):

$$
\begin{aligned}
& (-d_{\hat{s}} + \mathbf{e}^\top \boldsymbol{u}_{\hat{s}}) \cdot (\underline{\lambda} + \Delta\underline{\lambda}) + \frac{1}{2\sigma} \cdot \left(\left(\underline{\lambda} + \Delta\underline{\lambda} - \underline{\lambda}'\right)^2 + \underline{\theta}'^\top \underline{\theta}'\right) - \frac{\sigma}{2} \cdot \left\| -\boldsymbol{Q}_{\hat{s}} \boldsymbol{u} - \frac{1}{\sigma} \cdot \underline{\theta}' \right\|_2^2 \\
\le \quad & \underline{\lambda}(-d_{\hat{s}} + \mathbf{e}^\top \boldsymbol{u}_{\hat{s}}) + \frac{1}{2\sigma} \cdot \left(\left(\underline{\lambda} - \underline{\lambda}'\right)^2 + \|\underline{\lambda} \cdot \hat{p} + (\phi_{\hat{s}} - \xi) \cdot \mathbf{e}\|_2^2 + \underline{\theta}'^\top \underline{\theta}' \right. \\
& \left. + 2 \cdot \left|\sigma \cdot \boldsymbol{Q}_{\hat{s}} \boldsymbol{u} + \underline{\theta}'\right|^\top (\underline{\lambda} \cdot \hat{p} + (\phi_{\hat{s}} - \xi) \cdot \mathbf{e})\right),
\end{aligned}
$$

which is equivalent to

$$
(\Delta\underline{\lambda})^2 + 2\left(\sigma(-d_{\hat{s}} + \mathbf{e}^\top \boldsymbol{u}_{\hat{s}}) + \underline{\lambda} - \underline{\lambda}'\right)(\Delta\underline{\lambda}) - \left\| \begin{array}{c} \sigma \cdot \boldsymbol{Q}_{\hat{s}} \boldsymbol{u} + \underline{\theta}' \\ \underline{\lambda} \cdot \hat{p} + (\phi_{\hat{s}} - \xi) \cdot \mathbf{e} \end{array} \right\|_2^2
$$
$$
- 2 \cdot \left|\sigma \cdot \boldsymbol{Q}_{\hat{s}} \boldsymbol{u} + \underline{\theta}'\right|^\top (\underline{\lambda} \cdot \hat{p} + (\phi_{\hat{s}} - \xi) \cdot \mathbf{e}) \le 0.
$$

Hence, we have an upper bound for $\Delta\underline{\lambda}$ as follows:

$$
\begin{aligned}
\Delta\underline{\lambda} \le \quad & -\left(\sigma(-d_{\hat{s}} + \mathbf{e}^\top \boldsymbol{u}_{\hat{s}}) + \underline{\lambda} - \underline{\lambda}'\right) + \left[\left(\sigma(-d_{\hat{s}} + \mathbf{e}^\top \boldsymbol{u}_{\hat{s}}) + \underline{\lambda} - \underline{\lambda}'\right)^2 + \left\| \begin{array}{c} \sigma \cdot \boldsymbol{Q}_{\hat{s}} \boldsymbol{u} + \underline{\theta}' \\ \underline{\lambda} \cdot \hat{p} + (\phi_{\hat{s}} - \xi) \cdot \mathbf{e} \end{array} \right\|_2^2 + \right. \\
& \left. 2 \cdot \left|\sigma \cdot \boldsymbol{Q}_{\hat{s}} \boldsymbol{u} + \underline{\theta}'\right|^\top (\underline{\lambda} \cdot \hat{p} + (\phi_{\hat{s}} - \xi) \cdot \mathbf{e})\right]^{1/2},
\end{aligned}
$$

followed by which

$$
\begin{aligned}
\underline{\lambda}^\star \leq \quad & \underline{\lambda} - \left(\sigma(-d_{\hat{s}} + \mathbf{e}^\top \boldsymbol{u}_{\hat{s}}) + \underline{\lambda} - \underline{\lambda}'\right) + \left[ \left(\sigma(-d_{\hat{s}} + \mathbf{e}^\top \boldsymbol{u}_{\hat{s}}) + \underline{\lambda} - \underline{\lambda}'\right)^2 + \left\| \begin{array}{c} \sigma \cdot \boldsymbol{Q}_{\hat{s}} \boldsymbol{u} + \underline{\boldsymbol{\theta}}' \\ \underline{\lambda} \cdot \hat{\boldsymbol{p}} + (\phi_{\hat{s}} - \xi) \cdot \mathbf{e} \end{array} \right\|_2^2 + \right. \\
& \left. 2 \cdot \left|\sigma \cdot \boldsymbol{Q}_{\hat{s}} \boldsymbol{u} + \underline{\boldsymbol{\theta}}'\right|^\top (\underline{\lambda} \cdot \hat{\boldsymbol{p}} + (\phi_{\hat{s}} - \xi) \cdot \mathbf{e}) \right]^{1/2}.
\end{aligned}
$$

Since $(\underline{\lambda}, \underline{\boldsymbol{\theta}})$ is taken arbitrarily, our conclusion follows by taking $(\underline{\lambda}, \underline{\boldsymbol{\theta}}) = (\underline{\lambda}', \underline{\lambda}' \cdot \hat{\boldsymbol{p}})$. □

Lemma 8 provides a upper bound for the golden section search for computing the optimal $\underline{\lambda}^\star$ of problem $\underline{K}_s(\xi)$ (when $\xi < \phi_s$). The golden section search requires the function $\underline{f}_{\hat{s}}(\underline{\lambda})$ to be well-defined for all $\underline{\lambda} \in [0, \underline{\lambda}^{\mathrm{up}}]$, *i.e.*, the problem $\underline{f}_{\hat{s}}(\underline{\lambda})$ should be able to attain its optimal value for all $\underline{\lambda} \in [0, \underline{\lambda}^{\mathrm{up}}]$. This is guaranteed by the following lemma.

**Lemma 9** *Problem $\underline{f}_s(\underline{\lambda})$ can attain its optimal value for all $\underline{\lambda} \in \mathbb{R}_+$.*

*Proof of Lemma 9* Take $\underline{\lambda} \in \mathbb{R}_+$ arbitrarily. It holds for the second set of constraints in problem $\underline{f}_{\hat{s}}(\underline{\lambda})$ that

$$
\underline{\lambda} \cdot \mathbf{e} = B\underline{\boldsymbol{\theta}} \iff \mathbf{e}^\top \underline{\boldsymbol{\theta}}_{s,a} = \underline{\lambda} \ \forall s \in \mathcal{S}, a \in \mathcal{A}
$$

be the definition of $B$. This equivalence and constraint $\underline{\boldsymbol{\theta}} \in \mathbb{R}_+^{S \cdot A \cdot S}$ together implies that $\underline{f}_{\hat{s}}(\underline{\lambda})$ has a bounded feasible region. Moreover, note that problem $\underline{f}_{\hat{s}}(\underline{\lambda})$ has a continuous objective function a non-empty (with a feasible solution $\underline{\boldsymbol{\theta}} = \underline{\lambda} \cdot \hat{\boldsymbol{p}}$) and closed feasible set. Our conclusion then follows by the Weierstrass theorem. □

The golden section search returns a globally optimal solution in problem $\min_{\underline{\lambda} \in [0, \underline{\lambda}^{\mathrm{up}}]} \underline{f}_{\hat{s}}(\underline{\lambda})$ since $\underline{f}_{\hat{s}}(\underline{\lambda})$ is convex on $[0, \underline{\lambda}^{\mathrm{up}}]$.

**Lemma 10** *The function $\underline{f}_{\hat{s}}(\underline{\lambda})$ is convex on $[0, \underline{\lambda}^{\mathrm{up}}]$.*

*Proof of Lemma 10* Fix arbitrary $\underline{\lambda}, \underline{\lambda}' \in [0, \underline{\lambda}^{\mathrm{up}}]$ and $\kappa \in [0, 1]$. Let $\underline{\boldsymbol{\theta}} \in \mathcal{D}_{\hat{s}}(\underline{\lambda})$ and $\underline{\boldsymbol{\theta}}' \in \mathcal{D}_{\hat{s}}(\underline{\lambda}')$ satisfy

$$
h_{\hat{s}}(\underline{\lambda}, \underline{\boldsymbol{\theta}}) = \underline{f}_{\hat{s}}(\underline{\lambda}) \text{ and } h_{\hat{s}}(\underline{\lambda}', \underline{\boldsymbol{\theta}}') = \underline{f}_{\hat{s}}(\underline{\lambda}'),
$$

where we use

$$
\mathcal{D}_{\hat{s}}(\underline{\lambda}) = \left\{ \underline{\boldsymbol{\theta}} \in \mathbb{R}_+^{S \cdot A \cdot S} \ \middle| \ \begin{array}{ll} \|\underline{\boldsymbol{\theta}} - \underline{\lambda} \cdot \hat{\boldsymbol{p}}\|_\infty \leq \phi_{\hat{s}} - \xi & \\ \mathbf{e}^\top \underline{\boldsymbol{\theta}}_{s,a} = \underline{\lambda} & \forall (s, a) \in \mathcal{S} \times \mathcal{A} \end{array} \right\}
$$

to denote the feasible set of problem $\underline{f}_{\hat{s}}(\underline{\lambda})$ and

$$
h_{\hat{s}}(\underline{\lambda}, \underline{\boldsymbol{\theta}}) = -\underline{\lambda}(d_{\hat{s}} - \mathbf{e}^\top \boldsymbol{u}_{\hat{s}}) - \underline{\boldsymbol{\theta}}^\top \boldsymbol{Q}_{\hat{s}} \boldsymbol{u} + \frac{1}{2\sigma}\left((\underline{\lambda} - \underline{\lambda}')^2 + \|\underline{\boldsymbol{\theta}} - \underline{\boldsymbol{\theta}}'\|_2^2\right)
$$

as the objective function of this problem. Note the the solution $((1 - \kappa) \cdot \underline{\lambda} + \kappa \underline{\lambda}', (1 - \kappa) \cdot \underline{\boldsymbol{\theta}} + \kappa \cdot \underline{\boldsymbol{\theta}}')$ is feasible to problem $\underline{K}_{\hat{s}}(\xi)$ since its feasible set is convex. It then follows that $(1 - \kappa) \cdot \underline{\boldsymbol{\theta}} + \kappa \cdot \underline{\boldsymbol{\theta}}' \in \mathcal{D}_{\hat{s}}((1 - \kappa) \cdot \underline{\lambda} + \kappa \underline{\lambda}')$ and

$$
\begin{aligned}
\underline{f}_{\hat{s}}((1 - \kappa) \cdot \underline{\lambda} + \kappa \underline{\lambda}') \leq \quad & h_{\hat{s}}((1 - \kappa) \cdot \underline{\lambda} + \kappa \underline{\lambda}', (1 - \kappa) \cdot \underline{\boldsymbol{\theta}} + \kappa \cdot \underline{\boldsymbol{\theta}}') \\
\leq \quad & (1 - \kappa) \cdot h_{\hat{s}}(\underline{\lambda}, \underline{\boldsymbol{\theta}}) + \kappa \cdot h_{\hat{s}}(\underline{\lambda}', \underline{\boldsymbol{\theta}}') \\
= \quad & (1 - \kappa) \cdot \underline{f}_{\hat{s}}(\underline{\lambda}) + \kappa \cdot \underline{f}_{\hat{s}}(\underline{\lambda}').
\end{aligned}
$$

Here the first inequality follows because of the definition of $\underline{f}_{\hat{s}}$, the second one holds since the function $h_{\hat{s}}$ is convex. Our conclusion follows since we take $\underline{\lambda}, \underline{\lambda}'$ and $\kappa$ arbitrarily. □

Similar as $\overline{f}_{\hat{s}}(\overline{\lambda})$, problem $\underline{f}_{\hat{s}}(\underline{\lambda})$ is also decomposable into $SA$ subproblems and the $(s,a)$-th subproblem is

$$
\begin{aligned}
\min \quad & \frac{1}{2\sigma} \cdot \underline{\boldsymbol{\theta}}_{s,a}^{\top}\underline{\boldsymbol{\theta}}_{s,a} + \underline{\boldsymbol{\theta}}_{s,a}^{\top}\left(\boldsymbol{x}_{\hat{s},s,a} - \frac{1}{\sigma}\cdot\boldsymbol{\theta}'_{s,a}\right) \\
\text{s.t.} \quad & [\underline{\lambda}\hat{p}_{s,a,s'} - (\phi_{\hat{s}} - \xi)]_{+} \leq \underline{\theta}_{s,a,s'} \leq \underline{\lambda}\hat{p}_{s,a,s'} + (\phi_{\hat{s}} - \xi) \quad \forall s' \in \mathcal{S} \\
& \mathbf{e}^{\top}\underline{\boldsymbol{\theta}}_{s,a} = \underline{\lambda} \\
& \underline{\boldsymbol{\theta}}_{s,a} \in \mathbb{R}^{S}
\end{aligned}
\tag{42}
$$

for all $(s,a) \in \mathcal{S} \times \mathcal{A}$. We solve problem (42) via interval search as we did for $\overline{f}_{\hat{s}}(\overline{\lambda})$.

**Proposition 15** *Problem* (42) *can be solved in time* $\mathcal{O}(S \log S)$.

*Proof of Proposition 15* The Lagrangian function of problem (42) is

$$
\begin{aligned}
L(\underline{\boldsymbol{\theta}}_{s,a}, \boldsymbol{\eta}, \boldsymbol{\varphi}, \rho) = \quad & \frac{1}{2\sigma} \cdot \underline{\boldsymbol{\theta}}_{s,a}^{\top}\underline{\boldsymbol{\theta}}_{s,a} + \underline{\boldsymbol{\theta}}_{s,a}^{\top}\left(\boldsymbol{x}_{\hat{s},s,a} - \frac{1}{\sigma}\cdot\boldsymbol{\theta}'_{s,a}\right) + \sum_{s' \in \mathcal{S}} \eta_{s'} \cdot ([\underline{\lambda}\hat{p}_{s,a,s'} - (\phi_{\hat{s}} - \xi)]_{+} - \underline{\theta}_{s,a,s'}) \\
& + \sum_{s' \in \mathcal{S}} \varphi_{s'} \cdot (\underline{\theta}_{s,a,s'} - (\underline{\lambda}\hat{p}_{s,a,s'} + (\phi_{\hat{s}} - \xi))) + \rho \cdot (\underline{\lambda} - \mathbf{e}^{\top}\underline{\boldsymbol{\theta}}_{s,a})
\end{aligned}
$$

with dual variables $\boldsymbol{\eta}$, $\boldsymbol{\varphi} \in \mathbb{R}_{+}^{S}$ and $\rho \in \mathbb{R}$. We then can provide the KKT conditions of problem (42) as follows:

$$
\begin{cases}
\underline{\theta}_{s,a,s'} \geq [\underline{\lambda}\hat{p}_{s,a,s'} - (\phi_{\hat{s}} - \xi)]_{+} & \forall s' \in \mathcal{S} \\
\underline{\theta}_{s,a,s'} \leq \underline{\lambda}\hat{p}_{s,a,s'} + (\phi_{\hat{s}} - \xi) & \forall s' \in \mathcal{S} \\
\mathbf{e}^{\top}\underline{\boldsymbol{\theta}}_{s,a} = \underline{\lambda} \\
\boldsymbol{\eta} \geq \mathbf{0} \\
\boldsymbol{\varphi} \geq \mathbf{0} \\
\eta_{s'} \cdot ([\underline{\lambda}\hat{p}_{s,a,s'} - (\phi_{\hat{s}} - \xi)]_{+} - \underline{\theta}_{s,a,s'}) = 0 & \forall s' \in \mathcal{S} \\
\varphi_{s'} \cdot (\underline{\theta}_{s,a,s'} - (\underline{\lambda}\hat{p}_{s,a,s'} + (\phi_{\hat{s}} - \xi))) = 0 & \forall s' \in \mathcal{S} \\
\nabla_{\underline{\boldsymbol{\theta}}_{s,a}} L(\underline{\boldsymbol{\theta}}_{s,a}, \boldsymbol{\eta}, \boldsymbol{\varphi}, \rho) = \frac{1}{\sigma} \cdot \underline{\boldsymbol{\theta}}_{s,a} + \left(\boldsymbol{x}_{\hat{s},s,a} - \frac{1}{\sigma} \cdot \boldsymbol{\theta}'_{s,a}\right) - \boldsymbol{\eta} + \boldsymbol{\varphi} - \rho \cdot \mathbf{e} = \mathbf{0}.
\end{cases}
$$

It then follows that

$$
\underline{\theta}_{s,a,s'} = \begin{cases}
\underline{\lambda}\hat{p}_{s,a,s'} + (\phi_{\hat{s}} - \xi) & \forall s' \in \mathcal{S} : \varphi_{s'} \neq 0 \\
\sigma \cdot \left(\rho + \frac{1}{\sigma}\theta'_{s,a,s'} - x_{\hat{s},s,a,s'}\right) & \forall s' \in \mathcal{S} : \eta_{s'} = 0 \text{ and } \varphi_{s'} = 0 \\
[\underline{\lambda}\hat{p}_{s,a,s'} - (\phi_{\hat{s}} - \xi)]_{+} & \forall s' \in \mathcal{S} : \eta_{s'} \neq 0.
\end{cases}
$$

It then suffices to solve the equation $H_{s,a}(\rho) = \underline{\lambda}$, after which we obtain $\underline{\theta}_{s,a,s'}^{\star} = H_{s,a,s'}(\rho^{\star}) \; \forall s \in \mathcal{S}$, where $H_{s,a}(\rho) = \sum_{s' \in \mathcal{S}} H_{s,a,s'}(\rho)$ and

$$
H_{s,a,s'}(\rho) = \begin{cases}
\underline{\lambda}\hat{p}_{s,a,s'} + (\phi_{\hat{s}} - \xi) & \text{if } \rho \geq \frac{1}{\sigma} \cdot (\underline{\lambda}\hat{p}_{s,a,s'} + (\phi_{\hat{s}} - \xi)) + x_{\hat{s},s,a,s'} - \frac{1}{\sigma}\theta'_{s,a,s'} \\
[\underline{\lambda}\hat{p}_{s,a,s'} - (\phi_{\hat{s}} - \xi)]_{+} & \text{if } \rho < \frac{1}{\sigma} \cdot [\underline{\lambda}\hat{p}_{s,a,s'} - (\phi_{\hat{s}} - \xi)]_{+} + x_{\hat{s},s,a,s'} - \frac{1}{\sigma}\theta'_{s,a,s'} \\
\sigma \cdot \left(\rho + \frac{1}{\sigma}\theta'_{s,a,s'} - x_{\hat{s},s,a,s'}\right) & \text{otherwise}
\end{cases}
$$

for all $s' \in \mathcal{S}$. Since by definition, $H_{s,a,s'}, s' \in \mathcal{S}$ are all piecewise linear and non-decreasing, their sum $H_{s,a} = \sum_{s' \in \mathcal{S}} H_{s,a,s'}$ is thus also piecewise linear and non-decreasing with $2S$ breakpoints $\frac{1}{\sigma} \cdot (\underline{\lambda}\hat{p}_{s,a,s'} + (\phi_{\hat{s}} - \xi)) - \frac{1}{\sigma}\theta'_{s,a,s'} + x_{\hat{s},s,a,s'}, \; s' \in \mathcal{S}$ (that we call "upper breakpoints") and $\frac{1}{\sigma} \cdot [\underline{\lambda}\hat{p}_{s,a,s'} - (\phi_{\hat{s}} - \xi)]_{+} - \frac{1}{\sigma}\theta'_{s,a,s'} + x_{\hat{s},s,a,s'}, \; s' \in \mathcal{S}$ (that we call "lower breakpoints"). Sorting

---

**Algorithm 3** Interval-Searching Algorithm for Problem (42)

---

Compute all the upper breakpoints $\overline{\rho}_{s'} \leftarrow \frac{1}{\sigma}(\underline{\lambda}\hat{p}_{s,a,s'} + (\phi_{\hat{s}} - \xi)) - \frac{1}{\sigma}\underline{\theta}'_{s,a,s'} + x_{\hat{s},s,a,s'}, s' \in \mathcal{S}$
and lower breakpoints $\underline{\rho}_{s'} \leftarrow \frac{1}{\sigma}[\underline{\lambda}\hat{p}_{s,a,s'} - (\phi_{\hat{s}} - \xi)]_+ - \frac{1}{\sigma}\underline{\theta}'_{s,a,s'} + x_{\hat{s},s,a,s'}, s' \in \mathcal{S}$
Sort the breakpoints in ascending order as $\rho_1 \leq \cdots \leq \rho_{2S}$
Initialize $\chi \leftarrow \sigma$ and $\psi \leftarrow \sum_{s' \in \mathcal{S}: s' \neq p_1(1)}[\underline{\lambda}\hat{p}_{s,a,s'} - (\phi_{\hat{s}} - \xi)]_+ + \sigma \cdot (\frac{1}{\sigma}\underline{\theta}'_{s,a,p_1(1)} - x_{\hat{s},s,a,p_1(1)})$
Initialize the index set for the upper breakpoints $\mathcal{U} \leftarrow \emptyset$ and the one for the lower breakpoints
$\mathcal{L} \leftarrow \mathcal{S} \setminus p_1(1)$
**for** $k = 1, \cdots, 2S - 1$ **do**
  **if** $\chi \cdot \rho_{k+1} + \psi \geq \underline{\lambda}$ **then**
    $\rho^{\star} \leftarrow \frac{\underline{\lambda} - \psi}{\chi}$
    **for** $s' = 1, \cdots, S$ **do**

$$
\underline{\theta}^{\star}_{s,a,s'} \leftarrow \begin{cases} \underline{\lambda}\hat{p}_{s,a,s'} + (\phi_{\hat{s}} - \xi) & \forall s' \in \mathcal{U} \\ [\underline{\lambda}\hat{p}_{s,a,s'} - (\phi_{\hat{s}} - \xi)]_+ & \forall s' \in \mathcal{L} \\ \sigma \cdot (\rho^{\star} + \frac{1}{\sigma}\underline{\theta}'_{s,a,s'} - x_{\hat{s},s,a,s'}) & \forall s' \in \mathcal{S} \setminus (\mathcal{U} \cup \mathcal{L}); \end{cases}
$$

    **end for**
  **else if** $p_2(k+1) = $ "upper" **then**
    $\chi \leftarrow \chi - \sigma$
    $\psi \leftarrow \psi - \sigma \cdot (\frac{1}{\sigma}\underline{\theta}'_{s,a,p_1(k+1)} - x_{\hat{s},s,a,p_1(k+1)}) + \underline{\lambda}\hat{p}_{s,a,p_1(k+1)} + (\phi_{\hat{s}} - \xi)$
  **else**
    $\chi \leftarrow \chi + \sigma$
    $\psi \leftarrow \psi + \sigma \cdot (\frac{1}{\sigma}\underline{\theta}'_{s,a,p_1(k+1)} - x_{\hat{s},s,a,p_1(k+1)}) - [\underline{\lambda}\hat{p}_{s,a,p_1(k+1)} - (\phi_{\hat{s}} - \xi)]_+$
  **end if**
**end for**
**Output:** Solution $\underline{\theta}^{\star}_{s,a}$

---

all $2S$ breakpoints in an ascending order $\rho_1 \leq \ldots \leq \rho_{2S}$ and sequentially searching the intervals $[\rho_1, \rho_2], [\rho_2, \rho_3], \cdots, [\rho_{2S-1}, \rho_{2S}]$, we can then obtain $\rho^{\star}$ and $\underline{\theta}^{\star}_{s,a,s'} = H_{s,a,s'}(\rho^{\star}) \; \forall s' \in \mathcal{S}$.

The time complexity $\mathcal{O}(S \log S)$ is from sorting the breakpoints. $\qquad \square$

We provide the pseudocode for the interval-searching algorithm in Algorithm 3. Here, the functions $p_1(\cdot) : [2S] \mapsto \mathcal{S}$ and $p_2(\cdot) : [2S] \mapsto \{$"lower", "upper"$\}$ map the indices of the non-decreasing breakpoint sequence to the indices and types of breakpoints (*i.e.*, "lower" or "upper"), respectively. For example, if $\rho_4$ corresponds to $\underline{\rho}_6$, then we have $p_1(4) = 6$ and $p_2(4) = $ "upper".

# F    ADDITIONAL DETAILS OF THE SIMULATION STUDY

## F.1    SAMPLING WEIGHT SAMPLES VIA BAYESIAN IRL

As we introduced in Section 1, we assume that the rewards can be parameterized as a linear combination of $K$ features $r = Fw \in \mathbb{R}^{S \cdot A}$, where $F \in \mathbb{R}^{S \cdot A \times K}$ is the feature matrix and $w \in \mathbb{R}^K$ is the reward weight vector (Brown et al., 2020b). Under this assumption, learning the reward function is reduced to learning the reward weight vector. In the first part of our SRIRL, based on the demonstration $\mathcal{D} = \{(s_1, a_1), (s_2, a_2), \cdots, (s_L, a_L)\}$ of the expert, we follow the Bayesian IRL (Ramachandran & Amir, 2007) to learn the posterior distribution of the weights $\mathbb{P}(w \mid \mathcal{D}) \propto \mathbb{P}(\mathcal{D} \mid w) \cdot \mathbb{P}(w)$, where

$$
\mathbb{P}(\mathcal{D}|w) = \prod_{(s,a) \in \mathcal{D}} \mathbb{P}((s,a) \mid w) = \prod_{(s,a) \in \mathcal{D}} \frac{\exp(\delta q^{\star}_w(s,a))}{\sum_{a' \in \mathcal{A}} \exp(\delta q^{\star}_w(s,a'))}
$$

is the likelihood function. Here $q^{\star}_w : \mathcal{S} \times \mathcal{A} \to \mathbb{R}$ is the optimal Q-value function given the weights $w$, and $\delta \in \mathbb{R}_{++}$ represents the confidence of the expert in the optimality. The first equality is due to our assumption that the expert is following a stationary policy, and the second one is because we assume the expert follows a soft-max policy. We then follow the Markov chain Monte Carlo (MCMC) sampling to generate samples from the posterior distribution (Brown et al., 2020b), as in

---

**Algorithm 4** Bayesian IRL

---

**Input:** An empty weight sample recorder $\mathcal{W}$
*// Generate initial weights*
Randomly generate $\boldsymbol{w}_{\mathrm{curr}}$, where each of its entries is randomly drawn from the normal distribution $\mathcal{N}(0, \sigma^2)$;
$\boldsymbol{w}_{\mathrm{curr}} \leftarrow (1/\|\boldsymbol{w}_{\mathrm{curr}}\|_2) \cdot \boldsymbol{w}_{\mathrm{curr}}$;
*// Compute Q values*
Compute the optimal Q-value function $\boldsymbol{q}_{\mathrm{curr}} \in \mathbb{R}^{S \cdot A}$ given $\boldsymbol{w}_{\mathrm{curr}}$;
*// Compute log-likelihood*
$L_{\mathrm{curr}} \leftarrow \sum_{(s,a) \in \mathcal{D}} \left\{ \delta \cdot q_{\mathrm{curr},s,a} - \log(\sum_{a' \in \mathcal{A}} \exp(\delta \cdot q_{\mathrm{curr},s,a'})) \right\}$;
**for** $k = 1, \ldots, K$ **do**
   *// Generate proposal weights*
   Randomly generate proposal weights $\boldsymbol{w}_{\mathrm{prop}}$ from the proposal distribution $\mathcal{P}_{\boldsymbol{w}_{\mathrm{curr}}}$;
   $\boldsymbol{w}_{\mathrm{prop}} \leftarrow (1/\|\boldsymbol{w}_{\mathrm{prop}}\|_2) \cdot \boldsymbol{w}_{\mathrm{prop}}$;
   *// Compute Q values*
   Compute the optimal Q-value function $\boldsymbol{q}_{\mathrm{prop}} \in \mathbb{R}^{S \cdot A}$ given $\boldsymbol{w}_{\mathrm{prop}}$;
   *// Compute log-likelihood*
   $L_{\mathrm{prop}} \leftarrow \sum_{(s,a) \in \mathcal{D}} \left\{ \delta \cdot q_{\mathrm{prop},s,a} - \log(\sum_{a' \in \mathcal{A}} \exp(\delta \cdot q_{\mathrm{prop},s,a'})) \right\}$;
   Compute acceptance probability $p_{\mathrm{acp}} = \min\{1.0, \exp(L_{\mathrm{prop}} - L_{\mathrm{curr}})\}$
   Randomly generate a number $t$ from the uniform distribution on $[0,1]$;
   **if** $t < p_{\mathrm{acp}}$ **then**
     Push back $\boldsymbol{w}_{\mathrm{prop}}$ to $\mathcal{W}$;
     $L_{\mathrm{curr}} \leftarrow L_{\mathrm{prop}}$;
     $\boldsymbol{w}_{\mathrm{curr}} \leftarrow \boldsymbol{w}_{\mathrm{prop}}$;
   **else**
     Push back $\boldsymbol{w}_{\mathrm{curr}}$ to $\mathcal{W}$;
   **end if**
**end for**
**Output:** $\mathcal{W}$

---

Algorithm 4. Here we set $\sigma = 0.2$, $\delta = 10$, the proposal distribution $\mathcal{P}_{\boldsymbol{w}_{\mathrm{curr}}}$ is a multivariate normal distribution with mean $\boldsymbol{w}_{\mathrm{curr}}$ and covariance $\sigma^2 \cdot \boldsymbol{I}$. Note that, here we will discard the first $N_{\mathrm{burn}}$ samples in $\mathcal{W}$ as we consider a length-$N_{\mathrm{burn}} = 500$ burn-in period, and we will skip four samples every time after we accept one in order to reduce auto-correlation. *E.g.*, if $K = 2000$, *i.e.*, there are 2000 weight samples in $\mathcal{W}$ output by Algorithm 4, then we will take samples $501, 506, 511, \ldots$ as the final weight samples that we consider.

### F.2 POLLUTION TO TRANSITION KERNELS

In our simulation, under the polluted transition kernel, the agent may slip to a neighboring cell along the direction towards which she chooses to move. For ease of description, in Figure 4 we present the lava corridor with each cell numbered. The possible next states of the agent are then as shown in Table 2.

Let $\mathcal{M} \subseteq \mathcal{S} \times \mathcal{A} \times \mathcal{S}$ be the set of state-action-state tuples in which the polluted transition probability can be nonzero. For example, we can check Table 2 and see $(1, \mathrm{Left}, 1), (1, \mathrm{Left}, 6) \in \mathcal{M}$, while $(1, \mathrm{Left}, 2), (1, \mathrm{Left}, 3) \notin \mathcal{M}$. To generate the polluted transition kernel $\boldsymbol{p}^{\mathrm{ag}}$, we will first generate a noise vector $\boldsymbol{p}^{\mathrm{noise}} \in \mathbb{R}^{S \cdot A \cdot S}$, where for each $(s, a, s') \in \mathcal{M}$, $p^{\mathrm{noise}}_{s,a,s'}$ is randomly generated from a uniform distribution on $[0, 1]$, while $p^{\mathrm{noise}}_{s,a,s'} = 0$ for all $(s, a, s') \in \mathcal{S} \times \mathcal{A} \times \mathcal{S} \backslash \mathcal{M}$. Let $\delta \in [0, 1]$ be the pollution rate. The polluted kernel then is obtained by normalizing $(1 - \delta) \cdot \boldsymbol{p}^{\mathrm{ex}} + \delta \cdot \boldsymbol{p}^{\mathrm{noise}}$ so that $\mathbf{e}^\top \boldsymbol{p}^{\mathrm{ag}}_{s,a} = 1 \ \forall s \in \mathcal{S}, a \in \mathcal{A}$.

### F.3 SRIRL WITH LIMITED NEXT STATES

As in Appendix F.2, we let $\mathcal{M} \subseteq \mathcal{S} \times \mathcal{A} \times \mathcal{S}$ be the set of state-action-state tuples in which the polluted transition probability can be nonzero. In our SRIRL (2), we consider a support set $\mathcal{P} = \{\boldsymbol{p} \in \mathbb{R}^{S \cdot A \cdot S}_+ \mid \mathbf{e}^\top \boldsymbol{p}_{s,a} = 1 \ \forall s \in \mathcal{S}, a \in \mathcal{A}\}$, impling that we consider $\mathcal{M} = \mathcal{S} \times \mathcal{A} \times \mathcal{S}$.

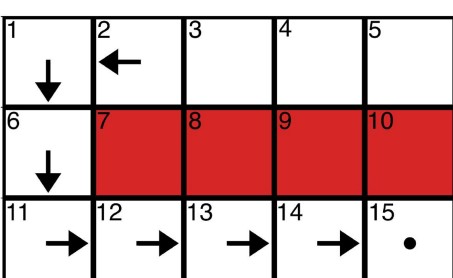

Figure 4: Lava corridor with cell numbers.

For all $s \in \mathcal{S}$ and $a \in \mathcal{A}$, let $\mathcal{R}(s,a) = \{s' \in \mathcal{S} \mid (s,a,s') \in \mathcal{M}\}$ be the set of all the possible next states when current state is $s$ and action $a$ is taken. We argue that all our theoretical results still holds when considering

$$\mathcal{P} = \left\{ \boldsymbol{p} \in \mathbb{R}_+^{S \cdot A \cdot S} \; \middle| \; \sum_{s' \in \mathcal{S}} p_{s,a,s'} = 1 \quad \forall s \in \mathcal{S}, a \in \mathcal{A}, \; p_{s,a,s'} = 0 \quad \forall (s,a,s') \notin \mathcal{M} \right\}. \quad (43)$$

Let

$$\mathcal{P}^{\mathrm{trim}} = \left\{ \boldsymbol{p} \in \mathbb{R}_+^{\sum_{s \in \mathcal{S}} \sum_{a \in \mathcal{A}} \mathcal{R}(s,a)} \; \middle| \; \sum_{s' \in \mathcal{R}(s,a)} p_{s,a,s'} = 1 \quad \forall s \in \mathcal{S}, a \in \mathcal{A} \right\}.$$

We can then formulate our SRIRL equipped with the modified support set $\mathcal{P}$ in (43) as follows:

$$
\begin{aligned}
\min \quad & \boldsymbol{\phi}^\top \boldsymbol{k} \\
\text{s.t.} \quad & \mathbf{e}^\top \boldsymbol{u}_s - \boldsymbol{p}^\top \boldsymbol{Q}_s^{\mathrm{trim}} \boldsymbol{u} - d_s \leq k_s \cdot \ell(\boldsymbol{p}, \hat{\boldsymbol{p}}^{\mathrm{trim}}) && \forall \boldsymbol{p} \in \mathcal{P}^{\mathrm{trim}}, s \in \mathcal{S} \\
& \mathbf{e}^\top \boldsymbol{u}_s - \boldsymbol{p}^\top \boldsymbol{Q}_s^{\mathrm{trim}} \boldsymbol{u} - d_s \geq -k_s \cdot \ell(\boldsymbol{p}, \hat{\boldsymbol{p}}^{\mathrm{trim}}) && \forall \boldsymbol{p} \in \mathcal{P}^{\mathrm{trim}}, s \in \mathcal{S} \\
& \omega \cdot \mathbb{E}_{\mathbb{P}(\boldsymbol{w} \mid \mathcal{D})} [\tilde{\boldsymbol{w}}^\top (\boldsymbol{F}^\top \boldsymbol{u} - \boldsymbol{f}_{\mathrm{E}})] \\
& + (1 - \omega) \cdot \mathbb{P}(\boldsymbol{w} \mid \mathcal{D})\text{-CVaR}_\varepsilon \left[ \tilde{\boldsymbol{w}}^\top (\boldsymbol{F}^\top \boldsymbol{u} - \boldsymbol{f}_{\mathrm{E}}) \right] \geq \tau \\
& \boldsymbol{u} \in \mathbb{R}_+^{S \cdot A}, \boldsymbol{k} \in \mathbb{R}_+^S,
\end{aligned}
\quad (44)
$$

where $\boldsymbol{Q}_s^{\mathrm{trim}} \in \mathbb{R}^{(\sum_{s \in \mathcal{S}} \sum_{a \in \mathcal{A}} \mathcal{R}(s,a)) \times S \cdot A}$ is comprised of the rows of $\boldsymbol{Q}_s$ for all the row index $(s,a,s') \in \mathcal{M}$, and $\hat{\boldsymbol{p}}^{\mathrm{trim}}$ is formed by the components of $\hat{\boldsymbol{p}}$ whose index $(s,a,s') \in \mathcal{M}$.

**Proposition 16** *Equipped with $\ell(\boldsymbol{p}, \hat{\boldsymbol{p}}) = \|\boldsymbol{p} - \hat{\boldsymbol{p}}\|$, RSIRl (2) equipped with $\mathcal{P}$ as in (43) is equivalent to (44).*

*Proof of Proposition 16* It is sufficient to argue that the first (respectively, the second) set of constraints in (2) is equivalent to the first (respectively, the second) set of constraints in (44).

| Current State | Action | Possible Next States | Current State | Action | Possible Next States | Current State | Action | Possible Next States |
|---|---|---|---|---|---|---|---|---|
| 1 | Left | $1,6$ | 6 | Left | $1,6,11$ | 11 | Left | $6,11$ |
|  | Right | $1,2,6,7$ |  | Right | $1,2,6,7,11,12$ |  | Right | $6,7,11,12$ |
|  | Up | $1,2$ |  | Up | $1,2,6,7$ |  | Up | $6,7,11,12$ |
|  | Down | $1,2,6,7$ |  | Down | $6,7,11,12$ |  | Down | $11,12$ |
| 2 | Left | $1,2,6,7$ | 7 | Left | $1,2,6,7,11,12$ | 12 | Left | $6,7,11,12$ |
|  | Right | $2,3,7,8$ |  | Right | $2,3,7,8,12,13$ |  | Right | $7,8,12,13$ |
|  | Up | $1,2,3$ |  | Up | $1,2,3,6,7,8$ |  | Up | $6,7,8,11,12,13$ |
|  | Down | $1,2,3,6,7,8$ |  | Down | $6,7,8,11,12,13$ |  | Down | $11,12,13$ |
| 3 | Left | $2,3,7,8$ | 8 | Left | $2,3,7,8,12,13$ | 13 | Left | $7,8,12,13$ |
|  | Right | $3,4,8,9$ |  | Right | $3,4,8,9,13,14$ |  | Right | $8,9,13,14$ |
|  | Up | $2,3,4$ |  | Up | $2,3,4,7,8,9$ |  | Up | $7,8,9,12,13,14$ |
|  | Down | $2,3,4,7,8,9$ |  | Down | $7,8,9,12,13,14$ |  | Down | $12,13,14$ |
| 4 | Left | $3,4,8,9$ | 9 | Left | $3,4,8,9,13,14$ | 14 | Left | $8,9,13,14$ |
|  | Right | $4,5,9,10$ |  | Right | $4,5,9,10,14,15$ |  | Right | $9,10,14,15$ |
|  | Up | $3,4,5$ |  | Up | $3,4,5,8,9,10$ |  | Up | $8,9,10,13,14,15$ |
|  | Down | $3,4,5,8,9,10$ |  | Down | $8,9,10,13,14,15$ |  | Down | $13,14,15$ |
| 5 | Left | $4,5,9,10$ | 10 | Left | $4,5,9,10,14,15$ | 15 | Left | $9,10,14,15$ |
|  | Right | $5,10$ |  | Right | $9,10,14,15$ |  | Right | $10,15$ |
|  | Up | $4,5$ |  | Up | $4,5,9,10$ |  | Up | $9,10,14,15$ |
|  | Down | $4,5,9,10$ |  | Down | $9,10,14,15$ |  | Down | $14,15$ |

Table 2: Possible next states for the *lava corridor* environment when the transition kernel is polluted.

Let $s \in \mathcal{S}$ be arbitrarily fixed. On the one hand, for any $\boldsymbol{p} \in \mathcal{P}$, by definitions of $\mathcal{P}$ (43) and $\mathcal{P}^{\mathrm{trim}}$, there must exist only one $\boldsymbol{p}' \in \mathcal{P}^{\mathrm{trim}}$ that satisfies

$$p'_{s,a,s'} = p_{s,a,s'} \quad \forall s \in \mathcal{S}, a \in \mathcal{A}, s' \in \mathcal{R}(s,a).$$

Then, by definition of $\boldsymbol{Q}_s^{\mathrm{trim}}$ and $\hat{\boldsymbol{p}}^{\mathrm{trim}}$, we must have

$$\boldsymbol{p}'^{\top} \boldsymbol{Q}_s^{\mathrm{trim}} = \boldsymbol{p}^{\top} \boldsymbol{Q}_s$$

and

$$\ell(\boldsymbol{p}', \hat{\boldsymbol{p}}^{\mathrm{trim}}) = \ell(\boldsymbol{p}, \hat{\boldsymbol{p}}).$$

On the other hand, for any $\boldsymbol{p}' \in \mathcal{P}^{\mathrm{trim}}$, by definition of $\mathcal{P}$ (43) and $\mathcal{P}^{\mathrm{trim}}$, there must exist only one $\boldsymbol{p} \in \mathcal{P}$ that satisfies

$$p_{s,a,s'} = \begin{cases} p'_{s,a,s'} & (s,a,s') \in \mathcal{S} \times \mathcal{A} \times \mathcal{R}(s,a) \\ 0 & \text{otherwise.} \end{cases}$$

Then, by definition of $\boldsymbol{Q}_s^{\mathrm{trim}}$ and $\hat{\boldsymbol{p}}^{\mathrm{trim}}$, we must have

$$\boldsymbol{p}^{\top} \boldsymbol{Q}_s = \boldsymbol{p}'^{\top} \boldsymbol{Q}_s^{\mathrm{trim}}$$

and

$$\ell(\boldsymbol{p}, \hat{\boldsymbol{p}}) = \ell(\boldsymbol{p}', \hat{\boldsymbol{p}}^{\mathrm{trim}}).$$

Our conclusion then follows since $s$ is arbitrarily taken. □

By Proposition 16, we then can focus on our SRIRL (2) equipped with $\mathcal{P}$ (43) by looking at its equivalent problem (44), where our theoretical results and tailored algorithm applies.

### F.4 Additional Details of MAXENT

The benchmark model maximum entropy inverse reinforcement learning (MAXENT) is proposed by Ziebart et al. (2008), and we follow its implementation by Brown et al. (2020b), where we assume that the probability that the expert outputs a trajectory $\zeta$ is proportional to the exponential to $\beta R(\zeta)$, where $\beta$ is a Boltzmann parameter and $R(\zeta)$ is the total discounted reward of the trajectory of $\zeta$. We set $\beta = 10$. We use projected gradient descent to compute the maximum likelihood estimation of the weight vector of the expert, where in every iteration we project the weight vector to $\{\boldsymbol{w} \in \mathbb{R}^K \mid \|\boldsymbol{w}\|_2 = 1\}$. The learning rate is set as $0.01$, and we stop the algorithm when the $L_2$-norm of the gradient is less than $10^{-5}$ or a maximal number of iterations is reached. Here we set the maximal number of iterations to be $S$,

### F.5 Addtional Details of LPAL

We use linear programming apprenticeship learning (LPAL) (Syed et al., 2008) as one of our benchmark models, and we consider its variant implemented by Brown et al. (2020b), where the latter is free from the restriction that the weight vector must be non-negative. Specifically, we aim to solve

$$
\begin{aligned}
\max_{\boldsymbol{u}} \min_{\boldsymbol{w}} \quad & \boldsymbol{u}^\top \boldsymbol{F} \boldsymbol{w} - \boldsymbol{u}_{\mathrm{E}}^\top \boldsymbol{F} \boldsymbol{w} \\
\text{s.t.} \quad & \mathbf{e}^\top \boldsymbol{u}_s - \boldsymbol{p}^\top \boldsymbol{Q}_s \boldsymbol{u} - d_s = 0 \quad \forall s \in \mathcal{S} \\
& \|\boldsymbol{w}\|_1 \leq 1 \\
& \boldsymbol{u} \in \mathbb{R}_+^{S \cdot A}, \boldsymbol{w} \in \mathbb{R}^K,
\end{aligned}
\tag{45}
$$

where $\boldsymbol{u}_{\mathrm{E}}$ records the empirical total discounted occupancy of different state action pairs of the expert, i.e., $u_{\mathrm{E},s,a} = (1/|\mathcal{T}|) \sum_{t \in [|\mathcal{T}|]} \sum_{l \in [L]} \gamma^{l-1} \mathbf{1}_{s_{t,l}=s \,\wedge\, a_{t,l}=a} \ \forall (s,a) \in \mathcal{S} \times \mathcal{A}$. It thus holds that $\boldsymbol{f}_{\mathrm{E}} = \boldsymbol{F}^\top \boldsymbol{u}_{\mathrm{E}}$.

Brown et al. (2020b) provides an equivalent reformulation of (45) as a linear program, which we provide in the following proposition only for ease of reference.

**Proposition 17** *Problem* (45) *is equivalent to a linear program as follows:*

$$
\begin{aligned}
\max \quad & -x \\
\text{s.t.} \quad & x \cdot \mathbf{e} - \boldsymbol{F}^\top \boldsymbol{u} \leq -\boldsymbol{f}_{\mathrm{E}} \\
& -x \cdot \mathbf{e} + \boldsymbol{F}^\top \boldsymbol{u} \leq \boldsymbol{f}_{\mathrm{E}} \\
& \mathbf{e}^\top \boldsymbol{u}_s - \boldsymbol{p}^\top \boldsymbol{Q}_s \boldsymbol{u} - d_s = 0 \quad \forall s \in \mathcal{S} \\
& \boldsymbol{u} \in \mathbb{R}_+^{S \cdot A}, x \in \mathbb{R}.
\end{aligned}
$$

### F.6 Addtional Results

Figure 5 reports the policies of BROIL/SRIRL $(\tau = \mathrm{T_B}(\hat{\boldsymbol{p}}))^2$ under different values of the weight parameter $\omega \in \{0, 0.5, 1.0\}$. Remember that a larger value of $\omega$ corresponds to a less risk-averse attitude towards reward uncertainty, with which the result here is consistent: with a larger value of $\omega$, the agent here in the lava corridor is more willing to take a shortcut by walking on the red cell (to be more specific, the rightmost red cell), reflecting a less risk-averse attitude. This observation verifies the flexible risk-averseness towards reward uncertainty in IRL of our SRIRL.

---

[2]Note that by Proposition 1, the optimal solution $\boldsymbol{u}^\star$ of SRIRL (2) under $\tau = \mathrm{T_B}(\hat{\boldsymbol{p}})$ is also an optimal solution of BROIL.

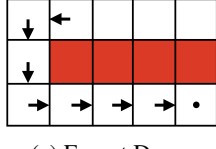 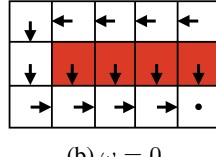 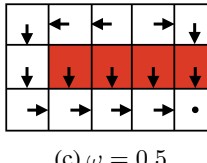 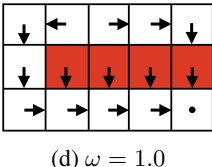

(a) Expert Demo          (b) $\omega = 0$          (c) $\omega = 0.5$          (d) $\omega = 1.0$

Figure 5: Expert demonstration and the policies of BROIL/SRIRL ($\tau = \mathrm{T_B}(\hat{\boldsymbol{p}})$) under different values of the weight parameter $\omega \in \{0, 0.5, 1\}$ in the *lava corridor* environment.

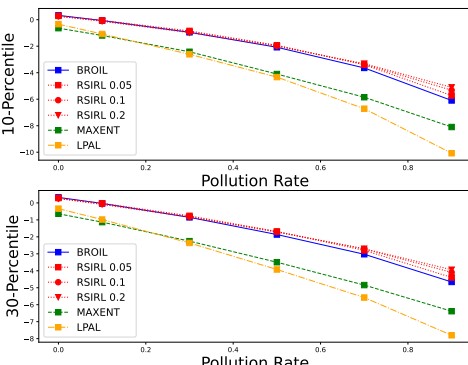 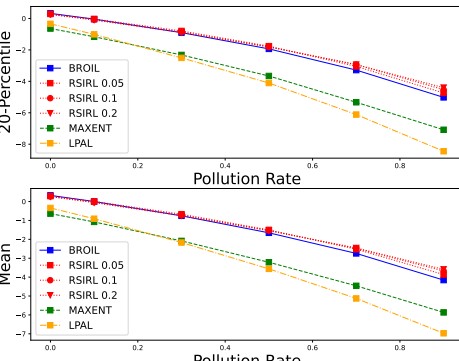

Figure 6: Percentile and average performances in the *lava corridor* application. For each nonzero pollution rate, we evaluate the models over 1000 randomly generated polluted transition kernels. For each kernel, we take the weighted average of the mean (with weight $\omega$) and CVaR (with weight $1-\omega$) of our random instances, where each instance corresponds to a weight sample generated from $\mathbb{P}(\boldsymbol{w} \mid \mathcal{D})$ and evaluates the regret (corresponding the current transition kernel and weights/rewards). Here we set $\omega = 0.5$, and the numbers in the legend are the difference between the optimal value of BROIL and the target parameter $\tau$ in SRIRL.

# G  ADDITIONAL DETAILS OF THE QUADRUPED ROBOT NAVIGATION APPLICATION

Ideally, the tracking error of the neural network controller should be minimal, so the robot's motion dynamics in a 2D plane can be modeled as:

$$x_{t+1} = x_t + v_{x_t}\Delta t, \quad y_{t+1} = y_t + v_{y_t}\Delta t. \tag{46}$$

where $x_t$ and $y_t$ represent the robot's x and y coordinates, and $v_{x_t}$ and $v_{y_t}$ represent the robot's velocity. To model this problem as an MDP, we treat $x_t$ and $y_t$ as states, and $v_{x_t}$ and $v_{y_t}$ as actions, while equation (46) serves as the (deterministic) transition kernel. The initial position of the robot is uniformly distributed, and the navigation target is set as the center of the state space.

It is worth noting that Equation (46) is not entirely realistic for an actual quadruped robot due to: (*i*) the dynamics of a quadruped robot being far more complex than 2D point-mass kinematics, and (*ii*) the neural network motion controller not being able to perfectly achieve the desired speed. Therefore, although (46) is an efficient description of the robot's motion transition kernel, it is not accurate. Similar to the lava corridor experiment, we pollute Equation (46) as follows:

$$x_{t+1} = x_t + (v_{x_t} + w_x)\Delta t, \quad y_{t+1} = y_t + (v_{y_t} + w_y)\Delta t. \tag{47}$$

where $w_x$ and $w_y$ are parameters used to pollute the original deterministic transition kernel, and they are used to compute the ambiguity set $\mathcal{P}$ in SRIRL (2). We remark that, the polluted motion dynamics (47) is only for the purpose of the construction of the support set $\mathcal{P}$ in the SRIRL. Neither the motion dynamics as described in (46) nor the one in (47) are an accurate description of the realistic dynamics of a quadruped robot.

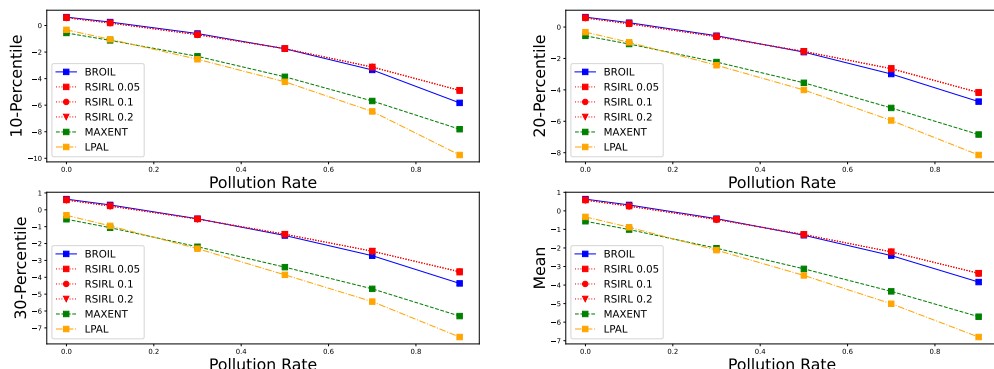

Figure 7: Percentile and average performances in the *lava corridor* application. For each nonzero pollution rate, we evaluate the models over 1000 randomly generated polluted transition kernels. For each kernel, we take the weighted average of the mean (with weight $\omega$) and CVaR (with weight $1-\omega$) of our random instances, where each instance corresponds to a weight sample generated from $\mathbb{P}(\boldsymbol{w} \mid \mathcal{D})$ and evaluates the regret (corresponding the current transition kernel and weights/rewards). Here we set $\omega = 1$, and the numbers in the legend are the difference between the optimal value of BROIL and the target parameter $\tau$ in SRIRL.

## H  ADDITIONAL DETAILS OF EXPERIMENTS ON ALGORITHMS

### H.1  DETAILED SETTINGS

The weight samples are generated as in Section 5.1, and every row of the feature matrix $\boldsymbol{F} \in \mathbb{R}^{SA \times K}$ is set as $(-1, 0)^\top$. The initial distribution is set to be a discrete uniform distribution. The discount factor is set as $\gamma = 0.95$. The entries of the transition kernel $\hat{\boldsymbol{p}}$ are all randomly sampled from a uniform distribution on $[0, 1]$, after which it is normalized so that $\mathbf{e}^\top \hat{\boldsymbol{p}}_{s,a} = 1 \ \forall s \in \mathcal{S}, a \in \mathcal{A}$. We stop our PDA when the change of objective value is less than $0.1\%$, and stop our PDA$_{\text{block}}$ when the maximal number of iterations is reached, which we set to be 6000.

### H.2  ADDTIONAL RESULTS

Table 3: The average computation times (in seconds) of different algorithms for SRIRL for different numbers of weight samples ($N$), the ratios of computation times of Gurobi to those of PDA and PDA$_{\text{block}}$, and the relative gaps to optimal values computed by Gurobi. The average is taken over 50 random instances. We fix $S = A = 10$ throughout all instances.

| | Computation times | | | Ratio of computation times | | Relative gaps (%) | |
|---|---|---|---|---|---|---|---|
| $N$ | Gurobi | PDA | PDA$_{\text{block}}$ | Gurobi/PDA | Gurobi/PDA$_{\text{block}}$ | PDA | PDA$_{\text{block}}$ |
| 10000 | 3.3 | **18.1** | 85.1 | 0.18 | 0.04 | 4.6 | $< 0.1$ |
| 100000 | 47.6 | **18.3** | 84.4 | 2.60 | 0.56 | 4.4 | $< 0.1$ |
| 190000 | 349.7 | **19.1** | 84.9 | 18.31 | 4.12 | 4.6 | $< 0.1$ |
| 280000 | 681.0 | **18.2** | 86.0 | 37.42 | 7.92 | 4.4 | $< 0.1$ |
| 370000 | 1212.8 | **17.6** | 85.7 | 68.91 | 14.15 | 4.3 | $< 0.1$ |

