# OpenReview forum: "Target-Oriented Soft-Robust Inverse Reinforcement Learning"
_ICLR.cc/2025/Conference — ICLR 2025 Conference Withdrawn Submission_

### Official Review · Reviewer_iWW5 · 2024-10-31

**Soundness:** 3
**Presentation:** 3
**Contribution:** 2
**Rating:** 3
**Confidence:** 4

**Summary:**

The paper proposes a method (SRIRL) for inverse reinforcement learning that is risk-averse with respect to reward-ambiguity (by adding an additional objective of maximizing the CVaR) and that is also robust to variations in the transition kernel (by enforcing that the Bellman flow constraints are approximately satisfied for all possible transition kernels, but with an error that can scale linearly with a norm of the difference). SRIRL is based on an LP formulation, assuming discrete states and actions and a reward function that is linear in given features.  Directly solving this optimization problem with an off-the-shelf optimizer (e.g. Gurobi) scales very badly with the number of states and already MDPs with more than 30 states can become infeasible. Hence, the paper also proposes a more efficient optimization using a primal-dual algorithm on a minmax formulation, and to only update a subset of the dual variable at each iteration to further improve the computational efficiency.
The method is evaluated on a lava corridor MDP (with 15 states and 4 actions), and on a 2D velocity-controlled point-mass environment (where the real transitions are generated by a simulated velocity-controlled quadruped). The method is compared with BROIL (which also served as basis for the formulation) and with MaxEnt-IRL and LPAL.

**Strengths:**

- The problem setting (tackling reward ambiguity and dynamics mismatch in IRL) is relevant.
- The problem formulation is novel and sound.
- The paper is sufficiently clear.

**Weaknesses:**

The proposed method is very limited. I don't see how the main contributions (the problem formulation, and the more efficient optimization) could transfer to real world problems. The main limiting assumptions are 1) discrete MDPs 2) linear reward function 3) known (approximate) transition kernel.

The experimental evaluation is quite weak, both regarding the considered environments and also the comparison with baselines. MaxEnt-IRL and LPAL do not address the considered problem setting, leaving only BROIL as a reasonable baseline. While the problem setting is quite specific and therefore a bit niche, it would make sense to independently evaluate the robustness to reward ambiguity and dynamics mismatch.

The scaling experiment, while showing the benefits compared to Gurobi, does only consider quite small MDPs with up to 35 states. It would be interesting to investigate the computation demands for slightly larger MDP. For example, how long would it take to solve a 10x10 Gridworld? Furthermore, the evaluation should also include MaxEnt-IRL and LPAL.

**Questions:**

How was the algorithm applied to the point-mass experiment, which seems to have continuous states and actions?

**Details Of Ethics Concerns:**

No ethics concerns

---

### Official Review · Reviewer_Evik · 2024-11-03

**Soundness:** 3
**Presentation:** 3
**Contribution:** 2
**Rating:** 3
**Confidence:** 3

**Summary:**

This paper introduces a target-oriented soft-robust inverse reinforcement learning (SRIRL) model that balances risk aversion with return maximization in response to reward uncertainty in IRL. The proposed SRIRL framework is also resilient to transition kernel ambiguity by leveraging a robust satisficing approach. A tractable reformulation of the SRIRL algorithm is derived using customized first-order optimization methods, improving scalability compared to commercial solvers. Empirical results demonstrate soft robustness to reward uncertainty and transition kernel ambiguity between the agent and the expert.

**Strengths:**

- The paper addresses an important issue by mitigating the effects of transition kernel ambiguity in IRL.
- It provides a tailored first-order optimization method to enhance the scalability of the SRIRL policy.

**Weaknesses:**

- Related Work: This section is essential to understanding the context and should be expanded in the main text; pseudocode could be moved to the appendix to allow more space.
- Limitations:
    -The approach assumes prior knowledge of optimal values within a domain.
    -Robust satisficing sacrifices some optimality to achieve robustness.
- Empirical Evaluation: The evaluation is limited, with minimal improvement over BROIL, poor presentation of the experimental design, and a lack of comparison with other state-of-the-art IRL algorithms, such as Adversarial IRL, which further reduces the perceived impact and robustness of the results.

**Questions:**

How is the value for \lambda set in the experiments?

---

### Official Review · Reviewer_LzFs · 2024-11-03

**Soundness:** 3
**Presentation:** 3
**Contribution:** 3
**Rating:** 6
**Confidence:** 3

**Summary:**

This paper introduces a novel, softly robust optimization framework for inverse reinforcement learning to tackle the problem of reward uncertainty and transition kernel ambiguity between the expert and policy transition kernel. The framework optimizes the average and CVaR of the return to a user-specified target. The authors reformulate the original objective to a min-max problem and design tailored first-order methods to provide better scalability. Finally, the authors provide empirical results, on a toy task, a robot navigation problem, as well as a scalability study.

**Strengths:**

- I believe the problem addressed is indeed of huge importance. While expert data can be abundant, it almost certainly is collected under (at least slight) differences in dynamics. Having algorithms capable of dealing robustly with these difference is very important, and a promising future direction.
- I particularily like the experimental section, where the authors start with a toy task, pivot then to a more complex task to put more emphasis problems arising when having differences in dynamics, and finally providing a scalability study emphasiszing the need of tailored first-order optimization algorithm.
- The paper is clearly written.

**Weaknesses:**

- The proposed method is fairly complex, and not the most elegant solution. While the results shown are promising, the added complexity could limit future application.
- The approach is limited to discrete state and action spaces, and an extension to continuous spaces seems very challenging due to the already very complex structure of this algorithm in discrete space.

**Questions:**

- How can tau be chosen?
- The quadruped results are a bit confusing. It looks like the quadruped is falling for Broil? Isnt this rather a problem of the low-level policy?

Minor points:
- The paper should be written more compactly. Overviews sections suchs as “We organize the remainder of this paper as follows. We provide necessary preliminaries in Section 2. In Section 3, we study SRIRL and provide its tractable reformulation. Tailored first-order methods for SRIRL are introduced in Section 4. Numerical experiments are conducted in Section 5. A conclusion is drawn in Section 6. D [...]” are not required
- move notation from introduction to preliminary

---

### Official Review · Reviewer_kvb5 · 2024-11-04

**Soundness:** 2
**Presentation:** 2
**Contribution:** 2
**Rating:** 5
**Confidence:** 2

**Summary:**

The submission proposed a target-oriented soft-robust IRL model (so-called SRIRL) which balances the minimization of risk and maximization of expected reward. Another claimed contribution of this work is the robustness of handling transition kernel ambiguity. The proposed method is experimented in a grid-world toy example and a quadruped robot simulation environment.

**Strengths:**

- The proposed method is well-explained with clear notations.
- The authors provide sufficient complexity analysis on the proposed method.
- The idea of balances the minimization of risk and maximization of expected reward is intuitive and practical.

**Weaknesses:**

- Although the algorithm is technically "novel," it's unclear to me whether it effectively tackles any of the existing challenges associated with MaxEnt IRL or similar policy-inference methods. For instance, significant issues in IRL literature include (1) the assumption of Boltzmann rationality of the human demonstrator and (2) the assumption of linear reward features—neither of which are addressed by the proposed method. An additional discussion section on how SRIRL compares to other IRL approaches on the pre mentioned aspects would be helpful.

- It seems to me that the proposed method is only deployable in high-level motion planing tasks with discrete action space. How it compares to other non optimization based methods like BIRL, AIRL, etc.? Looks like these high-level planning tasks are already well-solved with those methods. Could you discuss the relative advantages/disadvantages of your proposed method compared to other non-opt based methods or a quantitative comparison between them?

- I'm somehow unconvinced by the pybullet experiment. The quadruped robot failed, but how can you disentangle the effect of high-level planner (SRIRL method) and low-level NN based controller? Could authors provide an ablation study that isolates the effects of the high-level planner vs. the low-level controller, or additional analysis demonstrating the specific contribution of SRIRL on the level of robot's performance?

**Questions:**

- Can the authors provide harder simulation scenarios? The current experiments seem insufficient to me.
- Can the authors clarify how the method handles cases where, for example, rewards are non-linear and MaxEnt IRL fails to make accurate inferences?
- Misc: Figure 2 caption should be SRIRL instead of RSIRL?

---

### Note · Authors · 2024-11-13

I have read and agree with the venue's withdrawal policy on behalf of myself and my co-authors.